# SAMPLE EFFICIENT STOCHASTIC POLICY EXTRAGRA-DIENT ALGORITHM FOR ZERO-SUM MARKOV GAME

**Ziyi Chen, Shaocong Ma, Yi Zhou**
Department of ECE
University of Utah
Salt Lake City, UT 84112, USA
`{u1276972,yi.zhou,s.ma}@utah.edu`

## ABSTRACT

Two-player zero-sum Markov game is a fundamental problem in reinforcement learning and game theory. Although many algorithms have been proposed for solving zero-sum Markov games in the existing literature, many of them either require a full knowledge of the environment or are not sample-efficient. In this paper, we develop a fully decentralized and sample-efficient stochastic policy extragradient algorithm for solving tabular zero-sum Markov games. In particular, our algorithm utilizes multiple stochastic estimators to accurately estimate the value functions involved in the stochastic updates, and leverages entropy regularization to accelerate the convergence. Specifically, with a proper entropy-regularization parameter, we prove that the stochastic policy extragradient algorithm has a sample complexity of the order $\widetilde{\mathcal{O}}(\frac{A_{\max}}{\mu_{\min}\epsilon^{5.5}(1-\gamma)^{13.5}})$ for finding a solution that achieves $\epsilon$-Nash equilibrium duality gap, where $A_{\max}$ is the maximum number of actions between the players, $\mu_{\min}$ is the lower bound of state stationary distribution, and $\gamma$ is the discount factor. Such a sample complexity result substantially improves the state-of-the-art complexity result.

## 1 INTRODUCTION

Competitive reinforcement learning (RL) is an emerging and popular framework that has broad applications in various areas, including market pricing applications (Kononen and Oja, 2004), real-time strategy-making (Vinyals et al., 2019), board games (Silver et al., 2017; Moerland et al., 2018) and inverse RL (Zhang et al., 2019). In particular, an important and fundamental formulation of competitive RL is the two-player zero-sum Markov game, which involves two competing players that interact with a common environment and receive zero-sum rewards. Both players aim to learn the optimal policy that achieves the Nash equilibrium of accumulated rewards.

Algorithms for solving Markov games are very different from conventional single-agent RL algorithms. In particular, both players must learn to improve their policies based on feedback from the opponent and the environment, but usually the opponent will not reveal any sensitive information (e.g., actions or policy) or cooperate with each other. In the existing literature, numerous algorithms have been developed for solving zero-sum Markov games, including Q-learning (Fan et al., 2020; Zhu and Zhao, 2020), fitted Q iteration (Zhang et al., 2021), policy gradient (Daskalakis et al., 2020; Zhao et al., 2021), policy extragradient (Cen et al., 2021), model-based Monte Carlo estimation (Zhang et al., 2020), optimistic gradient descent ascent (Wei et al., 2021), etc. However, many of these algorithms require both players to access their opponent's actions (Wei et al., 2017; Sidford et al., 2020; Huang et al., 2021; Jafarnia-Jahromi et al., 2021), which causes privacy issues. On the other hand, some algorithms need to know about the environment transition kernel and reward mapping (Cen et al., 2021), which are usually unknown a priori in practice. Moreover, other algorithms require both players to perform asymmetric policy updates using different numbers of iterations, learning rates or exploration probabilities (Zhao et al., 2021; Daskalakis et al., 2020), which are generally

hard to coordinate in advance between two competing players. Therefore, it is desired to develop an algorithm for solving Markov games that avoids all the aforementioned issues.

Moreover, from a theoretical perspective, the convergence and sample complexity of these existing algorithms for solving Markov games have not been comprehensively studied. Specifically, some studies established the convergence of the algorithms with i.i.d. samples (Zhao et al., 2021; Guo et al., 2021), which violates the dependent nature of samples collected from the dynamic Markov decision process. Also, other algorithms suffer from an extremely high sample complexity to achieve an approximate Nash equilibrium solution (Wei et al., 2021). Hence, we are motivated to answer the following fundamental question.

- **Q:** *Can we develop a fully decentralized algorithm that is model-free and takes symmetric and private policy updates for solving zero-sum Markov games, with provable convergence guarantee and an improved sample complexity for achieving a Nash equilibrium solution?*

In this paper, we provide positive and comprehensive answers to this question by developing a fully decentralized stochastic policy extragradient algorithm. In Table 1 of Appendix F, we compare the key properties and sample complexity of our algorithm with those of the existing algorithms.

## 1.1 OUR CONTRIBUTIONS

We consider a standard zero-sum Markov game with discounted reward over infinite horizon. To solve such a Markov game, we propose a stochastic variant of the policy extragradient algorithm (Cen et al., 2021) that satisfies the following amenable properties.

- Our algorithm uses multiple stochastic estimators to estimate the value functions involved in the predictive updates for solving entropy-regularized matrix games, and therefore the algorithm does not rely on any prior knowledge of the environment transition kernel (model-free). Moreover, the resulting stochastic policy updates of our algorithm for both players are symmetric and do not involve any sensitive information of the opponent (private).

- Compared with the stochastic estimators used in (Wei et al., 2021), our estimators have much smaller variance that helps improve the estimation accuracy. Specifically, by developing new techniques (explained in the next bullet), we establish tight high-probability estimation error bounds for our stochastic estimators with Markovian samples. Then, with a proper entropy-regularization parameter, we show that our stochastic policy extragradient algorithm requires a sample complexity of the order $\mathcal{O}\big(\frac{A_{\max}}{\mu_{\min}\epsilon^{5.5}(1-\gamma)^{13.5}}\big)$ to achieve an $\epsilon$-Nash equilibrium duality gap, which substantially improves the state-of-the-art complexity result of (Wei et al., 2021).

- We develop novel techniques to bound the estimation error of the proposed stochastic estimators, whose numerator and denominator involve sample average approximations. First, we propose a special estimation error decomposition that avoids divergence of the bound caused by possibly small numerical values of the sample average involved in the denominator of the stochastic estimators. Second, we leverage this error decomposition and the recursive structure of the stochastic estimators to derive a contraction property of the estimation errors, which eventually leads to tight bounds for the estimation error. We refer to Section 4 for more elaboration on our technical novelties.

## 1.2 OTHER RELATED WORK

**Other settings of two-player zero-sum Markov games.** In this paper, we focus on a standard setting of two-player zero-sum Markov game with discount and infinite horizon in the discrete time domain. There are other settings of two-player zero-sum Markov games. For example, Bai et al. (2020); Huang et al. (2021) studied a two-player zero-sum Markov game with finite horizon and without discount, whereas Daskalakis et al. (2020) considered finite random horizon without discount. Jafarnia-Jahromi et al. (2021) also considered the setting without discount, and it allows one of the players to constantly adjust its policy based on the entire history of states and actions. Ghosh et al. (2021) studied a two-player zero-sum Markov game in the continuous time domain.

**Multi-agent general-sum Markov game.** Some works studied multi-agent Markov games, which extend the two-player zero-sum Markov games to multiple players without the zero-sum constraint (Wang and Sandholm, 2002; Hu and Wellman, 2003; Deng et al., 2021; Leonardos et al., 2021). More specifically, Leonardos et al. (2021) defined and studied Markov potential game which has a potential

function assumption. Guo et al. (2019); Elie et al. (2020); Gu et al. (2021) studied mean-field games with a large number of players.

**Entropy regularization and value iteration** Our algorithm leverages entropy regularization and value iteration to accelerate convergence. Entropy regularization is a popular technique that has been widely used in reinforcement learning (Neu et al., 2017; Geist et al., 2019; Mei et al., 2020; Cen et al., 2020) and Markov game (Mertikopoulos and Sandholm, 2016; Savas et al., 2019; Cen et al., 2021) to encourage environment exploration and accelerate algorithm convergence. Value iteration is also a classical method that is widely used in both single-agent reinforcement learning (Ernst et al., 2005; Tamar et al., 2016; Farahmand and Ghavamzadeh, 2021) and Markov games (Zhu and Zhao, 2020; Cen et al., 2021). With full knowledge of the environment, it exponentially converges to the fixed point of Bellman operator (Cen et al., 2021). Compared to another similar classical method called policy iteration, value iteration does not need policy evaluation which involves additional computation (Sutton and Barto, 2018).

## 2 BACKGROUND OF MARKOV GAME AND ENTROPY REGULARIZATION

### 2.1 TWO-PLAYER ZERO-SUM MARKOV GAME

In a zero-sum Markov game, two players compete with each other in a common environment. Throughout, the state space is denoted as $\mathcal{S}$. The action spaces and policies of both players are denoted as $\mathcal{A}^{(1)}, \pi^{(1)}$ and $\mathcal{A}^{(2)}, \pi^{(2)}$, respectively. Here, $\pi^{(1)} \in \Delta(|\mathcal{A}^{(1)}|), \pi^{(2)} \in \Delta(|\mathcal{A}^{(2)}|)$ are random policies defined over the corresponding simplex sets. The reward function is denoted as $R : \mathcal{S} \times \mathcal{A}^{(1)} \times \mathcal{A}^{(2)} \to [0, 1]$, and the discount factor is denoted as $\gamma \in (0, 1)$.

At any time $t$, both players observe state $s_t \in \mathcal{S}$ of the environment. Then, both players respectively select their actions following their own policies, i.e., $a_t^{(1)} \sim \pi^{(1)}(\cdot|s_t)$ and $a_t^{(2)} \sim \pi^{(2)}(\cdot|s_t)$. After that, the environment state transfers to a new state $s_{t+1}$ following the underlying transition kernel $\mathcal{P}(\cdot|s_t, a_t^{(1)}, a_t^{(2)})$, and both players receive zero-sum rewards, i.e., $R_t^{(1)} = -R_t^{(2)} = R_t$, where $R_t := R(s_t, a_t^{(1)}, a_t^{(2)})$. With this Markov decision process, we can define the following state value function associated with the players' policies $\pi^{(1)}$ and $\pi^{(2)}$ for any environment state $s \in \mathcal{S}$.

$$V_{\pi^{(1)}, \pi^{(2)}}(s) = \mathbb{E}\Big[ \sum_{t=0}^{\infty} \gamma^t R_t \Big| s_0 = s \Big]. \tag{1}$$

The goal of both players is to compete via the following minimax game in all states $s$.

$$\min_{\pi^{(2)}} \max_{\pi^{(1)}} V_{\pi^{(1)}, \pi^{(2)}}(s). \tag{2}$$

In particular, it has been shown in (Shapley, 1953) that there exists a Nash equilibrium policy pair $\pi_*^{(1)}, \pi_*^{(2)}$ for zero-sum Markov games, i.e., $V_{\pi^{(1)}, \pi_*^{(2)}}(s) \le V_{\pi_*^{(1)}, \pi_*^{(2)}}(s) \le V_{\pi_*^{(1)}, \pi^{(2)}}(s)$ holds for any other policies $\pi^{(1)}, \pi^{(2)}$ and for all states $s$.

### 2.2 ENTROPY-REGULARIZED MARKOV GAME

Entropy regularization is a popular technique that has been widely used in reinforcement learning (Neu et al., 2017; Geist et al., 2019; Mei et al., 2020; Cen et al., 2020) and Markov game (Mertikopoulos and Sandholm, 2016; Savas et al., 2019; Cen et al., 2021) to encourage environment exploration and accelerate algorithm convergence.

Specifically, for the zero-sum Markov game, we can define an entropy-regularized state value function by adding entropy regularization to the state value function in (1) as follows (Cen et al., 2021).

$$V_{\pi^{(1)}, \pi^{(2)}}^{(\tau)}(s) := \mathbb{E}\Big[ \sum_{t=0}^{\infty} \gamma^t \big[ R_t - \tau \ln \pi^{(1)}(a_t^{(1)}|s_t) + \tau \ln \pi^{(2)}(a_t^{(2)}|s_t) \big] \Big| s_0 = s \Big], \tag{3}$$

where $\tau > 0$ is called the regularization parameter. With the above definition, we further define the following entropy-regularized state-action value function (also called $Q$-function) (Cen et al., 2021).

$$Q_{\pi^{(1)}, \pi^{(2)}}^{(\tau)}(s, a^{(1)}, a^{(2)}) := R(s, a^{(1)}, a^{(2)}) + \gamma \mathbb{E}_{s' \sim \mathcal{P}(\cdot|s, a^{(1)}, a^{(2)})} \big[ V_{\pi^{(1)}, \pi^{(2)}}^{(\tau)}(s') \big]. \tag{4}$$

In particular, $V_{\pi^{(1)},\pi^{(2)}}^{(\tau)}$ can be obtained from $Q_{\pi^{(1)},\pi^{(2)}}^{(\tau)}$ as follows.

$$
\begin{aligned}
V_{\pi^{(1)},\pi^{(2)}}^{(\tau)}(s) =& [\pi^{(1)}(s)]^\top Q_{\pi^{(1)},\pi^{(2)}}^{(\tau)}(s)\pi^{(2)}(s) + \tau\mathcal{H}\big(\pi^{(1)}(s)\big) - \tau\mathcal{H}\big(\pi^{(2)}(s)\big) \\
:=& f_\tau\big(Q_{\pi^{(1)},\pi^{(2)}}^{(\tau)}(s), \pi^{(1)}(s), \pi^{(2)}(s)\big),
\end{aligned}
\tag{5}
$$

where $\mathcal{H}(\pi)$ denotes the entropy of policy $\pi$, and we define this mapping as $f_\tau$ for convenience.

For the entropy regularized Markov game, it has an equilibrium policy pair that solves the minimax optimization problem $\min_{\pi^{(2)}} \max_{\pi^{(1)}} V_{\pi^{(1)},\pi^{(2)}}^{(\tau)}(s)$. Such a policy pair is called the quantal response equilibrium (QRE). Our goal is to find the equilibrium policy pair of the original Markov game in (2) by solving the entropy-regularized Markov game with a proper regularization parameter $\tau$. In particular, compared with the equilibrium policy of the Markov game, the QRE tends to have a larger entropy due to the entropy regularization, which encourages the players to explore and obtain a better understanding of the environment. Another advantage of considering the entropy-regularized Markov game is that the entropy regularization makes the minimax problem have a better optimization geometry that accelerates the convergence of the optimization process.

# 3 STOCHASTIC POLICY EXTRAGRADIENT ALGORITHM FOR ENTROPY-REGULARIZED MARKOV GAME

In this section, we develop a stochastic policy extragradient (SPE) algorithm for solving entropy-regularized Markov games. First, we recap the policy extragradient (PE) algorithm, which is introduced in (Cen et al., 2021) to solve entropy-regularized Markov games with full knowledge of the environment transition kernel and reward mapping. Then, we propose the model-free SPE algorithm that solves entropy-regularized Markov games using only stochastic samples.

## 3.1 REVIEW OF POLICY EXTRAGRADIENT ALGORITHM

Value iteration is a classical reinforcement learning algorithm that requires full knowledge of the environment and achieves an exponential convergence rate. In particular, for the entropy-regularized Markov game, the $k$-th value iteration update is defined as follows.

$$
Q_k(s, a^{(1)}, a^{(2)}) = R(s, a^{(1)}, a^{(2)}) + \gamma\mathbb{E}_{s'\sim\mathcal{P}(\cdot|s,a^{(1)},a^{(2)})}\big[V_k(s')\big], \quad \forall s, a^{(1)}, a^{(2)}, \tag{6}
$$

$$
V_{k+1}(s) = \min_{\pi^{(2)}(s)} \max_{\pi^{(1)}(s)} f_\tau\big(Q_k(s); \pi^{(1)}(s), \pi^{(2)}(s)\big), \quad \forall s, \tag{7}
$$

where we define $Q_k(s) := Q_k(s, \cdot, \cdot) \in \mathbb{R}^{|\mathcal{A}^{(1)}|\times|\mathcal{A}^{(2)}|}$. This algorithm alternatively updates all the entries of the value functions $Q_k$ and $V_k$. Thanks to the entropy regularization in the function $f_\tau$ (see (5) for the definition), the minimax matrix game in (7) is $\tau$-strongly concave in $\pi^{(1)}$ and $\tau$-strongly convex in $\pi^{(2)}$, and therefore it has a unique solution.

To solve the entropy-regularized minimax matrix game in (7), Cen et al. (2021) proposed a predictive update (PU) algorithm. Specifically, with uniform policy initialization, i.e., $\pi_{k,0}^{(m)}(s) = \frac{1}{|\mathcal{A}^{(m)}|}, \forall m \in \{1, 2\}, \forall s \in \mathcal{S}$, the PU algorithm performs the following policy updates: for $t = 0, 1, 2, \dots$

$$
\text{(PU):} \begin{cases}
\overline{\pi}_{k,t+1}^{(1)}(a^{(1)}|s) \propto \pi_{k,t}^{(1)}(a^{(1)}|s)^{1-\eta\tau} \exp\big(\eta Q_{k,t}^{(1)}(s, a^{(1)})\big) \\
\overline{\pi}_{k,t+1}^{(2)}(a^{(2)}|s) \propto \pi_{k,t}^{(2)}(a^{(2)}|s)^{1-\eta\tau} \exp\big(-\eta Q_{k,t}^{(2)}(s, a^{(2)})\big) \\
\pi_{k,t+1}^{(1)}(a^{(1)}|s) \propto \pi_{k,t}^{(1)}(a^{(1)}|s)^{1-\eta\tau} \exp\big(\eta\overline{Q}_{k,t+1}^{(1)}(s, a^{(1)})\big) \\
\pi_{k,t+1}^{(2)}(a^{(2)}|s) \propto \pi_{k,t}^{(2)}(a^{(2)}|s)^{1-\eta\tau} \exp\big(-\eta\overline{Q}_{k,t+1}^{(2)}(s, a^{(2)})\big)
\end{cases}, \tag{8}
$$

where we use the following notations (superscript $(\backslash m)$ denotes the opponent of the $m$-th player.).

$$
Q_{k,t}^{(m)}(s, a^{(m)}) := \mathbb{E}_{a^{(\backslash m)}\sim\pi_{k,t}^{(\backslash m)}(s)}\big[Q_k(s, a^{(1)}, a^{(2)})\big], \quad m \in \{1, 2\} \tag{9}
$$

$$
\overline{Q}_{k,t+1}^{(m)}(s, a^{(m)}) := \mathbb{E}_{a^{(\backslash m)}\sim\overline{\pi}_{k,t+1}^{(\backslash m)}(s)}\big[Q_k(s, a^{(1)}, a^{(2)})\big], \quad m \in \{1, 2\}. \tag{10}
$$

Once we obtain the output policy pair $(\pi_k^{(1)}, \pi_k^{(2)})$ of the PU algorithm, we can obtain an approximation of $V_{k+1}(s)$ as $V'_{k+1}(s) = f_\tau\big(Q_k(s); \pi_k^{(1)}(s), \pi_k^{(2)}(s)\big)$, which will be further used in the next $Q$-value function update (6) to replace $V_{k+1}(s')$. The updates (6), (7) & (8) are referred to as policy extragradient (PE) algorithm.

In the PE algorithm, the PU update in (8) allows both players to take symmetric updates without revealing their private actions, and has been shown to converge to the unique solution of the entropy-regularized matrix game (7) exponentially fast (Cen et al., 2021). However, the PE algorithm has several limitations. First, in the PU update, each player $m \in \{1, 2\}$ needs to query the quantities $Q_{k,t}^{(m)}(s, a^{(m)}), \overline{Q}_{k,t}^{(m)}(s, a^{(m)})$ from its opponent. To compute these quantities, the opponent needs to multiply the entire $Q$-table by its own policy vector. This requires both players to coordinate with each other and share a $Q$-table. Second, the update of the $Q$-table in (6) requires full knowledge of the environment transition kernel $\mathcal{P}$ and the reward mapping $R$, which are unknown a priori in practice. To overcome these limitations, we develop a fully stochastic PE algorithm in the next subsection.

## 3.2 STOCHASTIC POLICY EXTRAGRADIENT ALGORITHM

The major challenge of the PE algorithm is computing the quantities $Q_{k,t}^{(m)}, \overline{Q}_{k,t}^{(m)}$ and $V'_{k+1}$, which requires coordinating with the opponent and involves the environment information. Here, we develop a model-free and fully stochastic variant of PE that estimates these key quantities using Markovian stochastic samples. We refer to this algorithm as **stochastic policy extragradient (SPE)**.

Specifically, we first estimate the quantity $V'_{k+1}(s) = f_\tau\big(Q_k(s); \pi_k^{(1)}(s), \pi_k^{(2)}(s)\big)$. By definition of $f_\tau$ in (5) and the update of $Q_k$ in (6) (now we use $V'_k(s')$ instead of $V_k(s')$) and using some standard tricks on random variables (see Lemma 2 in Appendix B for a full proof), we can rewrite $V'_{k+1}(s)$ as

$$V'_{k+1}(s) = \frac{\mathbb{E}\Big[\big(R(\widetilde{s}, \widetilde{a}^{(1)}, \widetilde{a}^{(2)}) + \gamma V'_k(s')\big)\mathbb{1}\{\widetilde{s} = s\}\Big]}{\mu_k(s)} + \tau\mathcal{H}\big(\pi_k^{(1)}(s)\big) - \tau\mathcal{H}\big(\pi_k^{(2)}(s)\big), \quad (11)$$

where $\mu_k(s)$ denotes the stationary state distribution associated with the policy pair $(\pi_k^{(1)}, \pi_k^{(2)})$, and the expectation is taken over $\widetilde{s} \sim \mu_k, \widetilde{a}^{(1)} \sim \pi_k^{(1)}(s), \widetilde{a}^{(2)} \sim \pi_k^{(2)}(s), s' \sim \mathcal{P}(\cdot|\widetilde{s}, \widetilde{a}^{(1)}, \widetilde{a}^{(2)})$. To estimate this quantity, we query a set $\mathcal{N}_{k+1}$ (with cardinality $N_{k+1}$) of samples from the Markov decision process following the pair of policies $(\pi_k^{(1)}, \pi_k^{(2)})$. Then, we estimate $V'_{k+1}(s)$ as

$$\widehat{V}_{k+1}(s) = \frac{\frac{1}{N_{k+1}}\sum_{i \in \mathcal{N}_{k+1}}\big(R_i + \gamma\widehat{V}_k(s_{i+1})\big)\mathbb{1}\{s_i = s\}}{\frac{1}{N_{k+1}}\sum_{i \in \mathcal{N}_{k+1}}\mathbb{1}\{s_i = s\}} + \tau\mathcal{H}\big(\pi_k^{(1)}(s)\big) - \tau\mathcal{H}\big(\pi_k^{(2)}(s)\big). \quad (12)$$

Intuitively, we use the sample average of Markovian samples to estimate the expectation terms in (11). Thanks to the concentration phenomenon of dependent samples (Paulin, 2015), these sample averages converge to the desired expected values provided that the sample size is sufficiently large.

Next, we estimate $Q_{k,t}^{(m)}, m \in 1, 2$. Leveraging (9) and (6), we obtain the following equivalent characterization for both players $m \in 1, 2$ (see Lemma 2 in Appendix B for the proof of equivalence).

$$Q_{k,t}^{(m)}(s, a^{(m)}) = \frac{\mathbb{E}\Big[\big(R(\widetilde{s}, \widetilde{a}^{(1)}, \widetilde{a}^{(2)}) + \gamma V'_k(s')\big)\mathbb{1}\{\widetilde{s} = s, \widetilde{a}^{(m)} = a^{(m)}\}\Big]}{\mu_{k,t}(s)\pi_{k,t}^{(m)}(a^{(m)}|s)}, \quad (13)$$

where $\mathbb{1}\{\cdot\}$ denotes the indicator function, $\mu_{k,t}$ denotes the stationary state distribution associated with the policy pair $(\pi_{k,t}^{(1)}, \pi_{k,t}^{(2)})$, and the expectation is taken over $\widetilde{s} \sim \mu_{k,t}, \widetilde{a}^{(1)} \sim \pi_{k,t}^{(1)}(s), \widetilde{a}^{(2)} \sim \pi_{k,t}^{(2)}(s), s' \sim \mathcal{P}(\cdot|\widetilde{s}, \widetilde{a}^{(1)}, \widetilde{a}^{(2)})$. To estimate this quantity, we query a set $\mathcal{N}_{k,t}$ (with cardinality $N_{k,t}$) of samples from the Markov decision process following a pair of *smoothed policies* $\pi_{k,t}^{'(m)}(s) = (1 - \epsilon')\pi_{k,t}^{(m)}(s) + \frac{\epsilon'}{|\mathcal{A}^{(m)}|}\mathbf{1}$, where $\epsilon' \in [0, 1]$ is a small smoothing constant that will be theoretically determined later. Then, we estimate $Q_{k,t}^{(m)}(s, a^{(m)})$ as follows.

$$\widehat{Q}_{k,t}^{(m)}(s, a^{(m)}) = \frac{\frac{1}{N_{k,t}}\sum_{i \in \mathcal{N}_{k,t}}\big(R_i + \gamma\widehat{V}_k(s_{i+1})\big)\mathbb{1}\{s_i = s, a_i^{(m)} = a^{(m)}\}}{\big(\frac{1}{N_{k,t}}\sum_{i \in \mathcal{N}_{k,t}}\mathbb{1}\{s_i = s\}\big)\pi_{k,t}^{'(m)}(a^{(m)}|s)}, \quad (14)$$

where we have replaced the expectations with sample averages, and replaced $V_k'(s')$ with $\widehat{V}_k(s')$. Here, the Markovian samples are queried following the $\epsilon'$-smoothed policies $(\pi_{k,t}'^{(1)}, \pi_{k,t}'^{(2)})$. On one hand, $\epsilon'$ should not be too small so that it keeps the denominator of the above estimation away from zero. On the other hand, $\epsilon'$ should not be too large so that it is sufficiently close to the original policy.

Similarly, to estimate $\overline{Q}_{k,t}^{(m)}$, we query another set $\overline{\mathcal{N}}_{k,t}$ (with cardinality $\overline{N}_{k,t}$) of samples from the Markov decision process following a pair of smoothed policies $\overline{\pi}_{k,t}'^{(m)}(s) = (1 - \overline{\epsilon}')\overline{\pi}_{k,t}^{(m)}(s) + \frac{\overline{\epsilon}'}{|\mathcal{A}^{(m)}|}\mathbf{1}$, where $\overline{\epsilon}' \in [0, 1]$ will be theoretically determined later. Then, we estimate $\overline{Q}_{k,t}^{(m)}$ as follows.

$$\widehat{\overline{Q}}_{k,t}^{(m)}(s, a^{(m)}) = \frac{\frac{1}{\overline{N}_{k,t}} \sum_{i \in \overline{\mathcal{N}}_{k,t}} \left(R_i + \gamma \widehat{V}_k(s_{i+1})\right) \mathbb{1}\{s_i = s, a_i^{(m)} = a^{(m)}\}}{\left(\frac{1}{\overline{N}_{k,t}} \sum_{i \in \overline{\mathcal{N}}_{k,t}} \mathbb{1}\{s_i = s\}\right) \overline{\pi}_{k,t}'^{(m)}(a^{(m)}|s)}. \tag{15}$$

**Remark 1.** *The estimators* (12), (14) *&* (15) *essentially use ratio of sample averages to approximate ratio of expectations. Estimators with similar structure have been used in (*Ortner and Auer, 2007*; *Xia et al., 2016*; *Wei et al., 2021*). However, these works analyze the estimation error of their estimators with independence samples. As a comparison, our analysis of the estimation error bounds the additional bias induced by the correlated Markovian samples, and achieves an improved sample complexity in the main Theorem* 2.

We summarize our stochastic policy extragradient (SPE) algorithm in Algorithm 1. Specifically, in SPE, we estimate the quantities $Q_{k,t}^{(m)}, \overline{Q}_{k,t}^{(m)}, V_k'$ using their corresponding stochastic estimators. As a result, the SPE algorithm is model-free, and the updates for both players are symmetric and private.

---

**Algorithm 1** Stochastic policy extragradient (SPE) for entropy-regularized Markov game

**Initialize:** $V_0'(s)$ for all $s \in \mathcal{S}$.
**for** *value iterations* $k = 0, 1, \ldots, K - 1$ **do**
  Initialize $\pi_{k,0}^{(1)}, \pi_{k,0}^{(2)}$ with uniform distribution.
  **for** *PU iterations* $t = 0, 1, \ldots, T_k - 1$ **do**
    Players 1,2 sample $N_{k,t}$ Markovian samples following smoothed policies $\pi_{k,t}'^{(1)}, \pi_{k,t}'^{(2)}$.
    Every player $m \in \{1, 2\}$ computes $\widehat{Q}_{k,t}^{(m)}(s, a^{(m)})$ for all $s, a^{(m)}$ using (14).
    Players 1,2 sample $\overline{N}_{k,t}$ Markovian samples following smoothed policies $\overline{\pi}_{k,t}'^{(1)}, \overline{\pi}_{k,t}'^{(2)}$.
    Every player $m \in \{1, 2\}$ computes $\widehat{\overline{Q}}_{k,t}^{(m)}(s, a^{(m)})$ for all $s, a^{(m)}$ using (15).
    Implement the $t$-th PU iteration for all $s, a^{(1)}, a^{(2)}$ using (8) with estimations (14)&(15).
  **end**
  Let $\pi_k^{(m)} = \overline{\pi}_{k,T_k}^{(m)}$, $m = 1, 2$. Players sample $N_k$ Markovian samples following $\pi_k^{(1)}, \pi_k^{(2)}$.
  Compute $\widehat{V}_{k+1}(s)$ for all $s$ using (12).
**end**
**Output:** $\pi_{K-1}^{(1)}, \pi_{K-1}^{(2)}$.

---

## 4  FINITE-TIME CONVERGENCE ANALYSIS OF SPE

Throughout our convergence analysis, we adopt the following two standard assumptions.

**Assumption 1.** *Denote* $T_{\pi^{(1)}, \pi^{(2)}}(s, s') := \inf\{t \geq 1 : s_t = s' | s_0 = s\}$ *as the first-visit time under the policy pair* $\pi^{(1)}, \pi^{(2)}$. *We assume that*

$$\sup_{s, s' \in \mathcal{S}} \sup_{\pi^{(1)}, \pi^{(2)}} \mathbb{E}_{\pi^{(1)}, \pi^{(2)}}\left[T_{\pi^{(1)}, \pi^{(2)}}(s, s')\right] < +\infty. \tag{16}$$

Assumption 1 is widely used in the reinforcement learning literature (Ortner and Auer, 2007; Ortner, 2020; Wei et al., 2021; Jafarnia-Jahromi et al., 2021). It ensures that every state will be visited at least once within a finite duration of time, thus ensuring that all the states will be visited infinitely often.

This guarantees sufficient exploration. In our analysis, we use the following equivalent statement of Assumption 1 for convenience, which means that the stationary state distribution $\mu_{\pi^{(1)},\pi^{(2)}}$ has a uniform lower bound $\mu_{\min} > 0$. Their equivalence is based on Theorem 5.5.11 of (Durrett, 2019).

$$\mu_{\min} := \inf_{s \in \mathcal{S}} \inf_{\pi^{(1)},\pi^{(2)}} \mu_{\pi^{(1)},\pi^{(2)}}(s) = \left[ \sup_{s \in \mathcal{S}} \sup_{\pi^{(1)},\pi^{(2)}} \mathbb{E}_{\pi^{(1)},\pi^{(2)}} T_{\pi^{(1)},\pi^{(2)}}(s,s) \right]^{-1} > 0. \quad (17)$$

**Assumption 2.** *There exists a mixing time $t_{mix} \in \mathbb{N}$ such that for any policy pair $\pi^{(1)}, \pi^{(2)}$ and its corresponding stationary state distribution $\mu_{\pi^{(1)},\pi^{(2)}}$, we have*

$$d_{\mathrm{TV}}\left( \mathcal{P}_{\pi^{(1)},\pi^{(2)}}(s_{t_{mix}}), \mu_{\pi^{(1)},\pi^{(2)}} \right) \leq \frac{1}{4}.$$

*where $\mathcal{P}_{\pi^{(1)},\pi^{(2)}}(s_{t_{mix}})$ denotes the state distribution under the policy pair $\pi^{(1)}, \pi^{(2)}$ at time $t_{mix}$, and $d_{\mathrm{TV}}$ denotes the total variation distance between two probability distributions.*

In this subsection, we analyze the finite-time convergence of Algorithm 1 for solving the entropy-regularized Markov game (5). We focus on the convergence rate of the following Nash equilibrium duality gap, which is a standard optimality metric widely adopted in the existing literature (Xu et al., 2020; Jin and Sidford, 2021; Wei et al., 2021).

$$D^{(\tau)}(\pi^{(1)}, \pi^{(2)}) := \max_s \left( \max_\pi V^{(\tau)}_{\pi,\pi^{(2)}}(s) - \min_{\pi'} V^{(\tau)}_{\pi^{(1)},\pi'}(s) \right).$$

In particular, when $\tau = 0$, $D^{(0)}(\pi^{(1)}, \pi^{(2)})$ corresponds to the duality gap of the original Markov game. Throughout, we define $A_{\max} := \max\{|\mathcal{A}^{(1)}|, |\mathcal{A}^{(2)}|\}$, $Q_{\max} := \frac{1+\gamma\tau \ln A_{\max}}{1-\gamma}$ and $V_{\max} := \frac{1+\tau \ln A_{\max}}{1-\gamma}$. We also require that the batch sizes of Algorithm 1 satisfy the following conditions.

$$N_{k,t}, \overline{N}_{k,t} \geq \frac{650 t_{\mathrm{mix}} A_{\max}}{\mu_{\min}} \ln\left( \frac{20 T_{\mathrm{sum}} |\mathcal{S}| A_{\max}}{\delta \sqrt{\mu_{\min}}} \right), \quad \forall k, t, \quad (18)$$

$$N_{k+1} \geq \frac{650 t_{\mathrm{mix}}}{\mu_{\min}(1-\gamma)^2} \ln\left( \frac{4}{\delta \sqrt{\mu_{\min}}} \right), \quad \forall k. \quad (19)$$

Then, we obtain the following convergence result of Algorithm 1, where $T_{\mathrm{sum}} := \sum_{k=0}^{K-1} T_k$.

**Theorem 1** (Finite-time convergence rate). *Apply Algorithm 1 to solve the entropy-regularized Markov game with $\tau \in (0, 1]$. Choose learning rate $\eta = [2(\tau + Q_{\max})]^{-1}$, initialization $\|\widehat{V}_0\|_\infty \leq V_{\max}$ and batch sizes $N_{k,t}, \overline{N}_{k,t}, N_{k+1}$ that satisfy (18) & (19). Then, the Nash equilibrium duality gap converges at the following rate with probability at least $1 - \delta$.*

$$D^{(\tau)}\left(\pi^{(1)}_{K-1}, \pi^{(2)}_{K-1}\right) \leq \mathcal{O}\Bigg( \frac{V_{\max} \ln A_{\max}}{1-\gamma} \sum_{k=0}^{K-1} \gamma^{K-k}(1-\eta\tau)^{T_k-1}$$

$$+ \frac{V_{\max}}{1-\gamma} \left[ \frac{t_{mix} A_{\max}}{\mu_{\min}} \ln\left( \frac{T_{sum}|\mathcal{S}|A_{\max}}{\delta \mu_{\min}} \right) \right]^{2/3} \sum_{k=0}^{K-1} \gamma^{K-k-1} \sum_{t=0}^{T_k-1} (1-\eta\tau)^{T_k-2-t} \left( \frac{1}{N_{k,t}^{2/3}} + \frac{V_{\max}}{\tau \overline{N}_{k,t+1}^{2/3}} \right)$$

$$+ \frac{V_{\max}^3 \gamma^K}{\tau^2(1-\gamma)^2} + \frac{t_{mix}V_{\max}^3}{\tau^2 \mu_{\min}(1-\gamma)^3} \ln\left( \frac{K|\mathcal{S}|}{\delta\mu_{\min}} \right) \sum_{k=0}^{K-1} \frac{\gamma^{K-k-1}}{N_{k+1}} \Bigg). \quad (20)$$

**Remark 2.** *In the proof of Theorem 1, we also prove that the convergence rate of the Q-function estimation error $\|Q_K - Q_*^{(\tau)}\|_\infty$ is $(1-\gamma)$ times the convergence rate in (20). Here, $Q_K$ corresponds to the Q-function associated with the policy pair $(\pi^{(1)}_K, \pi^{(2)}_K)$ produced by Algorithm 1 in the $K$-th iteration, and $Q_*^{(\tau)}$ corresponds to the optimal Q-function associated with the Nash equilibrium policy pair $\pi^{(1)}_{*\tau}, \pi^{(2)}_{*\tau}$ of the entropy-regularized Markov game.*

Theorem 1 characterizes the convergence of duality gap of the SPE algorithm under general hyperparameter scheduling of the batch sizes $N_{k,t}, \overline{N}_{k,t}, N_{k+1}$ and number of inner iterations $T_k$. Specifically, it can be seen that as the number of outer iterations $K$ and inner iterations $T_k$ increase, the duality gap converges to an exponentially weighted average of $N_{k,t}^{-2/3}$, $\overline{N}_{k,t+1}^{-2/3}$ and $N_{k+1}^{-1}$, and the

gap can be made arbitrarily small by choosing sufficiently large batch sizes. To provide an intuitive understanding, we can set these hyper-parameters as constants, i.e., $T_k \equiv T$, $N_{k,t} \equiv N$, $\overline{N}_{k,t} \equiv \overline{N}$, $N_{k+1} \equiv N'$, and then the convergence rate in (20) simplifies to

$$\mathcal{O}\left( \frac{V_{\max}^3 \gamma^K}{\tau^2(1-\gamma)^2} + \frac{\gamma V_{\max} \ln A_{\max}}{(1-\gamma)^2}(1-\eta\tau)^{T-1} + \frac{t_{\mathrm{mix}}V_{\max}^3}{\tau^2 \mu_{\min}(1-\gamma)^4} \ln\left(\frac{K|\mathcal{S}|}{\delta\mu_{\min}}\right)\frac{1}{N} \right.$$

$$\left. + \frac{V_{\max}}{\eta\tau(1-\eta\tau)(1-\gamma)^2}\left[\frac{t_{\mathrm{mix}}A_{\max}}{\mu_{\min}}\ln\left(\frac{T_{\mathrm{sum}}|\mathcal{S}|A_{\max}}{\delta\mu_{\min}}\right)\right]^{2/3}\left(\frac{1}{N^{2/3}} + \frac{V_{\max}}{\tau\overline{N}^{2/3}}\right)\right). \qquad (21)$$

To elaborate, the first term characterizes the exponential convergence of the $K$ outer value iterations. This convergence is faster than the sublinear convergence $\mathcal{O}(\sqrt{\ln K/K})$ established in (Wei et al., 2021). The second term characterizes the exponential convergence of the $T$ inner PU iterations. Moreover, the last two terms characterize the estimation errors $\mathcal{O}(N^{-1})$ and $\mathcal{O}(N^{-2/3}) + \mathcal{O}(\overline{N}^{-2/3})$ of the estimators (12) and (14,15), respectively. As a comparison, Wei et al. (2021) established a much larger estimation error $\mathcal{O}(N^{-1/3})$. These improvements, as we show in Theorem 2 later, lead to a substantially improved overall sample complexity over the state-of-the-art result. In particular, due to the exponentially weighted average structure in (20), the sample complexity can be further optimized by an adaptive scheduling of the batch sizes.

**Technical Novelty.** We further comment on the proof of Theorem 1. Note that the analysis of the PE algorithm in (Cen et al., 2021) requires full knowledge of the environment and does not characterize the convergence of duality gap. As a comparison, to establish the duality gap convergence rate (20) of the model-free SPE, we need to make substantial new developments to tightly bound the estimation errors of the proposed stochastic estimators. We elaborate our technical contributions below.

- As we explained in Remark 1, the sample averages involved in our estimators are correlated Markovian samples. To bound the additional bias induced by these correlated samples, we apply the concentration inequalities developed in (Paulin, 2015) for dependent samples to establish tight high-probability estimation error bounds.

- We develop a much refined analysis of the state value function estimation error $\|\widehat{V}_{k+1} - V'_{k+1}\|_\infty$, which is the key to develop tight bounds for all the other estimation errors. Specifically, we first propose the following error decomposition for any state $s$

$$|\widehat{V}_{k+1}(s)-V'_{k+1}(s)| = \left|\frac{\widehat{v}_{k+1}(s)}{\widehat{\mu}_k(s)} - \frac{v_{k+1}(s)}{\mu_k(s)}\right| \le \frac{|\widehat{v}_{k+1}(s)-v_{k+1}(s)|}{\mu_k(s)} + |\widehat{v}_{k+1}(s)|\left|\frac{\mu_k(s)-\widehat{\mu}_k(s)}{\mu_k(s)\widehat{\mu}_k(s)}\right|,$$

where $\widehat{v}_{k+1}(s), \widehat{\mu}_{k+1}(s)$ are sample average estimators of $v_{k+1}(s), \mu_{k+1}(s)$, respectively, and we refer to Appendix B for the definitions of these terms. The motivation is that the $|\widehat{v}_{k+1}(s)|$ in the second term helps cancel out the estimator $\widehat{\mu}_k(s)$ in the denominator, and then all the denominators do not involve any sample average estimators, which may take a small numerical value that causes divergence and a loose concentration bound. By leveraging this special decomposition and the recursive structure of the stochastic estimator (12), we are able to establish the following key contraction property of the estimation error (see (85) in Appendix B).

$$\|\widehat{V}_{k+1} - V'_{k+1}\|_\infty \le \gamma\|\widehat{V}_k - V'_k\|_\infty + \mathcal{O}(N_{k+1}^{-1/2}).$$

By telescoping the above contraction bound, we obtain tight estimation error bounds for all the proposed stochastic estimators. As a comparison, Wei et al. (2021) directly applied the Azuma-Hoeffding inequality with independent samples to bound the entire estimator and obtain a loose error bound, and Liu et al. (2021) simply assumed a small upper bound for the estimation error.

- We develop a stochastic predictive update (SPU) algorithm with general inexact value function estimations and a finite-time convergence analysis of its duality gap (see Lemma 1 for the SPU algorithm and its convergence proof). This generalizes the convergence result of the PU algorithm established in (Cen et al., 2021), which uses exact value functions based on full knowledge of the environment. Finally, by incorporating our developed tight estimation error bounds into the finite-time duality gap bound of SPU, we obtain the desired convergence rate in Theorem 1.

Based on Theorem 1, we obtain the following sample complexity of SPE for achieving an $\epsilon$-Nash equilibrium duality gap of the original Markov game, i.e., $D^{(0)}(\pi_{K-1}^{(1)}, \pi_{K-1}^{(2)}) \le \epsilon$. Here, we adopt an adaptive batch size scheduling scheme to optimize the complexity order. The overall sample complexity is given by $\sum_{k=0}^{K-1}\left[N_{k+1} + 2\sum_{t=0}^{T_k-1}(N_{k,t} + \overline{N}_{k,t+1})\right]$.

**Theorem 2** (Sample complexity). *Implement Algorithm 1 with* $\eta = \mathcal{O}\big(1 - \gamma\big)$, $\tau = \mathcal{O}\big(\frac{\epsilon(1-\gamma)}{\ln A_{\max}}\big)$, $K = \mathcal{O}\Big[\frac{1}{1-\gamma} \ln\Big(\frac{\ln A_{\max}}{\epsilon(1-\gamma)}\Big)\Big]$ *and* $T_k = 1 + \frac{k \ln \gamma^{-1}}{\ln(1-\eta\tau)^{-1}}$. *Choose the following adaptive batch sizes.*

$$N_{k+1} = \widetilde{\mathcal{O}}\Big(\frac{t_{mix}(\ln^2 A_{\max})\gamma^{-\frac{k}{2}}}{\epsilon^2 \mu_{\min}(1-\gamma)^8}\Big), \ N_{k,t} = \overline{N}_{k,t}\epsilon^{\frac{3}{2}}(1-\gamma)^3 = \widetilde{\mathcal{O}}\Big(\frac{t_{mix}A_{\max}(1-\eta\tau)^{\frac{-3(t+1)}{5}}}{\mu_{\min}(1-\gamma)^3}\Big).$$

*Then, for any* $\epsilon \le \frac{\ln A_{\max}}{1-\gamma}$, *the overall sample complexity to achieve* $D^{(0)}(\pi_{K-1}^{(1)}, \pi_{K-1}^{(2)}) \le \epsilon$ *is* $\widetilde{\mathcal{O}}\big(\frac{t_{mix}A_{\max}}{\mu_{\min}\epsilon^{5.5}(1-\gamma)^{13.5}}\big)$. *Please refer to* (118) *in Appendix E for a complete expression.*

The above complexity result is obtained by choosing a small $\tau = \mathcal{O}\big(\frac{\epsilon(1-\gamma)}{\ln A_{\max}}\big)$ for the convergence rate result in Theorem 1. Specifically, we show in Lemma 6 that the duality gap is Lipschitz continuous with regard to the entropy regularization parameter, i.e., $\big|D^{(\tau)}(\pi^{(1)}, \pi^{(2)}) - D^{(0)}(\pi^{(1)}, \pi^{(2)})\big| \le \frac{2\tau \ln A_{\max}}{1-\gamma}$. Therefore, by choosing a proper small $\tau$, convergence of the duality gap $D^{(\tau)}$ of the entropy-regularized Markov game implies the convergence of the duality gap $D^{(0)}$ of the original Markov game.

**Remark on Improvement of Sample Complexity.** We elaborate on the improvement of sample complexity in two different levels: population level (with known environment) and stochastic level (using stochastic samples). First, in the population level, the original policy extragradient (PE) algorithm in (Cen et al., 2021) already achieves a faster convergence rate than the OGDA-based algorithm proposed in (Wei et al., 2021). Specifically, to achieve an $\epsilon$-Nash equilibrium point of the original Markov game, PE requires $\mathcal{O}\big(\frac{A_{\max}}{(1-\gamma)\epsilon} \ln^2 \epsilon^{-1}\big)$ iterations by choosing a proper entropy regularization parameter $\tau = \mathcal{O}\big(\frac{\epsilon(1-\gamma)}{\ln A_{\max}}\big)$. As a comparison, the OGDA-based algorithm requires $\widetilde{\mathcal{O}}\Big(\frac{|\mathcal{S}|^3}{(1-\gamma)^9\epsilon^2}\Big)$ iterations by substituting $\epsilon = 0$ (no stochastic error) and their choice of step size into the Theorem 1 of Wei et al. (2021). This improvement is in the order of $\widetilde{\mathcal{O}}(\epsilon^{-1})$ and is due to the use of entropy regularization in the PE algorithm, which improves the geometry of the bilinear matrix game. Second, the rest of the improvement of sample complexity (about $\widetilde{\mathcal{O}}(\epsilon^{-1.5})$) comes from the stochastic level. Specifically, our stochastic PE (SPE) allows to use a large constant learning rate $\eta = \mathcal{O}\big(1 - \gamma\big)$, whereas the OGDA-based algorithm in (Wei et al., 2021) needs to use a substantially smaller learning rate $\eta = \mathcal{O}(\sqrt{(1-\gamma)^5|\mathcal{S}|^{-1}})$, which significantly slows down its convergence in both the population and stochastic levels. Moreover, both the PE and our SPE take $\mathcal{O}(\ln \epsilon^{-1})$ inner updates to achieve an accurate solution of the matrix game (7), whereas the OGDA-based algorithm uses only one inner update to solve the matrix game and hence suffers from a larger optimization error. Finally, as we explained in the previous technical novelty paragraph, we improve the techniques used in bounding the estimation errors. Specifically, Wei et al. (2021) bounds the estimation errors in all the iterations by a small constant $\epsilon$ using independent samples. As a comparison, we establish a key contraction property of the estimation error over the iterations with correlated Markovian samples. Such a property allows us to use a growing batch size that bounds the errors loosely in the initial iterations and achieves tight error bounds in the end.

## 5 CONCLUSION

In this paper, we developed a model-free, provably convergent, sample efficient, symmetric and private stochastic policy extra gradient algorithm for solving two-player zero-sum Markov games. Our algorithm leverages entropy regularization to facilitate the algorithm convergence and develops new stochastic estimators to accurately estimate the value functions. We proved that our SPE algorithm achieved a fast convergence rate in terms of the Nash equilibrium duality gap and moreover, achieves a substantially improved sample complexity over the state-of-the-art result. We believe our algorithm deepens the understanding of Markov games from a computation complexity perspective. In the future study, it is interesting to extend SPE algorithm to the multi-agent setting for solving general-sum Markov games and competitive games that involve multiple cooperative teams. Another interesting direction is to improve the algorithm to further reduce the sample complexity to approach the theoretical lower bound established in (Zhang et al., 2020).

ACKNOWLEDGMENTS

The work of Ziyi Chen, Shaocong Ma and Yi Zhou was supported in part by U.S. National Science Foundation under the Grants CCF-2106216 and DMS-2134223.

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

# Appendix

## Table of Contents

## A    ANALYSIS OF STOCHASTIC PU (SPU) FOR ENTROPY-REGULARIZED MATRIX GAME

We first consider the following zero-sum entropy regularized matrix game, which can be considered as a simple special case of Markov game with only one state.

$$\max_{\mu \in \Delta(d_1)} \min_{\nu \in \Delta(d_2)} h_\tau(\mu, \nu) := \mu^\top Q \nu + \tau \mathcal{H}(\mu) - \tau \mathcal{H}(\nu). \tag{22}$$

where $Q \in \mathbb{R}^{d_1 \times d_2}$ is fixed. The solution $(\mu_\tau^*, \nu_\tau^*)$ of the above problem, also known as the quantal response equilibrium (QRE), satisfies the following condition (Cen et al., 2021)

$$\mu_\tau^* \propto \exp(Q \nu_\tau^* / \tau), \nu_\tau^* \propto \exp(-Q^\top \mu_\tau^* / \tau). \tag{23}$$

Our stochastic PU (SPU) algorithm for the above matrix game is written as follows,

$$\overline{\mu}_{t+1}(a) \propto \mu_t(a)^{1-\eta\tau} \exp\left(\eta \left[Q\nu_t + \delta_t^{(1)}\right]_a\right) \tag{24}$$

$$\overline{\nu}_{t+1}(b) \propto \nu_t(b)^{1-\eta\tau} \exp\left(-\eta \left[Q^\top \mu_t + \delta_t^{(2)}\right]_b\right) \tag{25}$$

$$\mu_{t+1}(a) \propto \mu_t(a)^{1-\eta\tau} \exp\left(\eta \left[Q\overline{\nu}_{t+1} + \overline{\delta}_t^{(1)}\right]_a\right) \tag{26}$$

$$\nu_{t+1}(b) \propto \nu_t(b)^{1-\eta\tau} \exp\left(-\eta \left[Q^\top \overline{\mu}_{t+1} + \overline{\delta}_t^{(2)}\right]_b\right), \tag{27}$$

where $\delta_t^{(1)}$ is an additive noise to the underlying true quantity $Q\nu_t$, and the other noises $\delta_t^{(2)}, \overline{\delta}_t^{(1)}$, $\overline{\delta}_t^{(2)}$ are similar. If all these noises are zero, the above stochastic PU algorithm reduces to the PU algorithm for matrix game defined in Section 2 of (Cen et al., 2021).

The convergence rate of stochastic PU for matrix game is shown below. The proof logic follows from that of (Cen et al., 2021).

**Lemma 1.** *Choose learning rate $\eta \leq \left[2(\tau + \|Q\|_\infty)\right]^{-1}$ and use the uniform policy initialization $\mu_0 = \mathbf{1}/d_1, \nu_0 = \mathbf{1}/d_2$. Then, the SPU algorithm defined by eqs. (24)-(27) for entropy-regularized matrix game has the following convergence rates.*

$$KL(\mu_\tau^* \| \mu_t) + KL(\nu_\tau^* \| \nu_t)$$
$$\leq (1 - \eta\tau)^t \ln(d_1 d_2)$$

$$+ \sum_{j=0}^{t-1} (1-\eta\tau)^{t-1-j} \Big[ 2\eta^2 \big( \|\delta_j^{(1)}\|_\infty^2 + \|\delta_j^{(2)}\|_\infty^2 \big) + \frac{2\eta}{\tau} \big( \|\overline{\delta}_j^{(1)}\|_\infty^2 + \|\overline{\delta}_j^{(2)}\|_\infty^2 \big) \Big]. \quad (28)$$

$$\big| h_\tau(\mu_\tau^\star, \nu_\tau^\star) - h_\tau(\overline{\mu}_{t+1}, \overline{\nu}_{t+1}) \big|$$

$$\leq \frac{4}{\eta} (1-\eta\tau)^t \ln(d_1 d_2)$$

$$+ \sum_{j=0}^{t} (1-\eta\tau)^{t-1-j} \Big[ 8\eta \big( \|\delta_j^{(1)}\|_\infty^2 + \|\delta_j^{(2)}\|_\infty^2 \big) + \frac{33}{4\tau} \big( \|\overline{\delta}_j^{(1)}\|_\infty^2 + \|\overline{\delta}_j^{(2)}\|_\infty^2 \big) \Big]. \quad (29)$$

$$\max_{\mu \in \Delta(d_1)} h_\tau(\mu, \overline{\nu}_{t+1}) - \min_{\nu \in \Delta(d_2)} h_\tau(\overline{\mu}_{t+1}, \nu)$$

$$\leq \frac{2}{\eta} (1-\eta\tau)^t \ln(d_1 d_2)$$

$$+ \sum_{j=0}^{t} (1-\eta\tau)^{t-1-j} \Big[ 16\eta \big( \|\delta_j^{(1)}\|_\infty^2 + \|\delta_j^{(2)}\|_\infty^2 \big) + \frac{12}{\tau} \big( \|\overline{\delta}_j^{(1)}\|_\infty^2 + \|\overline{\delta}_j^{(2)}\|_\infty^2 \big) \Big]. \quad (30)$$

*Proof.* Eq. (26) implies that for some constant $c \in \mathbb{R}$,

$$\ln \mu_{t+1} = (1-\eta\tau) \ln \mu_t + \eta(Q\overline{\nu}_{t+1} + \overline{\delta}_t^{(1)}) + c\mathbf{1}$$

$$\Rightarrow \big\langle \ln \mu_{t+1} - (1-\eta\tau) \ln \mu_t - \eta(Q\overline{\nu}_{t+1} + \overline{\delta}_t^{(1)}), \overline{\mu}_{t+1} - \mu_\tau^* \big\rangle = 0. \quad (31)$$

where we used $\langle \overline{\mu}_{t+1} - \mu_\tau^*, \mathbf{1} \rangle = 0$. Similarly, eqs. (27)&(23) respectively imply that

$$\big\langle \ln \nu_{t+1} - (1-\eta\tau) \ln \nu_t + \eta(Q^\top \overline{\mu}_{t+1} + \overline{\delta}_t^{(2)}), \overline{\nu}_{t+1} - \nu_\tau^* \big\rangle = 0 \quad (32)$$

$$\big\langle \ln \mu_\tau^* - Q\nu_\tau^*/\tau, \overline{\mu}_{t+1} - \mu_\tau^* \big\rangle = 0 \quad (33)$$

$$\big\langle \ln \nu_\tau^* + Q^\top \mu_\tau^*/\tau, \overline{\nu}_{t+1} - \nu_\tau^* \big\rangle = 0. \quad (34)$$

Similarly, eqs. (24)&(26) imply that

$$\big\langle \ln(\overline{\mu}_{t+1}/\mu_{t+1}), \overline{\mu}_{t+1} - \mu_{t+1} \big\rangle$$

$$= \big\langle \eta[Q(\nu_t - \overline{\nu}_{t+1}) + \delta_t^{(1)} - \overline{\delta}_t^{(1)}], \overline{\mu}_{t+1} - \mu_{t+1} \big\rangle$$

$$\leq \eta\|Q\|_\infty \|\overline{\mu}_{t+1} - \mu_{t+1}\|_1 \|\nu_t - \overline{\nu}_{t+1}\|_1 + \eta\|\delta_t^{(1)} - \overline{\delta}_t^{(1)}\|_\infty \|\overline{\mu}_{t+1} - \mu_{t+1}\|_1$$

$$\leq \frac{\eta}{2}\|Q\|_\infty \big( \|\overline{\mu}_{t+1} - \mu_{t+1}\|_1^2 + \|\nu_t - \overline{\nu}_{t+1}\|_1^2 \big) + \frac{\eta^2}{2(1-\eta\|Q\|_\infty)} \|\delta_t^{(1)} - \overline{\delta}_t^{(1)}\|_\infty^2$$

$$+ \frac{1}{2}(1-\eta\|Q\|_\infty)\|\overline{\mu}_{t+1} - \mu_{t+1}\|_1^2$$

$$\overset{(i)}{\leq} KL(\mu_{t+1}\|\overline{\mu}_{t+1}) + \eta\|Q\|_\infty KL(\overline{\nu}_{t+1}\|\nu_t) + 2\eta^2 \big( \|\delta_t^{(1)}\|_\infty^2 + \|\overline{\delta}_t^{(1)}\|_\infty^2 \big), \quad (35)$$

where (i) uses the Pinsker's inequality and $\eta \leq \big[ 2(\tau + \|Q\|_\infty) \big]^{-1}$.

Similarly to the above derivation, we can infer from eqs. (25)&(27) that

$$\big\langle \ln(\overline{\nu}_{t+1}/\nu_{t+1}), \overline{\nu}_{t+1} - \nu_{t+1} \big\rangle \leq KL(\nu_{t+1}\|\overline{\nu}_{t+1}) + \eta\|Q\|_\infty KL(\overline{\mu}_{t+1}\|\mu_t)$$

$$+ 2\eta^2 \big( \|\delta_t^{(2)}\|_\infty^2 + \|\overline{\delta}_t^{(2)}\|_\infty^2 \big). \quad (36)$$

Computing eq. (31)+eq. (32)-$\eta\tau$[eq. (33)+eq. (34)] yields that

$$\eta\langle \overline{\delta}_t^{(1)}, \overline{\mu}_{t+1} - \mu_\tau^* \rangle - \eta\langle \overline{\delta}_t^{(2)}, \overline{\nu}_{t+1} - \nu_\tau^* \rangle$$

$$= \big\langle \ln \mu_{t+1} - (1-\eta\tau) \ln \mu_t - \eta\tau \ln \mu_\tau^* - \eta Q(\overline{\nu}_{t+1} - \nu_\tau^*), \overline{\mu}_{t+1} - \mu_\tau^* \big\rangle$$

$$+ \big\langle \ln \nu_{t+1} - (1-\eta\tau) \ln \nu_t - \eta\tau \ln \nu_\tau^* + \eta Q^\top (\overline{\mu}_{t+1} - \mu_\tau^*), \overline{\nu}_{t+1} - \nu_\tau^* \big\rangle$$

$$= \big\langle \ln(\mu_{t+1}/\mu_\tau^*) - (1-\eta\tau) \ln(\mu_t/\mu_\tau^*), \overline{\mu}_{t+1} - \mu_\tau^* \big\rangle$$

$$+ \big\langle \ln(\nu_{t+1}/\nu_\tau^*) - (1-\eta\tau)\ln(\nu_t/\nu_\tau^*), \overline{\nu}_{t+1} - \nu_\tau^* \big\rangle$$

$$=KL(\mu_\tau^*\|\mu_{t+1}) - (1-\eta\tau)KL(\mu_\tau^*\|\mu_t)$$
$$+ \big\langle \ln(\overline{\mu}_{t+1}/\mu_\tau^*) - \ln(\overline{\mu}_{t+1}/\mu_{t+1}) - (1-\eta\tau)\big[\ln(\overline{\mu}_{t+1}/\mu_\tau^*) - \ln(\overline{\mu}_{t+1}/\mu_t)\big], \overline{\mu}_{t+1} \big\rangle$$
$$+ KL(\nu_\tau^*\|\nu_{t+1}) - (1-\eta\tau)KL(\nu_\tau^*\|\nu_t)$$
$$+ \big\langle \ln(\overline{\nu}_{t+1}/\nu_\tau^*) - \ln(\overline{\nu}_{t+1}/\nu_{t+1}) - (1-\eta\tau)\big[\ln(\overline{\nu}_{t+1}/\nu_\tau^*) - \ln(\overline{\nu}_{t+1}/\nu_t)\big], \overline{\nu}_{t+1} \big\rangle$$

$$=KL(\mu_\tau^*\|\mu_{t+1}) - (1-\eta\tau)KL(\mu_\tau^*\|\mu_t) + \eta\tau KL(\overline{\mu}_{t+1}\|\mu_\tau^*)$$
$$+ (1-\eta\tau)KL(\overline{\mu}_{t+1}\|\mu_t) + KL(\mu_{t+1}\|\overline{\mu}_{t+1})$$
$$+ KL(\nu_\tau^*\|\nu_{t+1}) - (1-\eta\tau)KL(\nu_\tau^*\|\nu_t) + \eta\tau KL(\overline{\nu}_{t+1}\|\nu_\tau^*)$$
$$+ (1-\eta\tau)KL(\overline{\nu}_{t+1}\|\nu_t) + KL(\nu_{t+1}\|\overline{\nu}_{t+1})$$
$$- \big\langle \ln(\overline{\mu}_{t+1}/\mu_{t+1}), \overline{\mu}_{t+1} - \mu_{t+1} \big\rangle - \big\langle \ln(\overline{\nu}_{t+1}/\nu_{t+1}), \overline{\nu}_{t+1} - \nu_{t+1} \big\rangle$$

$$\overset{(i)}{\ge} KL(\mu_\tau^*\|\mu_{t+1}) - (1-\eta\tau)KL(\mu_\tau^*\|\mu_t) + \eta\tau KL(\overline{\mu}_{t+1}\|\mu_\tau^*) + (1-\eta\tau-\eta\|Q\|_\infty)KL(\overline{\mu}_{t+1}\|\mu_t)$$
$$+ KL(\nu_\tau^*\|\nu_{t+1}) - (1-\eta\tau)KL(\nu_\tau^*\|\nu_t) + \eta\tau KL(\overline{\nu}_{t+1}\|\nu_\tau^*) + (1-\eta\tau-\eta\|Q\|_\infty)KL(\overline{\nu}_{t+1}\|\nu_t)$$
$$- 2\eta^2\big(\|\delta_t^{(1)}\|_\infty^2 + \|\overline{\delta}_t^{(1)}\|_\infty^2 + \|\delta_t^{(2)}\|_\infty^2 + \|\overline{\delta}_t^{(2)}\|_\infty^2\big), \tag{37}$$

where (i) uses eqs. (35)&(36). The left side of the above inequality has the following upper bound.

$$\eta\langle\overline{\delta}_t^{(1)}, \overline{\mu}_{t+1} - \mu_\tau^*\rangle - \eta\langle\overline{\delta}_t^{(2)}, \overline{\nu}_{t+1} - \nu_\tau^*\rangle$$
$$\le \eta\|\overline{\delta}_t^{(1)}\|_\infty\|\overline{\mu}_{t+1} - \mu_\tau^*\|_1 + \eta\|\overline{\delta}_t^{(2)}\|_\infty\|\overline{\nu}_{t+1} - \nu_\tau^*\|_1$$
$$\le \frac{\eta}{2}\Big(\frac{2\|\overline{\delta}_t^{(1)}\|_\infty^2}{\tau} + \frac{\tau}{2}\|\overline{\mu}_{t+1} - \mu_\tau^*\|_1^2 + \frac{2\|\overline{\delta}_t^{(2)}\|_\infty^2}{\tau} + \frac{\tau}{2}\|\overline{\nu}_{t+1} - \nu_\tau^*\|_1^2\Big)$$
$$\le \frac{\eta}{\tau}\big(\|\overline{\delta}_t^{(1)}\|_\infty^2 + \|\overline{\delta}_t^{(2)}\|_\infty^2\big) + \frac{\eta\tau}{2}\big(KL(\overline{\mu}_{t+1}\|\mu_\tau^*) + KL(\overline{\nu}_{t+1}\|\nu_\tau^*)\big). \tag{38}$$

Substituting eq. (38) into eq. (37) and rearranging it yields that

$$KL(\mu_\tau^*\|\mu_{t+1}) + KL(\nu_\tau^*\|\nu_{t+1}) + \frac{\eta\tau}{2}\big[KL(\overline{\mu}_{t+1}\|\mu_\tau^*) + KL(\overline{\nu}_{t+1}\|\nu_\tau^*)\big]$$
$$+ \frac{1}{2}\big[KL(\overline{\mu}_{t+1}\|\mu_t) + KL(\overline{\nu}_{t+1}\|\nu_t)\big]$$
$$\le (1-\eta\tau)\big[KL(\mu_\tau^*\|\mu_t) + KL(\nu_\tau^*\|\nu_t)\big] + 2\eta^2\big(\|\delta_t^{(1)}\|_\infty^2 + \|\delta_t^{(2)}\|_\infty^2\big)$$
$$+ \frac{2\eta}{\tau}\big(\|\overline{\delta}_t^{(1)}\|_\infty^2 + \|\overline{\delta}_t^{(2)}\|_\infty^2\big), \tag{39}$$

where we use $\eta \le \big[2(\tau + \|Q\|_\infty)\big]^{-1} \le (2\tau)^{-1}$. This further implies that

$$KL(\mu_\tau^*\|\mu_{t+1}) + KL(\nu_\tau^*\|\nu_{t+1})$$
$$\le (1-\eta\tau)\big[KL(\mu_\tau^*\|\mu_t) + KL(\nu_\tau^*\|\nu_t)\big]$$
$$+ 2\eta^2\big(\|\delta_t^{(1)}\|_\infty^2 + \|\delta_t^{(2)}\|_\infty^2\big) + \frac{2\eta}{\tau}\big(\|\overline{\delta}_t^{(1)}\|_\infty^2 + \|\overline{\delta}_t^{(2)}\|_\infty^2\big). \tag{40}$$

Consequently, eq. (28) can be proved via iterating the above inequality and using the facts that $KL(\mu_\tau^*\|\mu_0) \le \ln d_1$, $KL(\nu_\tau^*\|\nu_0) \le \ln d_2$ (Since $\mu_0$ and $\nu_0$ are uniform probability vectors).

Next, we will prove eq. (29).

Computing eq. (31) $-\eta\tau\cdot$eq. (33) yields that

$$\langle Q(\overline{\nu}_{t+1} - \nu_\tau^*), \overline{\mu}_{t+1} - \mu_\tau^*\rangle$$
$$= \eta^{-1}\big\langle \ln\mu_{t+1} - (1-\eta\tau)\ln\mu_t - \eta\tau\ln\mu_\tau^* - \eta\overline{\delta}_t^{(1)}, \overline{\mu}_{t+1} - \mu_\tau^*\big\rangle. \tag{41}$$

Hence, we conclude that

$$h_\tau(\overline{\mu}_{t+1}, \overline{\nu}_{t+1}) - h_\tau(\mu_\tau^\star, \nu_\tau^\star)$$
$$\le h_\tau(\overline{\mu}_{t+1}, \overline{\nu}_{t+1}) - h_\tau(\overline{\mu}_{t+1}, \nu_\tau^\star)$$

$$= \overline{\mu}_{t+1}^\top Q(\overline{\nu}_{t+1} - \nu_\tau^\star) + \tau \overline{\nu}_{t+1}^\top \ln \overline{\nu}_{t+1} - \tau \nu_\tau^{\star\top} \ln \nu_\tau^\star$$

$$= (\overline{\mu}_{t+1} - \mu_\tau^\star)^\top Q(\overline{\nu}_{t+1} - \nu_\tau^\star) + \mu_\tau^{\star\top} Q(\overline{\nu}_{t+1} - \nu_\tau^\star) + \tau \overline{\nu}_{t+1}^\top \ln \overline{\nu}_{t+1} - \tau \nu_\tau^{\star\top} \ln \nu_\tau^\star$$

$$\overset{(i)}{=} \eta^{-1} \langle \ln \mu_{t+1} - (1-\eta\tau) \ln \mu_t - \eta\tau \ln \mu_\tau^* - \eta \overline{\delta}_t^{(1)}, \overline{\mu}_{t+1} - \mu_\tau^* \rangle - \tau \langle \ln \nu_\tau^*, \overline{\nu}_{t+1} - \nu_\tau^* \rangle$$
$$+ \tau \overline{\nu}_{t+1}^\top \ln \overline{\nu}_{t+1} - \tau \nu_\tau^{\star\top} \ln \nu_\tau^\star$$

$$\overset{(ii)}{=} \eta^{-1} \big[ KL(\mu_\tau^*\|\mu_{t+1}) - (1-\eta\tau) KL(\mu_\tau^*\|\mu_t) + \eta\tau KL(\overline{\mu}_{t+1}\|\mu_\tau^*) + (1-\eta\tau) KL(\overline{\mu}_{t+1}\|\mu_t)$$
$$+ KL(\mu_{t+1}\|\overline{\mu}_{t+1}) - \langle \ln(\overline{\mu}_{t+1}/\mu_{t+1}), \overline{\mu}_{t+1} - \mu_{t+1} \rangle \big] + \tau KL(\overline{\nu}_{t+1}\|\nu_\tau^\star) - \langle \overline{\delta}_t^{(1)}, \overline{\mu}_{t+1} - \mu_\tau^* \rangle$$

$$\leq \eta^{-1} \big[ KL(\mu_\tau^*\|\mu_{t+1}) - (1-\eta\tau) KL(\mu_\tau^*\|\mu_t) + \eta\tau \big( KL(\overline{\mu}_{t+1}\|\mu_\tau^*) + KL(\overline{\nu}_{t+1}\|\nu_\tau^\star) \big)$$
$$+ (1-\eta\tau) KL(\overline{\mu}_{t+1}\|\mu_t) - KL(\overline{\mu}_{t+1}\|\mu_{t+1}) \big] + \frac{1}{2\tau} \|\overline{\delta}_t^{(1)}\|_\infty^2 + \frac{\tau}{2} \|\overline{\mu}_{t+1} - \mu_\tau^*\|_1^2$$

$$\overset{(iii)}{\leq} \eta^{-1} \big[ KL(\mu_\tau^*\|\mu_{t+1}) + 2\eta\tau \big( KL(\overline{\mu}_{t+1}\|\mu_\tau^*) + KL(\overline{\nu}_{t+1}\|\nu_\tau^\star) \big) + KL(\overline{\mu}_{t+1}\|\mu_t) \big] + \frac{1}{4\tau} \|\overline{\delta}_t^{(1)}\|_\infty^2$$

$$\overset{(iv)}{\leq} 4\eta^{-1} \big[ KL(\mu_\tau^*\|\mu_t) + KL(\nu_\tau^*\|\nu_t) + 2\eta^2 \big( \|\delta_t^{(1)}\|_\infty^2 + \|\delta_t^{(2)}\|_\infty^2 \big) + \frac{2\eta}{\tau} \big( \|\overline{\delta}_t^{(1)}\|_\infty^2 + \|\overline{\delta}_t^{(2)}\|_\infty^2 \big) \big]$$
$$+ \frac{1}{4\tau} \|\overline{\delta}_t^{(1)}\|_\infty^2$$

$$\overset{(v)}{\leq} 4\eta^{-1} \big[ KL(\mu_\tau^*\|\mu_t) + KL(\nu_\tau^*\|\nu_t) \big]$$
$$+ 8\eta \big( \|\delta_t^{(1)}\|_\infty^2 + \|\delta_t^{(2)}\|_\infty^2 \big) + \frac{33}{4\tau} \big( \|\overline{\delta}_t^{(1)}\|_\infty^2 + \|\overline{\delta}_t^{(2)}\|_\infty^2 \big), \tag{42}$$

where (i) uses eq. (41)&(34), (ii) follows the derivation of eq. (37), (iii) uses the Pinsker's inequality, (iv) uses eq. (39), and (v) uses $\eta \leq \big[ 2(\tau + \|Q\|_\infty) \big]^{-1}$. In a similar way, we can also prove that

$$h_\tau(\mu_\tau^\star, \nu_\tau^\star) - h_\tau(\overline{\mu}_{t+1}, \overline{\nu}_{t+1}) \leq h_\tau(\mu_\tau^\star, \overline{\nu}_{t+1}) - h_\tau(\overline{\mu}_{t+1}, \overline{\nu}_{t+1})$$
$$\leq 4\eta^{-1} \big[ KL(\mu_\tau^*\|\mu_t) + KL(\nu_\tau^*\|\nu_t) \big]$$
$$+ 8\eta \big( \|\delta_t^{(1)}\|_\infty^2 + \|\delta_t^{(2)}\|_\infty^2 \big) + \frac{33}{4\tau} \big( \|\overline{\delta}_t^{(1)}\|_\infty^2 + \|\overline{\delta}_t^{(2)}\|_\infty^2 \big). \tag{43}$$

Combining eqs. (42)&(43) yields that

$$\big| h_\tau(\mu_\tau^\star, \nu_\tau^\star) - h_\tau(\overline{\mu}_{t+1}, \overline{\nu}_{t+1}) \big|$$
$$\leq 4\eta^{-1} \big[ KL(\mu_\tau^*\|\mu_t) + KL(\nu_\tau^*\|\nu_t) \big] + 8\eta \big( \|\delta_t^{(1)}\|_\infty^2 + \|\delta_t^{(2)}\|_\infty^2 \big) + \frac{33}{4\tau} \big( \|\overline{\delta}_t^{(1)}\|_\infty^2 + \|\overline{\delta}_t^{(2)}\|_\infty^2 \big)$$
$$\overset{(i)}{\leq} 4\eta^{-1} (1-\eta\tau)^t \ln(d_1 d_2)$$
$$+ \sum_{j=0}^{t-1} (1-\eta\tau)^{t-1-j} \big[ 8\eta \big( \|\delta_j^{(1)}\|_\infty^2 + \|\delta_j^{(2)}\|_\infty^2 \big) + \frac{8}{\tau} \big( \|\overline{\delta}_j^{(1)}\|_\infty^2 + \|\overline{\delta}_j^{(2)}\|_\infty^2 \big) \big]$$
$$+ 8\eta \big( \|\delta_t^{(1)}\|_\infty^2 + \|\delta_t^{(2)}\|_\infty^2 \big) + \frac{33}{4\tau} \big( \|\overline{\delta}_t^{(1)}\|_\infty^2 + \|\overline{\delta}_t^{(2)}\|_\infty^2 \big)$$
$$\leq 4\eta^{-1} (1-\eta\tau)^t \ln(d_1 d_2)$$
$$+ \sum_{j=0}^{t} (1-\eta\tau)^{t-1-j} \big[ 8\eta \big( \|\delta_j^{(1)}\|_\infty^2 + \|\delta_j^{(2)}\|_\infty^2 \big) + \frac{33}{4\tau} \big( \|\overline{\delta}_j^{(1)}\|_\infty^2 + \|\overline{\delta}_j^{(2)}\|_\infty^2 \big) \big],$$

where (i) uses eq. (28). This proves eq. (29).

Next, to prove the duality gap (30), we first derive an upper bound of $KL(\mu_\tau^\star\|\overline{\mu}_{t+1}) + KL(\nu_\tau^\star\|\overline{\nu}_{t+1})$.

$$KL(\nu_\tau^\star\|\overline{\nu}_{t+1})$$
$$= KL(\nu_\tau^\star\|\nu_{t+1}) - KL(\overline{\nu}_{t+1}\|\nu_{t+1}) + \langle \overline{\nu}_{t+1} - \nu_\tau^\star, \ln \overline{\nu}_{t+1} - \ln \nu_{t+1} \rangle$$
$$\leq KL(\nu_\tau^\star\|\nu_{t+1}) + \langle \overline{\nu}_{t+1} - \nu_\tau^\star, \eta Q^\top (\overline{\mu}_{t+1} - \mu_t) + \eta(\overline{\delta}_t^{(2)} - \delta_t^{(2)}) \rangle$$
$$\leq KL(\nu_\tau^\star\|\nu_{t+1}) + \|\overline{\nu}_{t+1} - \nu_\tau^\star\|_1 \big( \eta\|Q\|_\infty \|\overline{\mu}_{t+1} - \mu_t\|_1 + \eta\|\overline{\delta}_t^{(2)} - \delta_t^{(2)}\|_\infty \big)$$

$$\leq KL(\nu_\tau^\star \| \nu_{t+1})$$
$$+ \frac{1}{2} \Big[ \Big( \eta \|Q\|_\infty + \frac{1}{4} \Big) \|\overline{\nu}_{t+1} - \nu_\tau^\star\|_1^2 + \eta \|Q\|_\infty \|\overline{\mu}_{t+1} - \mu_t\|_1^2 + 4\eta^2 \|\overline{\delta}_t^{(2)} - \delta_t^{(2)}\|_\infty^2 \Big]$$
$$\overset{(i)}{\leq} KL(\nu_\tau^\star \| \nu_{t+1}) + \frac{3}{4} KL(\nu_\tau^\star \| \overline{\nu}_{t+1}) + \frac{1}{2} KL(\overline{\mu}_{t+1} \| \mu_t) + 4\eta^2 \big( \|\delta_t^{(2)}\|_\infty^2 + \|\overline{\delta}_t^{(2)}\|_\infty^2 \big), \quad (44)$$

where (i) uses the Pinker's inequality and $\eta \leq \big[ 2(\tau + \|Q\|_\infty) \big]^{-1}$. Rearranging the above inequality yields that

$$KL(\nu_\tau^\star \| \overline{\nu}_{t+1}) \leq 4 KL(\nu_\tau^\star \| \nu_{t+1}) + 2 KL(\overline{\mu}_{t+1} \| \mu_t) + 16\eta^2 \big( \|\delta_t^{(2)}\|_\infty^2 + \|\overline{\delta}_t^{(2)}\|_\infty^2 \big). \quad (45)$$

Similarly, we can prove that

$$KL(\mu_\tau^\star \| \overline{\mu}_{t+1}) \leq 4 KL(\mu_\tau^\star \| \mu_{t+1}) + 2 KL(\overline{\nu}_{t+1} \| \nu_t) + 16\eta^2 \big( \|\delta_t^{(1)}\|_\infty^2 + \|\overline{\delta}_t^{(1)}\|_\infty^2 \big). \quad (46)$$

Summing up eqs. (45)&(46) yields that

$$KL(\mu_\tau^\star \| \overline{\mu}_{t+1}) + KL(\nu_\tau^\star \| \overline{\nu}_{t+1})$$
$$\leq 4 \big[ KL(\mu_\tau^\star \| \mu_{t+1}) + KL(\nu_\tau^\star \| \nu_{t+1}) \big] + 2 \big[ KL(\overline{\mu}_{t+1} \| \mu_t) + KL(\overline{\nu}_{t+1} \| \nu_t) \big]$$
$$+ 16\eta^2 \big( \|\delta_t^{(1)}\|_\infty^2 + \|\overline{\delta}_t^{(1)}\|_\infty^2 + \|\delta_t^{(2)}\|_\infty^2 + \|\overline{\delta}_t^{(2)}\|_\infty^2 \big)$$
$$\overset{(i)}{\leq} 4 \big[ KL(\mu_\tau^* \| \mu_t) + KL(\nu_\tau^* \| \nu_t) \big] + 24\eta^2 \big( \|\delta_t^{(1)}\|_\infty^2 + \|\delta_t^{(2)}\|_\infty^2 \big)$$
$$+ \frac{16\eta}{\tau} \big( \|\overline{\delta}_t^{(1)}\|_\infty^2 + \|\overline{\delta}_t^{(2)}\|_\infty^2 \big), \quad (47)$$

where (i) uses eq. (39) and $\eta \leq \big[ 2(\tau + \|Q\|_\infty) \big]^{-1} \leq (2\tau)^{-1}$.

Finally, we prove the duality gap bound (30).

$$h_\tau(\mu_\tau^\star, \nu_\tau^\star) - h_\tau(\mu, \nu_\tau^\star) = (\mu_\tau^\star - \mu)^\top Q \nu_\tau^\star + \tau \mu^\top \ln \mu - \tau \mu_\tau^{\star\top} \ln \mu_\tau^\star$$
$$\overset{(i)}{=} \tau(\langle \mu_\tau^\star - \mu, \ln \mu_\tau^\star \rangle + \mu^\top \ln \mu - \mu_\tau^{\star\top} \ln \mu_\tau^\star) = \tau KL(\mu \| \mu_\tau^\star), \quad (48)$$

where (i) uses eq. (33).

$$h_\tau(\mu, \overline{\nu}_{t+1}) - h_\tau(\mu, \nu_\tau^\star)$$
$$= \mu^\top Q(\overline{\nu}_{t+1} - \nu_\tau^\star) + \tau \overline{\nu}_{t+1}^\top \ln \overline{\nu}_{t+1} - \tau \nu_\tau^{\star\top} \ln \nu_\tau^\star$$
$$= (\mu - \mu_\tau^\star)^\top Q(\overline{\nu}_{t+1} - \nu_\tau^\star) + \mu_\tau^{\star\top} Q(\overline{\nu}_{t+1} - \nu_\tau^\star) + \tau \overline{\nu}_{t+1}^\top \ln \overline{\nu}_{t+1} - \tau \nu_\tau^{\star\top} \ln \nu_\tau^\star$$
$$\overset{(i)}{\leq} \|\mu - \mu_\tau^\star\|_1 \|Q\|_\infty \|\overline{\nu}_{t+1} - \nu_\tau^\star\|_1 - \tau \big\langle \ln \nu_\tau^*, \overline{\nu}_{t+1} - \nu_\tau^\star \big\rangle + \tau \overline{\nu}_{t+1}^\top \ln \overline{\nu}_{t+1} - \tau \nu_\tau^{\star\top} \ln \nu_\tau^\star$$
$$\leq \frac{1}{2} \big[ \tau \|\mu - \mu_\tau^\star\|_1^2 + \|Q\|_\infty^2 \|\overline{\nu}_{t+1} - \nu_\tau^\star\|_1^2 / \tau \big] + \tau KL(\overline{\nu}_{t+1} \| \nu_\tau^\star)$$
$$\overset{(ii)}{\leq} \tau KL(\mu \| \mu_\tau^\star) + \tau KL(\overline{\nu}_{t+1} \| \nu_\tau^\star) + \tau^{-1} \|Q\|_\infty^2 KL(\nu_\tau^\star \| \overline{\nu}_{t+1}), \quad (49)$$

where (i) uses eq. (34) and (ii) uses the Pinsker's inequality. Then, eq. (49) & eq. (48) imply that

$$h_\tau(\mu, \overline{\nu}_{t+1}) - h_\tau(\mu_\tau^\star, \nu_\tau^\star) \leq \tau KL(\overline{\nu}_{t+1} \| \nu_\tau^\star) + \tau^{-1} \|Q\|_\infty^2 KL(\nu_\tau^\star \| \overline{\nu}_{t+1}), \quad (50)$$

where (i) uses eq. (45), (ii) uses the Pinker's inequality and $\eta \leq \big[ 2(\tau + \|Q\|_\infty) \big]^{-1}$.

Similarly, it can be proved that

$$h_\tau(\mu_\tau^\star, \nu_\tau^\star) - h_\tau(\overline{\mu}_{t+1}, \nu) \leq \tau KL(\overline{\mu}_{t+1} \| \mu_\tau^\star) + \tau^{-1} \|Q\|_\infty^2 KL(\mu_\tau^\star \| \overline{\mu}_{t+1}). \quad (51)$$

Therefore, the duality gap (30) can be proved as follows.

$$\max_{\mu \in \Delta(d_1)} h_\tau(\mu, \overline{\nu}_{t+1}) - \min_{\nu \in \Delta(d_2)} h_\tau(\overline{\mu}_{t+1}, \nu)$$
$$= \max_{\mu \in \Delta(d_1), \nu \in \Delta(d_2)} \big[ h_\tau(\mu, \overline{\nu}_{t+1}) - h_\tau(\overline{\mu}_{t+1}, \nu) \big]$$

$$\overset{(i)}{\leq} \tau[KL(\overline{\mu}_{t+1}\|\mu_\tau^\star) + KL(\overline{\nu}_{t+1}\|\nu_\tau^\star)] + \tau^{-1}\|Q\|_\infty^2[KL(\mu_\tau^\star\|\overline{\mu}_{t+1}) + KL(\nu_\tau^\star\|\overline{\nu}_{t+1})]$$

$$\overset{(ii)}{\leq} \tau[KL(\overline{\mu}_{t+1}\|\mu_\tau^\star) + KL(\overline{\nu}_{t+1}\|\nu_\tau^\star)] + \tau^{-1}\|Q\|_\infty^2$$
$$\left[4\big[KL(\mu_\tau^*\|\mu_t) + KL(\nu_\tau^*\|\nu_t)\big] + 24\eta^2\big(\|\delta_t^{(1)}\|_\infty^2 + \|\delta_t^{(2)}\|_\infty^2\big) + \frac{16\eta}{\tau}\big(\|\overline{\delta}_t^{(1)}\|_\infty^2 + \|\overline{\delta}_t^{(2)}\|_\infty^2\big)\right]$$

$$\overset{(iii)}{\leq} \max\left(\frac{2}{\eta}, \frac{4\|Q\|_\infty^2}{\tau}\right)$$
$$\left[KL(\mu_\tau^*\|\mu_t) + KL(\nu_\tau^*\|\nu_t) + 2\eta^2\big(\|\delta_t^{(1)}\|_\infty^2 + \|\delta_t^{(2)}\|_\infty^2\big) + \frac{2\eta}{\tau}\big(\|\overline{\delta}_t^{(1)}\|_\infty^2 + \|\overline{\delta}_t^{(2)}\|_\infty^2\big)\right]$$
$$+ \tau^{-1}\|Q\|_\infty^2\left[24\eta^2\big(\|\delta_t^{(1)}\|_\infty^2 + \|\delta_t^{(2)}\|_\infty^2\big) + \frac{16\eta}{\tau}\big(\|\overline{\delta}_t^{(1)}\|_\infty^2 + \|\overline{\delta}_t^{(2)}\|_\infty^2\big)\right]$$

$$\overset{(iv)}{\leq} \frac{2}{\eta}\Big((1-\eta\tau)^t \ln(d_1 d_2)$$
$$+ \sum_{j=0}^{t-1}(1-\eta\tau)^{t-1-j}\Big[2\eta^2\big(\|\delta_j^{(1)}\|_\infty^2 + \|\delta_j^{(2)}\|_\infty^2\big) + \frac{2\eta}{\tau}\big(\|\overline{\delta}_j^{(1)}\|_\infty^2 + \|\overline{\delta}_j^{(2)}\|_\infty^2\big)\Big]\Big)$$
$$+ 16\eta\big(\|\delta_t^{(1)}\|_\infty^2 + \|\delta_t^{(2)}\|_\infty^2\big) + \frac{12}{\tau}\big(\|\overline{\delta}_t^{(1)}\|_\infty^2 + \|\overline{\delta}_t^{(2)}\|_\infty^2\big)$$

$$\leq \frac{2}{\eta}(1-\eta\tau)^t \ln(d_1 d_2)$$
$$+ \sum_{j=0}^{t}(1-\eta\tau)^{t-1-j}\Big[16\eta\big(\|\delta_j^{(1)}\|_\infty^2 + \|\delta_j^{(2)}\|_\infty^2\big) + \frac{12}{\tau}\big(\|\overline{\delta}_j^{(1)}\|_\infty^2 + \|\overline{\delta}_j^{(2)}\|_\infty^2\big)\Big], \tag{52}$$

where (i) adds up eqs. (50) & (51), (ii) uses eq. (47), (iii) uses eq. (39), and (iv) uses eq. (28) and $4\tau^{-1}\|Q\|_\infty^2 \leq 2/\eta$ (since $\eta \leq \big[2(\tau + \|Q\|_\infty)\big]^{-1}$, $\tau \leq 1$). This proves eq. (30) $\qquad\square$

# B  ESTIMATION ERROR BOUNDS

In this section, we derive error bounds for the estimators $\widehat{V}_{k+1} \approx V'_{k+1}, \widehat{Q}_{k,t}^{(m)} \approx Q_{k,t}^{(m)}, \widehat{\overline{Q}}_{k,t}^{(m)} \approx \overline{Q}_{k,t}^{(m)}$ used in Algorithm 1. For convenience, we define the following additional notations.

$$\widehat{\mu}_k(s) := \frac{1}{N_{k+1}}\sum_{i\in\mathcal{N}_{k+1}}\mathbb{1}\{s_i = s\} \approx \mu_k(s) \tag{53}$$

$$v_{k+1}(s) := \mathbb{E}_{\mu_k,\pi_k^{(1)},\pi_k^{(2)}}\left[\big[R(\widetilde{s},\widetilde{a}^{(1)},\widetilde{a}^{(2)}) + \gamma V'_k(s')\big]\mathbb{1}\{\widetilde{s} = s\}\right] \tag{54}$$

$$v'_{k+1}(s) := \mathbb{E}_{\mu_k,\pi_k^{(1)},\pi_k^{(2)}}\left[\big[R(\widetilde{s},\widetilde{a}^{(1)},\widetilde{a}^{(2)}) + \gamma\widehat{V}_k(s')\big]\mathbb{1}\{\widetilde{s} = s\}\right] \tag{55}$$

$$\widehat{v}_{k+1}(s) := \frac{1}{N_{k+1}}\sum_{i\in\mathcal{N}_{k+1}}\left[\big[R_i + \gamma\widehat{V}_k(s_{i+1})\big]\mathbb{1}\{s_i = s\}\right] \approx v'_{k+1}(s) \approx v_{k+1}(s) \tag{56}$$

$$Q_k(s,a^{(1)},a^{(2)}) := R(s,a^{(1)},a^{(2)}) + \gamma\mathbb{E}_{s'\sim\mathcal{P}(\cdot|s,a^{(1)},a^{(2)})}\big[V'_k(s')\big] \tag{57}$$

$$V'_{k+1}(s) = f_\tau\big(Q_k(s);\pi_k^{(1)}(s),\pi_k^{(2)}(s)\big) = \frac{v_{k+1}(s)}{\mu_k(s)} + \tau\mathcal{H}\big(\pi_k^{(1)}(s)\big) - \tau\mathcal{H}\big(\pi_k^{(2)}(s)\big) \tag{58}$$

$$\widehat{V}_{k+1}(s) := \frac{\widehat{v}_{k+1}(s)}{\widehat{\mu}_k(s)} + \tau\mathcal{H}\big(\pi_k^{(1)}(s)\big) - \tau\mathcal{H}\big(\pi_k^{(2)}(s)\big) \approx V'_{k+1}(s) \tag{59}$$

$$\widehat{\mu}_{k,t}(s) := \frac{1}{N_{k,t}}\sum_{i\in\mathcal{N}_{k,t}}\mathbb{1}\{s_i = s\} \approx \mu'_{k,t}(s) \approx \mu_{k,t}(s), \tag{60}$$

$$q_{k,t}^{(m)}(s,a^{(m)}) := \mathbb{E}_{\mu'_{k,t},\pi'^{(1)}_{k,t},\pi'^{(2)}_{k,t}}\left[\big[R(\widetilde{s},\widetilde{a}^{(1)},\widetilde{a}^{(2)}) + \gamma V'_k(s')\big]\mathbb{1}\{\widetilde{s} = s, \widetilde{a}^{(m)} = a^{(m)}\}\right] \tag{61}$$

$$q'^{(m)}_{k,t}(s,a^{(m)}) := \mathbb{E}_{\mu'_{k,t},\pi'^{(1)}_{k,t},\pi'^{(2)}_{k,t}}\left[\big[R(\widetilde{s},\widetilde{a}^{(1)},\widetilde{a}^{(2)}) + \gamma\widehat{V}_k(s')\big]\mathbb{1}\{\widetilde{s} = s, \widetilde{a}^{(m)} = a^{(m)}\}\right] \tag{62}$$

$$\widehat{q}_{k,t}^{(m)}(s, a^{(m)}) := \frac{1}{N_{k,t}} \sum_{i \in \mathcal{N}_{k,t}} \left[ \left[ R_i + \gamma \widehat{V}_k(s_{i+1}) \right] \mathbb{1}\{ s_i = s, a_i^{(m)} = a^{(m)} \} \right]$$

$$\approx q_{k,t}^{\prime(m)}(s, a^{(m)}) \approx q_{k,t}^{(m)}(s, a^{(m)}) \tag{63}$$

$$Q_{k,t}^{\prime(m)}(s, a^{(m)}) := \mathbb{E}_{a^{(\backslash m)} \sim \pi_{k,t}^{\prime(\backslash m)}(s)} \left[ Q_k(s, a^{(1)}, a^{(2)}) \right] = \frac{q_{k,t}^{(m)}(s, a^{(m)})}{\mu_{k,t}'(s)\pi_{k,t}^{\prime(m)}(a^{(m)}|s)} \tag{64}$$

$$\widehat{Q}_{k,t}^{(m)}(s, a^{(m)}) := \frac{\widehat{q}_{k,t}^{(m)}(s, a^{(m)})}{\widehat{\mu}_{k,t}(s)\pi_{k,t}^{\prime(m)}(a^{(m)}|s)} \approx Q_{k,t}^{\prime(m)}(s, a^{(m)}) \approx Q_{k,t}^{(m)}(s, a^{(m)}), \tag{65}$$

where eqs. (53)-(59) and (60)-(65) are introduced for the description and error bound proof of the estimations $\widehat{V}_{k+1} \approx V_{k+1}'$ and $\widehat{Q}_{k,t}^{(m)} \approx Q_{k,t}^{(m)}$, respectively. Specifically, eq. (53) estimates the stationary state distribution $\mu_k$ associated with the policy pair $\pi_k^{(1)}, \pi_k^{(2)}$ using state frequency $\widehat{\mu}_k(s)$. In eqs. (54)&(55), the expectation is taken over $\widetilde{s} \sim \mu_k, \widetilde{a}^{(1)} \sim \pi_k^{(1)}(s), \widetilde{a}^{(2)} \sim \pi_k^{(2)}(s), s' \sim \mathcal{P}(\cdot|\widetilde{s}, \widetilde{a}^{(1)}, \widetilde{a}^{(2)})$. The definition of $\mathbb{E}_{\mu_{k,t}', \pi_{k,t}^{\prime(1)}, \pi_{k,t}^{\prime(2)}}$ in eqs. (61)&(62) is similar. Equations (58) & (64) give two equivalent definitions of $V_{k+1}'$ and $Q_{k,t}^{\prime(m)}(s, a^{(m)})$, respectively [1] (We will prove the equivalence in Lemma 2 below). Equation (60) estimates the stationary state distribution $\mu_{k,t}'$ associated with the smoothed policy pair $\pi_{k,t}^{\prime(1)}, \pi_{k,t}^{\prime(2)}$, and thus approximates the stationary state distribution $\mu_{k,t}$ associated with the current policy pair $\pi_{k,t}^{(1)}, \pi_{k,t}^{(2)}$.

To prove the estimation error bounds, we first prove the following two lemmas.

**Lemma 2.** *Eq. (13) and the second "=" of eqs. (58) & (64) hold for all $s$, $a^{(1)}, a^{(2)}$.*

*Proof.* The second "=" of eq. (64) under $m = 1$ can be proved as follows.

$$q_{k,t}^{(1)}(s, a^{(1)})$$

$$\overset{(i)}{=} \mathbb{E}_{\mu_{k,t}', \pi_{k,t}^{\prime(1)}, \pi_{k,t}^{\prime(2)}} \left[ \left[ R(\widetilde{s}, \widetilde{a}^{(1)}, \widetilde{a}^{(2)}) + \gamma V_k'(s') \right] \mathbb{1}\{ \widetilde{s} = s, \widetilde{a}^{(m)} = a^{(m)} \} \right]$$

$$= \mathbb{E}_{\widetilde{s} \sim \mu_{k,t}', \widetilde{a}^{(1)} \sim \pi_{k,t}^{\prime(1)}(s), \widetilde{a}^{(2)} \sim \pi_{k,t}^{\prime(2)}(s), s' \sim \mathcal{P}(\cdot|s, a^{(1)}, \widetilde{a}^{(2)})}$$

$$\left[ \left[ R(s, a^{(1)}, \widetilde{a}^{(2)}) + \gamma V_k'(s') \right] \mathbb{1}\{ \widetilde{s} = s, \widetilde{a}^{(1)} = a^{(1)} \} \right]$$

$$\overset{(ii)}{=} \mathbb{E}_{\widetilde{s} \sim \mu_{k,t}', \widetilde{a}^{(1)} \sim \pi_{k,t}^{\prime(1)}(s)} \left[ \mathbb{1}\{ \widetilde{s} = s, \widetilde{a}^{(1)} = a^{(1)} \} \right]$$

$$\mathbb{E}_{\widetilde{a}^{(2)} \sim \pi_{k,t}^{\prime(2)}(s), s' \sim \mathcal{P}(\cdot|s, a^{(1)}, \widetilde{a}^{(2)})} \left[ R(s, a^{(1)}, \widetilde{a}^{(2)}) + \gamma V_k'(s') \right]$$

$$\overset{(iii)}{=} \mu_{k,t}'(s)\pi_{k,t}^{\prime(1)}(a^{(1)}|s) \mathbb{E}_{\widetilde{a}^{(2)} \sim \pi_{k,t}^{\prime(2)}(s)} \left[ Q_k(s, a^{(1)}, \widetilde{a}^{(2)}) \right]$$

$$\overset{(iv)}{=} \mu_{k,t}'(s)\pi_{k,t}^{\prime(1)}(a^{(1)}|s) Q_{k,t}'(s, a^{(1)}),$$

where (i) uses eq. (61) which denotes $\mathbb{E}_{\mu_{k,t}', \pi_{k,t}^{\prime(1)}, \pi_{k,t}^{\prime(2)}}$ as expectation over $\widetilde{s} \sim \mu_{k,t}', \widetilde{a}^{(1)} \sim \pi_{k,t}^{\prime(1)}(\widetilde{s})$, $\widetilde{a}^{(2)} \sim \pi_{k,t}^{\prime(2)}(\widetilde{s}), s' \sim \mathcal{P}(\cdot|\widetilde{s}, \widetilde{a}^{(1)}, \widetilde{a}^{(2)})$, (ii) uses independence between $\widetilde{s} \sim \mu_{k,t}', \widetilde{a}^{(1)} \sim \pi_{k,t}^{\prime(1)}(s)$ and $\widetilde{a}^{(2)} \sim \pi_{k,t}^{\prime(2)}(s), s' \sim \mathcal{P}(\cdot|s, a^{(1)}, \widetilde{a}^{(2)})$, (iii) uses eq. (57), and (iv) uses the first definition of eq. (64). The second "=" of eq. (64) under $m = 2$ and eq. (13) can be proved in a similar way.

The second "=" of eq. (58) can be proved as follows.

$$v_{k+1}(s)$$

$$\overset{(i)}{=} \mathbb{E}_{\mu_k, \pi_k^{(1)}, \pi_k^{(2)}} \left[ \left[ R(\widetilde{s}, \widetilde{a}^{(1)}, \widetilde{a}^{(2)}) + \gamma V_k'(s') \right] \mathbb{1}\{ \widetilde{s} = s \} \right]$$

$$= \mathbb{E}_{\widetilde{s} \sim \mu_k, \widetilde{a}^{(1)} \sim \pi_k^{(1)}(s), \widetilde{a}^{(2)} \sim \pi_k^{(2)}(s), s' \sim \mathcal{P}(\cdot|s, \widetilde{a}^{(1)}, \widetilde{a}^{(2)})} \left[ \left[ R(s, \widetilde{a}^{(1)}, \widetilde{a}^{(2)}) + \gamma V_k'(s') \right] \mathbb{1}\{ \widetilde{s} = s \} \right]$$

---

[1] The definition after the second "=" of eq. (58) is the same as eq. (11)

$$\overset{(ii)}{=} \mathbb{E}_{\widetilde{s}\sim\mu_k}\big[\mathbb{1}\{\widetilde{s}=s\}\big]\mathbb{E}_{\widetilde{a}^{(1)}\sim\pi_k^{(1)}(s),\widetilde{a}^{(2)}\sim\pi_k^{(2)}(s),s'\sim\mathcal{P}(\cdot|s,\widetilde{a}^{(1)},\widetilde{a}^{(2)})}\big[R(s,\widetilde{a}^{(1)},\widetilde{a}^{(2)})+\gamma V_k'(s')\big]$$

$$\overset{(iii)}{=} \mu_k(s)\mathbb{E}_{\widetilde{a}^{(1)}\sim\pi_k^{(1)}(s),\widetilde{a}^{(2)}\sim\pi_k^{(2)}(s)}\big[Q_k(s,\widetilde{a}^{(1)},\widetilde{a}^{(2)})\big]$$

$$= \mu_{k,t}(s)\big[f_\tau\big(Q_k(s);\pi_k^{(1)}(s),\pi_k^{(2)}(s)\big)-\tau\mathcal{H}\big(\pi_k^{(1)}(s)\big)+\tau\mathcal{H}\big(\pi_k^{(2)}(s)\big)\big]$$

$$\overset{(iv)}{=} \mu_k(s)\big[V_{k+1}'(s)-\tau\mathcal{H}\big(\pi_k^{(1)}(s)\big)+\tau\mathcal{H}\big(\pi_k^{(2)}(s)\big)\big]$$

where (i) uses eq. (54), (ii) uses independence between $s'\sim\mu_k$ and $\widetilde{a}^{(1)}\sim\pi_k^{(1)}(s),\widetilde{a}^{(2)}\sim\pi_k^{(2)}(s),s'\sim\mathcal{P}(\cdot|s,\widetilde{a}^{(1)},\widetilde{a}^{(2)})$, (iii) uses eq. (57), and (iv) uses eq. (58). $\qquad\square$

**Lemma 3.** *If $\|\widehat{V}_0\|_\infty \le V_{\max} := \frac{1+\tau\ln A_{\max}}{1-\gamma}$, then for all $k\ge 0$, $0\le t\le T_k-1$, we have*

$$\|\widehat{V}_k\|_\infty,\|V_k'\|_\infty,\|V^*\|_\infty \le V_{\max}, \tag{66}$$

$$|\widehat{v}_{k+1}(s)| \le 2V_{\max}\widehat{\mu}_k(s),\forall s\in\mathcal{S}, \tag{67}$$

$$\max\big(\|Q_k\|_\infty,\|Q_{k,t}'^{(m)}\|_\infty,\|Q^*\|_\infty\big) \le Q_{\max}, \tag{68}$$

*where $Q_{\max} := 1+\gamma V_{\max} = \frac{1+\gamma\tau\ln A_{\max}}{1-\gamma}$ with $A_{\max} := \max\big(|\mathcal{A}^{(1)}|,|\mathcal{A}^{(2)}|\big)$.*

*Proof.* We will first prove $\|\widehat{V}_k\|_\infty \le V_{\max}$ in eq. (66) by induction. Suppose $\|\widehat{V}_{k'}\|_\infty \le V_{\max}$ holds for a certain $k'\in\mathbb{N}$. Then, eq. (59) implies that for all $s\in\mathcal{S}$,

$$|\widehat{V}_{k'+1}(s)| \le \Big|\frac{\widehat{v}_{k'+1}(s)}{\widehat{\mu}_{k'}'(s)}\Big|+\tau\big[\mathcal{H}\big(\pi_{k'}^{(1)}(s)\big)-\mathcal{H}\big(\pi_{k'}^{(2)}(s)\big)\big]$$

$$\le \frac{\sum_{i\in\mathcal{N}_{k'+1}}\big[\big[|R_i|+\gamma|\widehat{V}_{k'}(s_{i+1})|\big]\mathbb{1}\{s_i=s\}\big]}{\sum_{i\in\mathcal{N}_{k'+1}}\mathbb{1}\{s_i=s\}}+\tau\ln A_{\max}$$

$$\overset{(i)}{\le} \frac{\sum_{i\in\mathcal{N}_{k'+1}}\big[\big(1+\gamma V_{\max}\big)\mathbb{1}\{s_i=s\}\big]}{\sum_{i\in\mathcal{N}_{k'+1}}\mathbb{1}\{s_i=s\}}+\tau\ln A_{\max}$$

$$= 1+\frac{\gamma(1+\tau\ln A_{\max})}{1-\gamma}+\tau\ln A_{\max} = V_{\max}, \tag{69}$$

where (i) uses the inequality that $0\le\mathcal{H}(\pi^{(m)})\le\ln|\mathcal{A}^{(m)}|\le\ln A_{\max},\forall\pi^{(m)}\in\Delta(|\mathcal{A}^{(m)}|)$. Since $\|\widehat{V}_0\|_\infty\le V_{\max}$, $\|\widehat{V}_k\|_\infty\le V_{\max}$ for all $k\in\mathbb{N}$. A similar induction yields that $\|V_k'\|_\infty\le V_{\max}$.

Next, we prove $\|V^*\|_\infty\le V_{\max}$, $\|Q^*\|_\infty\le Q_{\max}$ in eqs. (66) & (68) respectively. Notice that $V^*$ and $Q^*$ have the following relation.

$$Q^*(s,a^{(1)},a^{(2)}) = R(s,a^{(1)},a^{(2)})+\gamma\mathbb{E}_{s'\sim\mathcal{P}(\cdot|s,a^{(1)},a^{(2)})}V^*(s'),$$

$$V^*(s) = \pi^{*(1)}(s)^\top Q^*(s)\pi^{*(2)}(s)+\tau\big[\mathcal{H}(\pi^{*(1)}(s))-\mathcal{H}(\pi^{*(2)}(s))\big].$$

Hence,

$$\|Q^*\|_\infty \le 1+\gamma\|V^*\|_\infty \tag{70}$$

$$\|V^*\|_\infty \le \|Q^*\|_\infty+\tau\ln A_{\max} \tag{71}$$

Substituting eq. (70) into eq. (71) yields that

$$\|V^*\|_\infty \le 1+\gamma\|V^*\|_\infty+\tau\ln A_{\max} \Rightarrow \|V^*\|_\infty \le V_{\max}.$$

The proof of eq. (66) is finished. Then, substituting $\|V^*\|_\infty \le V_{\max}$ into eq. (70) proves that $\|Q^*\|_\infty \le Q_{\max}$.

Equation (67) can be proved as follows.

$$\Big|\frac{\widehat{v}_{k+1}(s)}{\widehat{\mu}_k(s)}\Big| \overset{(i)}{\le} |\widehat{V}_{k+1}(s)|+\tau\big|\mathcal{H}\big(\pi_k^{(2)}(s)\big)-\mathcal{H}\big(\pi_k^{(1)}(s)\big)\big| \le V_{\max}+\tau\ln A_{\max} \overset{(ii)}{\le} 2V_{\max},$$

where (i) uses eq. (58) and (ii) uses $V_{\max} := \frac{1+\tau \ln A_{\max}}{1-\gamma} \geq \tau \ln A_{\max}$.

Next, we prove eq. (68). It can be easily seen from eqs. (57) & (66) that

$$\left|Q_k(s, a^{(1)}, a^{(2)})\right| \leq \left|R(s, a^{(1)}, a^{(2)})\right| + \gamma \mathbb{E}_{s' \sim \mathcal{P}(\cdot|s, a^{(1)}, a^{(2)})}\left|V_k'(s')\right| \leq 1 + \gamma V_{\max} = Q_{\max},$$

which implies that $\|Q_k\|_\infty \leq Q_{\max}$. Hence,

$$\left|Q_{k,t}'^{(m)}(s, a^{(m)})\right| \overset{(i)}{\leq} \mathbb{E}_{a^{(\backslash m)} \sim \pi_{k,t}'^{(\backslash m)}(s)}\left|Q_k(s, a^{(1)}, a^{(2)})\right| \leq Q_{\max}$$

where (i) uses eq. (64). This proves eq. (68). $\qquad\square$

With the above two Lemmas, we can derive the estimation error bounds as follows.

**Lemma 4.** *Use hyperparameter choices $N_{k,t}, \overline{N}_{k,t} \geq \frac{650 t_{mix} A_{\max}}{\mu_{\min}} \ln\left(\frac{20 T_{sum}|\mathcal{S}|A_{\max}}{\delta\sqrt{\mu_{\min}}}\right)$ and $N_{k+1} \geq \frac{650 t_{mix}}{\mu_{\min}(1-\gamma)^2} \ln\left(\frac{4}{\delta\sqrt{\mu_{\min}}}\right)$ in Algorithm 1. Then with probability at least $1 - \delta$, all the following bounds hold for all $0 \leq k \leq K - 1$, $0 \leq t \leq T_k - 1$, $m = 1, 2$, $s \in \mathcal{S}$, $a^{(m)} \in \mathcal{A}^{(m)}$, where $T_{sum} := \sum_{k=0}^{K-1} T_k$.*

$$\left\|\widehat{V}_k - V_k'\right\|_\infty \leq 2 V_{\max} \gamma^k + \frac{171 V_{\max}}{1-\gamma} \sqrt{\frac{t_{mix}}{N_{k+1}\mu_{\min}} \ln\left(\frac{4K|\mathcal{S}|}{\delta\sqrt{\mu_{\min}}}\right)} \tag{72}$$

$$\left\|\widehat{Q}_{k,t}^{(m)} - Q_{k,t}^{(m)}\right\|_\infty$$
$$\leq Q_{\max}\left[\frac{640 t_{mix} A_{\max}}{N_{k,t}\mu_{\min}\epsilon'} \ln\left(\frac{20 T_{sum}|\mathcal{S}|A_{\max}}{\delta\sqrt{\mu_{\min}}}\right) + 25\sqrt{\frac{t_{mix}A_{\max}}{N_{k,t}\mu_{\min}\epsilon'} \ln\left(\frac{20 T_{sum}|\mathcal{S}|A_{\max}}{\delta\sqrt{\mu_{\min}}}\right)} + \epsilon'\right]$$
$$+ 2\left\|\widehat{V}_k - V_k'\right\|_\infty \tag{73}$$

$$\left\|\widehat{\overline{Q}}_{k,t}^{(m)} - \overline{Q}_{k,t}^{(m)}\right\|_\infty$$
$$\leq Q_{\max}\left[\frac{640 t_{mix} A_{\max}}{\overline{N}_{k,t}\mu_{\min}\overline{\epsilon}'} \ln\left(\frac{20 T_{sum}|\mathcal{S}|A_{\max}}{\delta\sqrt{\mu_{\min}}}\right) + 25\sqrt{\frac{t_{mix}A_{\max}}{\overline{N}_{k,t}\mu_{\min}\overline{\epsilon}'} \ln\left(\frac{20 T_{sum}|\mathcal{S}|A_{\max}}{\delta\sqrt{\mu_{\min}}}\right)} + \overline{\epsilon}'\right]$$
$$+ 2\left\|\widehat{V}_k - V_k'\right\|_\infty. \tag{74}$$

*More specifically, the upper bounds (73)&(74) can be simplified respectively as follows when $\epsilon' = \left[\frac{650 t_{mix} A_{\max}}{N_{k,t}\mu_{\min}} \ln\left(\frac{20 T_{sum}|\mathcal{S}|A_{\max}}{\delta\sqrt{\mu_{\min}}}\right)\right]^{1/3}$ and $\overline{\epsilon}' = \left[\frac{650 t_{mix} A_{\max}}{\overline{N}_{k,t}\mu_{\min}} \ln\left(\frac{20 T_{sum}|\mathcal{S}|A_{\max}}{\delta\sqrt{\mu_{\min}}}\right)\right]^{1/3}$.*

$$\left\|\widehat{Q}_{k,t}^{(m)} - Q_{k,t}^{(m)}\right\|_\infty \leq 18 Q_{\max}\left[\frac{t_{mix}A_{\max}}{N_{k,t}\mu_{\min}} \ln\left(\frac{20 T_{sum}|\mathcal{S}|A_{\max}}{\delta\sqrt{\mu_{\min}}}\right)\right]^{1/3} + 2\left\|\widehat{V}_k - V_k'\right\|_\infty, \tag{75}$$

$$\left\|\widehat{\overline{Q}}_{k,t}^{(m)} - \overline{Q}_{k,t}^{(m)}\right\|_\infty \leq 18 Q_{\max}\left[\frac{t_{mix}A_{\max}}{\overline{N}_{k,t}\mu_{\min}} \ln\left(\frac{20 T_{sum}|\mathcal{S}|A_{\max}}{\delta\sqrt{\mu_{\min}}}\right)\right]^{1/3} + 2\left\|\widehat{V}_k - V_k'\right\|_\infty. \tag{76}$$

*Proof.* If the initial state distribution of the minibatch $\mathcal{N}_{k,t}$ equals $\mu_{k,t}'$, i.e., $s_{k,t,0} \sim \mu_{k,t}'$, then $N_{k,t}\widehat{q}_{k,t}^{(m)}(s, a^{(m)}) := \sum_{i \in \mathcal{N}_{k,t}} g(s_i, a_i^{(1)}, a_i^{(2)}, s_{i+1})$ $(g(s_i, a_i^{(1)}, a_i^{(2)}, s_{i+1}) := [R_i + \gamma\widehat{V}_k(s_{i+1})]\mathbb{1}\{s_i = s, a_i^{(m)} = a^{(m)}\})$ and its expected value $N_{k,t}q_{k,t}'^{(m)}(s, a^{(m)})$ satisfy the following concentration bound.

$$\mathbb{P}_{\mu_{k,t}'}\left\{\left|N_{k,t}\widehat{q}_{k,t}^{(m)}(s, a^{(m)}) - N_{k,t}q_{k,t}'^{(m)}(s, a^{(m)})\right| \geq u\right\}$$
$$\overset{(i)}{\leq} 2\exp\left[-\frac{u^2\gamma_{ps}}{8(N_{k,t} + 1/\gamma_{ps})Q_{\max}^2\mu_{k,t}'(s)\pi_{k,t}'^{(m)}(a^{(m)}|s) + 40u Q_{\max}}\right]$$
$$\overset{(ii)}{\leq} 2\exp\left[-\frac{u^2/(2t_{\mathrm{mix}})}{8(N_{k,t} + 2t_{\mathrm{mix}})Q_{\max}^2\mu_{k,t}'(s)\pi_{k,t}'^{(m)}(a^{(m)}|s) + 40u Q_{\max}}\right], \tag{77}$$

where $\mathbb{P}_{\mu'_{k,t}}$ is under the initial state distribution $\mu'_{k,t}$, (i) uses Theorem 3.4 of (Paulin, 2015) and the inequalities that $\left|g(s_i, a_i^{(1)}, a_i^{(2)}, s_{i+1}) - \mathbb{E}_{s_i \sim \mu'_{k,t}} g(s_i, a_i^{(1)}, a_i^{(2)}, s_{i+1})\right| \leq 2(1 + \gamma V_{\max}) = 2Q_{\max}$ (Based on Lemma 3) and that $\text{var}_{s_i \sim \mu'_{k,t}}\left[g(s_i, a_i^{(1)}, a_i^{(2)}, s_{i+1})\right] \leq \mathbb{E}_{s_i \sim \mu'_{k,t}}\left[\left[R_i + \gamma V'_k(s_{i+1})\right]^2 \mathbb{1}\{s_i = s, a_i^{(m)} = a^{(m)}\}\right] \leq (1 + \gamma V_{\max})^2 \mu'_{k,t}(s) \pi'^{(m)}_{k,t}(a^{(m)}|s) = Q_{\max}^2 \mu'_{k,t}(s) \pi'^{(m)}_{k,t}(a^{(m)}|s)$, (ii) uses Proposition 3.4 of (Paulin, 2015) which states that the pseudo spectral gap $\gamma_{ps}$ has a lower bound $1/(2t_{\text{mix}})$ for any uniformly ergodic Markov chain (This condition holds for our MDP with finitely many states and actions). Then, for any initial state distribution $s_{t,k,0} \sim \xi$, Proposition 3.10 of (Paulin, 2015) implies that

$$
\mathbb{P}_\xi\left\{\left|N_{k,t}\widehat{q}^{(m)}_{k,t}(s, a^{(m)}) - N_{k,t}q'^{(m)}_{k,t}(s, a^{(m)})\right| \geq u\right\}
$$
$$
\leq \sqrt{\mathbb{E}_{s \sim \xi}\left[\frac{\xi(s)}{\mu'_{k,t}(s)}\right] \mathbb{P}_{\mu'_{k,t}}\left\{\left|N_{k,t}\widehat{q}^{(m)}_{k,t}(s, a^{(m)}) - N_{k,t}q'^{(m)}_{k,t}(s, a^{(m)})\right| \geq u\right\}}
$$
$$
\stackrel{(i)}{\leq} \sqrt{\frac{1}{\mu_{\min}}} \exp\left[-\frac{u^2/(4t_{\text{mix}})}{8(N_{k,t} + 2t_{\text{mix}})Q_{\max}^2 \mu'_{k,t}(s) \pi'^{(m)}_{k,t}(a^{(m)}|s) + 40uQ_{\max}}\right], \qquad (78)
$$

where (i) uses eq. (77) and the inequality that $\mathbb{E}_{s \sim \xi}\left[\frac{\xi(s)}{\mu'_{k,t}(s)}\right] \leq \frac{1}{\mu_{\min}}$.

In a similar way, the following concentration bound can be proved for $N_{k,t}\widehat{\mu}_{k,t}(s) := \sum_{i \in \mathcal{N}_{k,t}} \mathbb{1}\{s_i = s\}$ and $N_{k,t}\mu'_{k,t}(s) = \mathbb{E}_{\mu'_{k,t}}\left[N_{k,t}\widehat{\mu}_{k,t}(s)\right]$.

$$
\mathbb{P}_\xi\left\{\left|N_{k,t}\widehat{\mu}_{k,t}(s) - N_{k,t}\mu'_{k,t}(s)\right| \geq u\right\} \leq \sqrt{\frac{1}{\mu_{\min}}} \exp\left[-\frac{u^2/(4t_{\text{mix}})}{8(N_{k,t} + 2t_{\text{mix}})\mu'_{k,t}(s) + 40u}\right], \quad (79)
$$

where we use the inequalities that $\text{var}_{\mu'_{k,t}} \mathbb{1}\{s_i = s\} \leq \mu'_{k,t}(s)$ and that $\left|\mathbb{1}\{s_i = s\} - \mathbb{E}_{\mu'_{k,t}} \mathbb{1}\{s_i = s\}\right| \leq 1$. Letting the right hand sides of eqs. (78)&(79) be upper bounded by $\delta/4$ and applying the union bound yields that, with probability at least $1-\delta/2$, the following two inequalities simultaneously hold.

$$
\left|\widehat{q}^{(m)}_{k,t}(s, a^{(m)}) - q'^{(m)}_{k,t}(s, a^{(m)})\right|
$$
$$
\leq \frac{160Q_{\max}}{N_{k,t}} t_{\text{mix}} \ln\left(\frac{4}{\delta\sqrt{\mu_{\min}}}\right)
$$
$$
+ \frac{6Q_{\max}}{\sqrt{N_{k,t}}} \sqrt{t_{\text{mix}}(1 + 2t_{\text{mix}}/N_{k,t})\mu'_{k,t}(s) \pi'^{(m)}_{k,t}(a^{(m)}|s) \ln\left(\frac{4}{\delta\sqrt{\mu_{\min}}}\right)}, \qquad (80)
$$
$$
\left|\widehat{\mu}_{k,t}(s) - \mu'_{k,t}(s)\right|
$$
$$
\leq \frac{160}{N_{k,t}} t_{\text{mix}} \ln\left(\frac{4}{\delta\sqrt{\mu_{\min}}}\right)
$$
$$
+ \frac{6}{\sqrt{N_{k,t}}} \sqrt{t_{\text{mix}}(1 + 2t_{\text{mix}}/N_{k,t})\mu'_{k,t}(s) \ln\left(\frac{4}{\delta\sqrt{\mu_{\min}}}\right)} \stackrel{(i)}{\leq} \frac{1}{2}\mu'_{k,t}(s), \qquad (81)
$$

where (i) holds since $N_{k,t} \geq \frac{650t_{\text{mix}}A_{\max}}{\mu_{\min}} \ln\left(\frac{20T_{\text{sum}}|\mathcal{S}|A_{\max}}{\delta\sqrt{\mu_{\min}}}\right) \geq 650t_{\text{mix}}$ and $\mu'_{k,t}(s) \geq \mu_{\min}$.

Similarly, it can be proved that the following two inequalities holds with probability at least $1 - \delta/2$.

$$
\left|\widehat{v}_{k+1}(s) - v'_{k+1}(s)\right|
$$
$$
\leq \frac{160Q_{\max}}{N_{k+1}} t_{\text{mix}} \ln\left(\frac{4}{\delta\sqrt{\mu_{\min}}}\right) + \frac{6Q_{\max}}{\sqrt{N_{k+1}}} \sqrt{t_{\text{mix}}(1 + 2t_{\text{mix}}/N_{k+1})\mu_k(s) \ln\left(\frac{4}{\delta\sqrt{\mu_{\min}}}\right)}, \quad (82)
$$
$$
\left|\widehat{\mu}_k(s) - \mu_k(s)\right|
$$
$$
\leq \frac{160}{N_{k+1}} t_{\text{mix}} \ln\left(\frac{4}{\delta\sqrt{\mu_{\min}}}\right) + \frac{6}{\sqrt{N_{k+1}}} \sqrt{t_{\text{mix}}(1 + 2t_{\text{mix}}/N_{k+1})\mu_k(s) \ln\left(\frac{4}{\delta\sqrt{\mu_{\min}}}\right)}. \quad (83)
$$

Hence, eqs. (80)-(83) hold with probability at least $1 - \delta$. In this case, we have that

$$
\begin{aligned}
&\left|\widehat{v}_{k+1}(s) - v_{k+1}(s)\right| \\
&\leq \left|\widehat{v}_{k+1}(s) - v'_{k+1}(s)\right| + \left|v'_{k+1}(s) - v_{k+1}(s)\right| \\
&\overset{(i)}{\leq} \frac{160 Q_{\max}}{N_{k+1}} t_{\mathrm{mix}} \ln\left(\frac{4}{\delta\sqrt{\mu_{\min}}}\right) + \frac{6 Q_{\max}}{\sqrt{N_{k+1}}} \sqrt{t_{\mathrm{mix}}(1 + 2t_{\mathrm{mix}}/N_{k+1})\mu_k(s) \ln\left(\frac{4}{\delta\sqrt{\mu_{\min}}}\right)} \\
&\quad + \gamma \mathbb{E}_{\mu_k, \pi_k^{(1)}, \pi_k^{(2)}}\left[\left[\widehat{V}_k(s') - V'_k(s')\right]\mathbb{1}\{\widetilde{s} = s\}\right] \\
&\leq \frac{160 Q_{\max}}{N_{k+1}} t_{\mathrm{mix}} \ln\left(\frac{4}{\delta\sqrt{\mu_{\min}}}\right) + \frac{6 Q_{\max}}{\sqrt{N_{k+1}}} \sqrt{t_{\mathrm{mix}}(1 + 2t_{\mathrm{mix}}/N_{k+1})\mu_k(s) \ln\left(\frac{4}{\delta\sqrt{\mu_{\min}}}\right)} \\
&\quad + \gamma \max_{s'}\left|\widehat{V}_k(s') - V'_k(s')\right| \mathbb{E}_{\mu_k} \mathbb{1}\{\widetilde{s} = s\} \\
&\leq \frac{160 Q_{\max}}{N_{k+1}} t_{\mathrm{mix}} \ln\left(\frac{4}{\delta\sqrt{\mu_{\min}}}\right) + \frac{6 Q_{\max}}{\sqrt{N_{k+1}}} \sqrt{t_{\mathrm{mix}}(1 + 2t_{\mathrm{mix}}/N_{k+1})\mu_k(s) \ln\left(\frac{4}{\delta\sqrt{\mu_{\min}}}\right)} \\
&\quad + \gamma \mu_k(s)\left\|\widehat{V}_k - V'_k\right\|_\infty,
\end{aligned}
\tag{84}
$$

where (i) uses eqs. (54), (55) & (82).

$$
\begin{aligned}
&|\widehat{V}_{k+1}(s) - V'_{k+1}(s)| \\
&\overset{(i)}{=} \left|\frac{\widehat{v}_{k+1}(s)}{\widehat{\mu}_k(s)} - \frac{v_{k+1}(s)}{\mu_k(s)}\right| \\
&\leq \mu_k^{-1}(s)\left|\widehat{v}_{k+1}(s) - v_{k+1}(s)\right| + \left|\widehat{v}_{k+1}(s)\right|\left|\frac{\mu_k(s) - \widehat{\mu}_k(s)}{\mu_k(s)\widehat{\mu}_k(s)}\right| \\
&\overset{(ii)}{\leq} \mu_k^{-1}(s)\left[\frac{160 Q_{\max}}{N_{k+1}} t_{\mathrm{mix}} \ln\left(\frac{4}{\delta\sqrt{\mu_{\min}}}\right) + \frac{6 Q_{\max}}{\sqrt{N_{k+1}}} \sqrt{t_{\mathrm{mix}}(1 + 2t_{\mathrm{mix}}/N_{k+1})\mu_k(s) \ln\left(\frac{4}{\delta\sqrt{\mu_{\min}}}\right)}\right. \\
&\quad \left. + \gamma \mu_k(s)\left\|\widehat{V}_k - V'_k\right\|_\infty\right] + 2 V_{\max} \mu_k^{-1}(s)\left[\frac{160}{N_{k+1}} t_{\mathrm{mix}} \ln\left(\frac{4}{\delta\sqrt{\mu_{\min}}}\right)\right. \\
&\quad \left. + \frac{6}{\sqrt{N_{k+1}}} \sqrt{t_{\mathrm{mix}}(1 + 2t_{\mathrm{mix}}/N_{k+1})\mu_k(s) \ln\left(\frac{4}{\delta\sqrt{\mu_{\min}}}\right)}\right] \\
&\overset{(iii)}{\leq} \gamma\|\widehat{V}_k - V'_k\|_\infty + \frac{V_{\max}}{\mu_k(s)}\left[\frac{480 t_{\mathrm{mix}}}{N_{k+1}} \ln\left(\frac{4}{\delta\sqrt{\mu_{\min}}}\right)\right. \\
&\quad \left. + 18 \sqrt{\frac{t_{\mathrm{mix}}\mu_k(s)}{N_{k+1}}(1 + 2t_{\mathrm{mix}}/N_{k+1}) \ln\left(\frac{4}{\delta\sqrt{\mu_{\min}}}\right)}\right] \\
&\overset{(iv)}{\leq} \gamma\|\widehat{V}_k - V'_k\|_\infty + V_{\max}\left[24 \frac{40 t_{\mathrm{mix}}}{N_{k+1}\mu_{\min}} \ln\left(\frac{4}{\delta\sqrt{\mu_{\min}}}\right) + 18 \sqrt{\frac{1.05 t_{\mathrm{mix}}}{N_{k+1}\mu_{\min}} \ln\left(\frac{4}{\delta\sqrt{\mu_{\min}}}\right)}\right] \\
&\overset{(v)}{\leq} \gamma\|\widehat{V}_k - V'_k\|_\infty + V_{\max}\left[24 \sqrt{\frac{40 t_{\mathrm{mix}}}{N_{k+1}\mu_{\min}} \ln\left(\frac{4}{\delta\sqrt{\mu_{\min}}}\right)} + 18 \sqrt{\frac{1.05 t_{\mathrm{mix}}}{N_{k+1}\mu_{\min}} \ln\left(\frac{4}{\delta\sqrt{\mu_{\min}}}\right)}\right] \\
&\leq \gamma\|\widehat{V}_k - V'_k\|_\infty + 171 V_{\max} \sqrt{\frac{t_{\mathrm{mix}}}{N_{k+1}\mu_{\min}} \ln\left(\frac{4}{\delta\sqrt{\mu_{\min}}}\right)},
\end{aligned}
\tag{85}
$$

where (i) uses eqs. (58)&(59), (ii) uses eqs. (67), (83)&(84), (iii) uses $Q_{\max} \leq V_{\max}$, (iv) uses $\mu_k(s) \geq \mu_{\min}$ and $N_{k+1} \geq \frac{40 t_{\mathrm{mix}}}{\mu_{\min}} \ln\left(\frac{4}{\delta\sqrt{\mu_{\min}}}\right) \geq 40 t_{\mathrm{mix}}$, and (v) uses $N_{k+1} \geq \frac{40 t_{\mathrm{mix}}}{\mu_{\min}} \ln\left(\frac{4}{\delta\sqrt{\mu_{\min}}}\right)$. Applying the union bound to the above inequality over all $0 \leq k \leq K - 1$, $s \in \mathcal{S}$ and taking maximum over $s \in \mathcal{S}$, we obtain that with probability at least $1 - \delta$,

$$
\|\widehat{V}_{k+1} - V'_{k+1}\|_\infty \leq \gamma\|\widehat{V}_k - V'_k\|_\infty + 171 V_{\max} \sqrt{\frac{t_{\mathrm{mix}}}{N_{k+1}\mu_{\min}} \ln\left(\frac{4}{\delta\sqrt{\mu_{\min}}}\right)}.
$$

Iterating the above inequality yields that with probability at least $1 - \delta$,

$$
\begin{aligned}
\|\widehat{V}_k - V'_k\|_\infty &\leq \gamma^k \|\widehat{V}_0 - V'_0\|_\infty + \frac{171 V_{\max}}{1 - \gamma} \sqrt{\frac{t_{\mathrm{mix}}}{N_{k+1}\mu_{\min}} \ln\left(\frac{4K|\mathcal{S}|}{\delta\sqrt{\mu_{\min}}}\right)} \\
&\overset{(i)}{\leq} 2V_{\max}\gamma^k + \frac{171 V_{\max}}{1 - \gamma} \sqrt{\frac{t_{\mathrm{mix}}}{N_{k+1}\mu_{\min}} \ln\left(\frac{4K|\mathcal{S}|}{\delta\sqrt{\mu_{\min}}}\right)},
\end{aligned}
\tag{86}
$$

where (i) uses the fact that $\|V'_0\|_\infty, \|\widehat{V}_0\|_\infty \leq V_{\max}$.

Hence, with probability at least $1 - 2\delta$, eqs. (80), (81) & (86) hold simultaneously. In this case,

$$
\begin{aligned}
\left|\widehat{Q}_{k,t}^{(m)}(s, a^{(m)}) - Q_{k,t}^{\prime(m)}(s, a^{(m)})\right| &\overset{(i)}{=} \left|\frac{\widehat{q}_{k,t}^{(m)}(s, a^{(m)})}{\widehat{\mu}_{k,t}(s)\pi_{k,t}^{\prime(m)}(a^{(m)}|s)} - \frac{q_{k,t}^{(m)}(s, a^{(m)})}{\mu'_{k,t}(s)\pi_{k,t}^{\prime(m)}(a^{(m)}|s)}\right| \\
&\overset{(ii)}{\leq} \frac{\left|\widehat{q}_{k,t}^{(m)}(s, a^{(m)}) - q_{k,t}^{(m)}(s, a^{(m)})\right|}{\widehat{\mu}_{k,t}(s)\pi_{k,t}^{\prime(m)}(a^{(m)}|s)} \\
&\quad + |\mu'_{k,t}(s)\pi_{k,t}^{\prime(m)}(a^{(m)}|s)Q_{k,t}^{\prime(m)}(s, a^{(m)})| \frac{\left|\widehat{\mu}_{k,t}(s) - \mu'_{k,t}(s)\right|}{\widehat{\mu}_{k,t}(s)\mu'_{k,t}(s)\pi_{k,t}^{\prime(m)}(a^{(m)}|s)} \\
&\overset{(iii)}{\leq} \frac{2\left|\widehat{q}_{k,t}^{(m)}(s, a^{(m)}) - q_{k,t}^{\prime(m)}(s, a^{(m)})\right| + 2\left|q_{k,t}^{\prime(m)}(s, a^{(m)}) - q_{k,t}^{(m)}(s, a^{(m)})\right|}{\mu'_{k,t}(s)\pi_{k,t}^{\prime(m)}(a^{(m)}|s)} \\
&\quad + \frac{2Q_{\max}\left|\widehat{\mu}_{k,t}(s) - \mu'_{k,t}(s)\right|}{\mu'_{k,t}(s)} \\
&\overset{(iv)}{\leq} \frac{2Q_{\max}}{\mu'_{k,t}(s)\pi_{k,t}^{\prime(m)}(a^{(m)}|s)} \left[\frac{320}{N_{k,t}} t_{\mathrm{mix}} \ln\left(\frac{4}{\delta\sqrt{\mu_{\min}}}\right)\right. \\
&\quad \left. + \frac{12}{\sqrt{N_{k,t}}} \sqrt{t_{\mathrm{mix}}(1 + 2t_{\mathrm{mix}}/N_{k,t})\mu'_{k,t}(s)\pi_{k,t}^{\prime(m)}(a^{(m)}|s) \ln\left(\frac{4}{\delta\sqrt{\mu_{\min}}}\right)}\right] + 2\|\widehat{V}_k - V'_k\|_\infty \\
&\overset{(v)}{\leq} Q_{\max}\left[\frac{640 t_{\mathrm{mix}} A_{\max}}{N_{k,t}\mu_{\min}\epsilon'} \ln\left(\frac{4}{\delta\sqrt{\mu_{\min}}}\right) + 25\sqrt{\frac{t_{\mathrm{mix}} A_{\max}}{N_{k,t}\mu_{\min}\epsilon'} \ln\left(\frac{4}{\delta\sqrt{\mu_{\min}}}\right)}\right] + 2\|\widehat{V}_k - V'_k\|_\infty,
\end{aligned}
\tag{87}
$$

where (i) uses eqs. (64)&(65), (ii) uses eq. (64), (iii) uses eq. (68) and the inequality that $\widehat{\mu}_{k,t}(s) \geq \mu'_{k,t}(s)/2$ implied by (i) of eq. (81), (iv) uses eqs. (80)&(81) and the following inequality based on eqs. (61)&(62), and (v) uses $\mu'_{k,t}(s) \geq \mu_{\min}$ (based on Assumption 1), $\pi_{k,t}^{\prime(m)}(a^{(m)}|s) = (1 - \epsilon')\pi_{k,t}^{(m)}(a^{(m)}|s) + \epsilon'/|\mathcal{A}^{(m)}| \geq \epsilon'/A_{\max}$ and $N_{k,t} \geq 650 t_{\mathrm{mix}}$.

$$
\begin{aligned}
&\left|q_{k,t}^{\prime(m)}(s, a^{(m)}) - q_{k,t}^{(m)}(s, a^{(m)})\right| \\
&\leq \gamma\mathbb{E}_{\mu'_{k,t},\pi_{k,t}^{\prime(1)},\pi_{k,t}^{\prime(2)}}\left[\left|\widehat{V}_k(s') - V'_k(s')\right|\mathbb{1}\{\widetilde{s} = s, \widetilde{a}^{(m)} = a^{(m)}\}\right] \\
&\leq \mu'_{k,t}(s)\pi_{k,t}^{\prime(m)}(s, a^{(m)})\|\widehat{V}_k - V'_k\|_\infty.
\end{aligned}
$$

Notice that the following inequality always holds.

$$
\begin{aligned}
&\left|Q_{k,t}^{\prime(m)}(s, a^{(m)}) - Q_{k,t}^{(m)}(s, a^{(m)})\right| \\
&\leq \sum_{a^{(\backslash m)}} \left|Q_k(s, a^{(1)}, a^{(2)})\right| \left|\pi_{k,t}^{\prime(\backslash m)}(a^{(\backslash m)}|s) - \pi_{k,t}^{(\backslash m)}(a^{(\backslash m)}|s)\right| \\
&\overset{(i)}{\leq} Q_{\max}\epsilon'\left|\pi_{k,t}^{(\backslash m)}(a^{(\backslash m)}|s) - |\mathcal{A}^{(m)}|^{-1}\right| \leq Q_{\max}\epsilon',
\end{aligned}
\tag{88}
$$

where (i) uses eq. (68).

Hence, combining eqs. (87)&(88), it can be seen that with probability at least $1 - 2\delta$, the following inequality holds

$$
\big|\widehat{Q}_{k,t}^{(m)}(s, a^{(m)}) - Q_{k,t}^{(m)}(s, a^{(m)})\big|
$$
$$
\leq Q_{\max}\left[\frac{640 t_{\mathrm{mix}} A_{\max}}{N_{k,t}\mu_{\min}\epsilon'}\ln\left(\frac{4}{\delta\sqrt{\mu_{\min}}}\right) + 25\sqrt{\frac{t_{\mathrm{mix}} A_{\max}}{N_{k,t}\mu_{\min}\epsilon'}\ln\left(\frac{4}{\delta\sqrt{\mu_{\min}}}\right)} + \epsilon'\right]
$$
$$
+ 2\big\|\widehat{V}_k - V_k'\big\|_\infty. \tag{89}
$$

Similarly, it can be proved that with probability at least $1 - 2\delta$,

$$
\big|\widehat{\overline{Q}}_{k,t}^{(m)}(s, a^{(m)}) - \overline{Q}_{k,t}^{(m)}(s, a^{(m)})\big|
$$
$$
\leq Q_{\max}\left[\frac{640 t_{\mathrm{mix}} A_{\max}}{\overline{N}_{k,t}\mu_{\min}\overline{\epsilon}'}\ln\left(\frac{4}{\delta\sqrt{\mu_{\min}}}\right) + 25\sqrt{\frac{t_{\mathrm{mix}} A_{\max}}{\overline{N}_{k,t}\mu_{\min}\overline{\epsilon}'}\ln\left(\frac{4}{\delta\sqrt{\mu_{\min}}}\right)} + \overline{\epsilon}'\right]
$$
$$
+ 2\big\|\widehat{V}_k - V_k'\big\|_\infty. \tag{90}
$$

Finally, eqs. (72)-(74) are proved by applying a union bound to eq. (86) and to eqs. (89)&(90) over all $0 \leq k \leq T - 1$, $0 \leq t \leq T_k - 1$, $m = 1, 2$, $s \in \mathcal{S}$, $a^{(m)} \in \mathcal{A}^{(m)}$.

Now we consider the hyperparameter choice $\epsilon' = \left[\frac{650 t_{\mathrm{mix}} A_{\max}}{N_{k,t}\mu_{\min}}\ln\left(\frac{20 T_{\mathrm{sum}}|\mathcal{S}|A_{\max}}{\delta\sqrt{\mu_{\min}}}\right)\right]^{1/3}$. First, this is a valid choice, since $N_{k,t} \geq \frac{650 t_{\mathrm{mix}} A_{\max}}{\mu_{\min}}\ln\left(\frac{20 T_{\mathrm{sum}}|\mathcal{S}|A_{\max}}{\delta\sqrt{\mu_{\min}}}\right)$ implies that $\epsilon' \in [0, 1]$. Then, for this $\epsilon'$, the upper bound (73) is simplified as follows.

$$
\big\|\widehat{Q}_{k,t}^{(m)} - Q_{k,t}^{(m)}\big\|_\infty
$$
$$
\leq Q_{\max}\left[\frac{640 t_{\mathrm{mix}} A_{\max}}{N_{k,t}\mu_{\min}\epsilon'}\ln\left(\frac{20 T_{\mathrm{sum}}|\mathcal{S}|A_{\max}}{\delta\sqrt{\mu_{\min}}}\right) + 25\sqrt{\frac{t_{\mathrm{mix}} A_{\max}}{N_{k,t}\mu_{\min}\epsilon'}\ln\left(\frac{20 T_{\mathrm{sum}}|\mathcal{S}|A_{\max}}{\delta\sqrt{\mu_{\min}}}\right)} + \epsilon'\right]
$$
$$
+ 2\big\|\widehat{V}_k - V_k'\big\|_\infty
$$
$$
\overset{(i)}{\leq} Q_{\max}\left(\frac{640}{650}\epsilon'^2 + \frac{25}{\sqrt{650}}\epsilon'\right) + 2\big\|\widehat{V}_k - V_k'\big\|_\infty
$$
$$
\overset{(ii)}{\leq} 2Q_{\max}\epsilon' + 2\big\|\widehat{V}_k - V_k'\big\|_\infty = 18 Q_{\max}\left[\frac{t_{\mathrm{mix}} A_{\max}}{N_{k,t}\mu_{\min}}\ln\left(\frac{20 T_{\mathrm{sum}}|\mathcal{S}|A_{\max}}{\delta\sqrt{\mu_{\min}}}\right)\right]^{1/3} + 2\big\|\widehat{V}_k - V_k'\big\|_\infty,
$$

where (i) uses $\frac{t_{\mathrm{mix}} A_{\max}}{N_{k,t}\mu_{\min}}\ln\left(\frac{20 T_{\mathrm{sum}}|\mathcal{S}|A_{\max}}{\delta\sqrt{\mu_{\min}}}\right) = \epsilon'^3/650$, and (ii) uses $\epsilon' \in [0, 1]$. This proves eq. (75). The proof of eq. (76) is similar. $\qquad\square$

## C PROPERTIES OF THE DUALITY GAP

In this section, we prove some useful properties of the duality gap

$$
D^{(\tau)}(\pi^{(1)}, \pi^{(2)}) := \max_{s,\pi'^{(1)},\pi'^{(2)}}\left[V_{\pi'^{(1)},\pi^{(2)}}^{(\tau)}(s) - V_{\pi^{(1)},\pi'^{(2)}}^{(\tau)}(s)\right].
$$

**Lemma 5.** *For any policy pair $\pi^{(1)}, \pi^{(2)}$, it holds that*

$$
D^{(\tau)}(\pi^{(1)}, \pi^{(2)}) \leq \frac{1}{1 - \gamma}\max_{s,\pi'^{(1)}(s),\pi'^{(2)}(s)}\left[f_\tau\big[Q_*^{(\tau)}(s), \pi'^{(1)}(s), \pi^{(2)}(s)\big] - f_\tau\big[Q_*^{(\tau)}(s), \pi^{(1)}(s), \pi'^{(2)}(s)\big]\right].
$$

This Lemma generalized the Lemma 32 of (Wei et al., 2021) to entropy-regularized Markov game.

*Proof.* Throughout this proof, we denote the policy pair $(\pi_{*\tau}^{(1)}, \pi_{*\tau}^{(2)})$ as the Nash equilibrium and their associated V-function and Q-function are respectively denoted as $V_*^{(\tau)}$ and $Q_*^{(\tau)}$.

Note that

$$
V^{(\tau)}_{\pi'^{(1)},\pi^{(2)}}(s) - V^{(\tau)}_*(s)
$$

$$
= \sum_{a^{(1)},a^{(2)}} \left[ Q^{(\tau)}_{\pi'^{(1)},\pi^{(2)}}(s,a^{(1)},a^{(2)})\pi'^{(1)}(a^{(1)}|s)\pi^{(2)}(a^{(2)}|s) \right.
$$

$$
\left. - Q^{(\tau)}_*(s,a^{(1)},a^{(2)})\pi^{(1)}_{*\tau}(a^{(1)}|s)\pi^{(2)}_{*\tau}(a^{(2)}|s) \right]
$$

$$
+ \tau \left[ \mathcal{H}\big(\pi'^{(1)}(s)\big) - \mathcal{H}\big(\pi^{(2)}(s)\big) - \mathcal{H}\big(\pi^{(1)}_{*\tau}(s)\big) + \mathcal{H}\big(\pi^{(2)}_{*\tau}(s)\big) \right]
$$

$$
= \sum_{a^{(1)},a^{(2)}} \left[ Q^{(\tau)}_{\pi'^{(1)},\pi^{(2)}}(s,a^{(1)},a^{(2)}) - Q^{(\tau)}_*(s,a^{(1)},a^{(2)}) \right]\pi'^{(1)}(a^{(1)}|s)\pi^{(2)}(a^{(2)}|s)
$$

$$
+ \sum_{a^{(1)},a^{(2)}} Q^{(\tau)}_*(s,a^{(1)},a^{(2)})\left[ \pi'^{(1)}(a^{(1)}|s)\pi^{(2)}(a^{(2)}|s) - \pi^{(1)}_{*\tau}(a^{(1)}|s)\pi^{(2)}_{*\tau}(a^{(2)}|s) \right]
$$

$$
+ \tau \left[ \mathcal{H}\big(\pi'^{(1)}(s)\big) - \mathcal{H}\big(\pi^{(2)}(s)\big) - \mathcal{H}\big(\pi^{(1)}_{*\tau}(s)\big) + \mathcal{H}\big(\pi^{(2)}_{*\tau}(s)\big) \right]
$$

$$
= \gamma \sum_{a^{(1)},a^{(2)}} \left[ \pi'^{(1)}(a^{(1)}|s)\pi^{(2)}(a^{(2)}|s)\mathbb{E}_{s'\sim\mathcal{P}(\cdot|s,a^{(1)},a^{(2)})}\big[ V^{(\tau)}_{\pi'^{(1)},\pi^{(2)}}(s') - V^{(\tau)}_*(s') \big] \right]
$$

$$
+ \pi'^{(1)}(s)^\top Q^{(\tau)}_*(s)\pi^{(2)}(s) - \pi^{(1)}_{*\tau}(s)^\top Q^{(\tau)}_*(s)\pi^{(2)}_{*\tau}(s)
$$

$$
+ \tau \left[ \mathcal{H}\big(\pi'^{(1)}(s)\big) - \mathcal{H}\big(\pi^{(2)}(s)\big) - \mathcal{H}\big(\pi^{(1)}_{*\tau}(s)\big) + \mathcal{H}\big(\pi^{(2)}_{*\tau}(s)\big) \right]
$$

$$
\leq \gamma \max_{s'} \left[ V^{(\tau)}_{\pi'^{(1)},\pi^{(2)}}(s') - V^{(\tau)}_*(s') \right]
$$

$$
+ \tau f_\tau\big(Q^{(\tau)}_*(s); \pi'^{(1)}(s),\pi^{(2)}(s)\big) - \tau f_\tau\big(Q^{(\tau)}_*(s); \pi^{(1)}_{*\tau}(s),\pi^{(2)}_{*\tau}(s)\big)
$$

$$
\overset{(i)}{\leq} \gamma \max_{s'} \left[ V^{(\tau)}_{\pi'^{(1)},\pi^{(2)}}(s') - V^{(\tau)}_*(s') \right]
$$

$$
+ \tau f_\tau\big(Q^{(\tau)}_*(s); \pi'^{(1)}(s),\pi^{(2)}(s)\big) - \tau \min_{\pi'^{(2)}} f_\tau\big(Q^{(\tau)}_*(s); \pi^{(1)}(s),\pi'^{(2)}(s)\big),
$$

where (i) uses $f_\tau\big(Q^{(\tau)}_*(s); \pi^{(1)}_{*\tau}(s),\pi^{(2)}_{*\tau}(s)\big) = \max_{\pi''^{(1)}} \min_{\pi'^{(2)}} f_\tau\big(Q^{(\tau)}_*(s); \pi''^{(1)}(s),\pi'^{(2)}(s)\big) \geq \min_{\pi'^{(2)}} f_\tau\big(Q^{(\tau)}_*(s); \pi^{(1)}(s),\pi'^{(2)}(s)\big)$.

Applying $\max_{s,\pi'^{(1)}}$ to both sides of the above inequality and rearranging it yields that

$$
\max_{s,\pi'^{(1)}} \left[ V^{(\tau)}_{\pi'^{(1)},\pi^{(2)}}(s) - V^{(\tau)}_*(s) \right]
$$

$$
\leq \frac{1}{1-\gamma} \max_{s,\pi'^{(1)}(s),\pi'^{(2)}(s)} \left[ f_\tau\big[Q^{(\tau)}_*(s),\pi'^{(1)}(s),\pi^{(2)}(s)\big] - f_\tau\big[Q^{(\tau)}_*(s),\pi^{(1)}(s),\pi'^{(2)}(s)\big] \right]. \quad (91)
$$

Similarly, we can obtain that

$$
\max_{s,\pi'^{(2)}} \left[ V^{(\tau)}_*(s) - V^{(\tau)}_{\pi^{(1)},\pi'^{(2)}}(s) \right]
$$

$$
\leq \frac{1}{1-\gamma} \max_{s,\pi'^{(1)}(s),\pi'^{(2)}(s)} \left[ f_\tau\big[Q^{(\tau)}_*(s),\pi'^{(1)}(s),\pi^{(2)}(s)\big] - f_\tau\big[Q^{(\tau)}_*(s),\pi^{(1)}(s),\pi'^{(2)}(s)\big] \right]. \quad (92)
$$

Therefore,

$$
D^{(\tau)}\big(\pi^{(1)},\pi^{(2)}\big)
$$

$$
= \max_{s,\pi'^{(1)},\pi'^{(2)}} \left[ V^{(\tau)}_{\pi'^{(1)},\pi^{(2)}}(s) - V^{(\tau)}_{\pi^{(1)},\pi'^{(2)}}(s) \right]
$$

$$
\leq \max_{s,\pi'^{(1)}} \left[ V^{(\tau)}_{\pi'^{(1)},\pi^{(2)}}(s) - V^{(\tau)}_*(s) \right] + \max_{s,\pi'^{(2)}} \left[ V^{(\tau)}_*(s) - V^{(\tau)}_{\pi^{(1)},\pi'^{(2)}}(s) \right]
$$

$$
\overset{(i)}{\leq} \frac{2}{1-\gamma} \max_{s,\pi'^{(1)}(s),\pi'^{(2)}(s)} \left[ f_\tau\big[Q^{(\tau)}_*(s),\pi'^{(1)}(s),\pi^{(2)}(s)\big] - f_\tau\big[Q^{(\tau)}_*(s),\pi^{(1)}(s),\pi'^{(2)}(s)\big] \right]
$$

where (i) uses eqs. (91)&(92). $\qquad\square$

**Lemma 6.** $D^{(\tau)}\big(\pi^{(1)},\pi^{(2)}\big)$ *is* $\frac{2\ln A_{\max}}{1-\gamma}$-*Lipschitz continuous with regard to* $\tau$.

*Proof.* The definition of the state value function $V_{\pi^{(1)},\pi^{(2)}}^{(\tau)}$ in (3) can be rewritten as follows.

$$V_{\pi^{(1)},\pi^{(2)}}^{(\tau)}(s) = \mathbb{E}\Big[\sum_{t=0}^{\infty}\gamma^t R_t\Big|s_0 = s\Big] + \sum_{t=0}^{\infty}\gamma^t\tau[\mathcal{H}(\pi^{(1)}(s_t)) - \mathcal{H}(\pi^{(2)}(s_t))]. \tag{93}$$

Hence, for any $\tau,\tau' \geq 0$,

$$\big|V_{\pi^{(1)},\pi^{(2)}}^{(\tau')}(s) - V_{\pi^{(1)},\pi^{(2)}}^{(\tau)}(s)\big|$$

$$\leq \sum_{t=0}^{\infty}\gamma^t|\tau' - \tau||\mathcal{H}(\pi^{(1)}(s_t)) - \mathcal{H}(\pi^{(2)}(s_t))|$$

$$\overset{(i)}{\leq} \sum_{t=0}^{\infty}\gamma^t|\tau' - \tau|\ln A_{\max} \leq \frac{\ln A_{\max}}{1 - \gamma}|\tau' - \tau|, \tag{94}$$

where (i) uses $0 \leq \mathcal{H}(\pi^{(m)}(s_t)) \leq \ln|\mathcal{A}^{(m)}| \leq \ln A_{\max}$. Hence, this Lemma can be proved as follows.

$$|D^{(\tau')}(\pi^{(1)},\pi^{(2)}) - D^{(\tau)}(\pi^{(1)},\pi^{(2)})|$$

$$\leq \Big|\max_{s,\pi'^{(1)},\pi'^{(2)}}\big[V_{\pi'^{(1)},\pi^{(2)}}^{(\tau')}(s) - V_{\pi^{(1)},\pi'^{(2)}}^{(\tau')}(s)\big] - \max_{s,\pi'^{(1)},\pi'^{(2)}}\big[V_{\pi'^{(1)},\pi^{(2)}}^{(\tau)}(s) - V_{\pi^{(1)},\pi'^{(2)}}^{(\tau)}(s)\big]\Big|$$

$$\leq \max_{s,\pi'^{(1)},\pi'^{(2)}}\Big|\big[V_{\pi'^{(1)},\pi^{(2)}}^{(\tau')}(s) - V_{\pi^{(1)},\pi'^{(2)}}^{(\tau')}(s)\big] - \big[V_{\pi'^{(1)},\pi^{(2)}}^{(\tau)}(s) - V_{\pi^{(1)},\pi'^{(2)}}^{(\tau)}(s)\big]\Big|$$

$$\leq \max_{s,\pi'^{(1)},\pi'^{(2)}}\Big[\big|V_{\pi'^{(1)},\pi^{(2)}}^{(\tau')}(s) - V_{\pi'^{(1)},\pi^{(2)}}^{(\tau)}(s)\big| + \big|V_{\pi^{(1)},\pi'^{(2)}}^{(\tau)}(s) - V_{\pi^{(1)},\pi'^{(2)}}^{(\tau')}(s)\big|\Big]$$

$$\overset{(i)}{\leq} \frac{2\ln A_{\max}}{1 - \gamma}|\tau' - \tau|,$$

where (i) uses eq. (94). $\qquad\square$

## D  PROOF OF THEOREM 1

**Theorem 1** (Finite-time convergence rate). *Apply Algorithm 1 to solve the entropy-regularized Markov game with $\tau \in (0, 1]$. Choose learning rate $\eta = [2(\tau + Q_{\max})]^{-1}$, initialization $\|\widehat{V}_0\|_\infty \leq V_{\max}$ and batch sizes $N_{k,t}, \overline{N}_{k,t}, N_{k+1}$ that satisfy (18) & (19). Then, the Nash equilibrium duality gap converges at the following rate with probability at least $1 - \delta$.*

$$D^{(\tau)}\big(\pi_{K-1}^{(1)},\pi_{K-1}^{(2)}\big) \leq \mathcal{O}\bigg(\frac{V_{\max}\ln A_{\max}}{1 - \gamma}\sum_{k=0}^{K-1}\gamma^{K-k}(1 - \eta\tau)^{T_k-1}$$

$$+ \frac{V_{\max}}{1 - \gamma}\Big[\frac{t_{mix}A_{\max}}{\mu_{\min}}\ln\Big(\frac{T_{sum}|\mathcal{S}|A_{\max}}{\delta\mu_{\min}}\Big)\Big]^{2/3}\sum_{k=0}^{K-1}\gamma^{K-k-1}\sum_{t=0}^{T_k-1}(1 - \eta\tau)^{T_k-2-t}\Big(\frac{1}{N_{k,t}^{2/3}} + \frac{V_{\max}}{\tau\overline{N}_{k,t+1}^{2/3}}\Big)$$

$$+ \frac{V_{\max}^3\gamma^K}{\tau^2(1 - \gamma)^2} + \frac{t_{mix}V_{\max}^3}{\tau^2\mu_{\min}(1 - \gamma)^3}\ln\Big(\frac{K|\mathcal{S}|}{\delta\mu_{\min}}\Big)\sum_{k=0}^{K-1}\frac{\gamma^{K-k-1}}{N_{k+1}}\bigg). \tag{20}$$

*Proof.* First, consider the SPU iterations that are defined by replacing $Q_{k,t}^{(m)}, \overline{Q}_{k,t+1}^{(m)}$ in (8) with $\widehat{Q}_{k,t}^{(m)}, \widehat{\overline{Q}}_{k,t+1}^{(m)}$, respectively, for $m = 1, 2$. We can apply Lemma 1 with the quantities being specified as follows.

- $Q := Q_k(s), h_\tau(\mu,\nu) := f_\tau\big(Q_k(s);\mu,\nu\big),$

- $\mu_t := \pi_{k,t}^{(1)}(s), \nu_t := \pi_{k,t}^{(2)}(s), \overline{\mu}_{t+1} := \overline{\pi}_{k,t+1}^{(1)}(s), \overline{\nu}_{t+1} := \overline{\pi}_{k,t+1}^{(2)}(s),$
  $\mu_\tau^* := \pi_k^{*(1)}(s), \nu_\tau^* := \pi_k^{*(2)}(s),$

- $d_m := |\mathcal{A}^{(m)}|$,

- $\delta_{k,t}^{(m)} := \widehat{Q}_{k,t}^{(m)}(s) - Q_{k,t}^{(m)}(s), \overline{\delta}_{k,t}^{(m)} := \widehat{\overline{Q}}_{k,t+1}^{(m)}(s) - \overline{Q}_{k,t+1}^{(m)}(s)$,

where $\pi_k^{(1)*}(s), \pi_k^{(2)*}(s)$ is the solution policy pair to the minimax optimization problem $\min_{\pi^{(2)}(s)} \max_{\pi^{(1)}(s)} f_\tau(Q_k(s); \pi^{(1)}(s), \pi^{(2)}(s))$.

Consequently, eqs. (29) & (30) with $t = T_k - 1$ imply the following two inequalities, respectively.

$$
\begin{aligned}
&\left| f_\tau\big(Q_k(s); \overline{\pi}_{k,T_k}^{(1)}(s), \overline{\pi}_{k,T_k}^{(1)}(s)\big) - f_\tau\big(Q_k(s); \pi_k^{*(1)}(s), \pi_k^{*(2)}(s)\big) \right| \\
&= \left| V'_{k+1}(s) - f_\tau\big(Q_k(s); \pi_k^{*(1)}(s), \pi_k^{*(2)}(s)\big) \right| \\
&\leq \frac{8}{\eta}(1 - \eta\tau)^{T_k - 1} \ln A_{\max} + \sum_{t=0}^{T_k - 1} (1 - \eta\tau)^{T_k - 2 - t} \\
&\quad \sum_{m=1}^{2} \left[ 8\eta \big\| \widehat{Q}_{k,t}^{(m)}(s) - Q_{k,t}^{(m)}(s) \big\|_\infty^2 + \frac{33}{4\tau} \big\| \widehat{\overline{Q}}_{k,t+1}^{(m)}(s) - \overline{Q}_{k,t+1}^{(m)}(s) \big\|_\infty^2 \right],
\end{aligned}
\tag{95}
$$

$$
\begin{aligned}
&\max_{\pi^{(1)}(s), \pi^{(2)}(s)} \left[ f_\tau\big(Q_k(s); \pi^{(1)}(s), \overline{\pi}_{k,T_k}^{(2)}(s)\big) - f_\tau\big(Q_k(s); \overline{\pi}_{k,T_k}^{(1)}(s), \pi^{(2)}(s)\big) \right] \\
&= \max_{\pi^{(1)}(s), \pi^{(2)}(s)} \left[ f_\tau\big(Q_k(s); \pi^{(1)}(s), \pi_k^{(2)}(s)\big) - f_\tau\big(Q_k(s); \pi_k^{(1)}(s), \pi^{(2)}(s)\big) \right] \\
&\leq \frac{4}{\eta}(1 - \eta\tau)^{T_k - 1} \ln A_{\max} + \sum_{t=0}^{T_k - 1} (1 - \eta\tau)^{T_k - 2 - t} \\
&\quad \sum_{m=1}^{2} \left[ 16\eta \big\| \widehat{Q}_{k,t}^{(m)}(s) - Q_{k,t}^{(m)}(s) \big\|_\infty^2 + \frac{12}{\tau} \big\| \widehat{\overline{Q}}_{k,t+1}^{(m)}(s) - \overline{Q}_{k,t+1}^{(m)}(s) \big\|_\infty^2 \right].
\end{aligned}
\tag{96}
$$

Then, define the soft Bellman operator $\mathcal{B}_\tau$ as follows.

$$
\begin{aligned}
&\mathcal{B}_\tau(Q)(s, a^{(1)}, a^{(2)}) \\
&:= R(s, a^{(1)}, a^{(2)}) + \gamma \mathbb{E}_{s' \sim \mathcal{P}(\cdot|s, a^{(1)}, a^{(2)})} \left[ \max_{\pi^{(1)}(s')} \min_{\pi^{(2)}(s')} f_\tau\big(Q(s'); \pi^{(1)}(s'), \pi^{(2)}(s')\big) \right],
\end{aligned}
\tag{97}
$$

where $Q : \mathcal{S} \times \mathcal{A}^{(1)} \times \mathcal{A}^{(2)} \to \mathbb{R}$ and $Q(s') \in \mathbb{R}^{|\mathcal{A}^{(1)}| \times |\mathcal{A}^{(2)}|}$ with each entry given by $[Q(s')]_{a^{(1)}, a^{(2)}} := Q(s, a^{(1)}, a^{(2)})$. It can be proved that $\mathcal{B}_\tau$ is a contraction operator and has a unique fixed point $Q_*^{(\tau)}$ (Cen et al., 2021), that is,

$$
\|\mathcal{B}_\tau(Q') - \mathcal{B}_\tau(Q)\|_\infty \leq \gamma \|Q' - Q\|_\infty,
\tag{98}
$$

$$
\mathcal{B}_\tau(Q_*^{(\tau)}) = Q_*^{(\tau)}.
\tag{99}
$$

As a result, we obtain that

$$
\begin{aligned}
&\|Q_{k+1} - Q_*^{(\tau)}\|_\infty \\
&\overset{(i)}{\leq} \|Q_{k+1} - \mathcal{B}_\tau(Q_k)\|_\infty + \|\mathcal{B}_\tau(Q_k) - \mathcal{B}_\tau(Q_*^{(\tau)})\|_\infty \\
&\overset{(ii)}{\leq} \max_{s, a^{(1)}, a^{(2)}} |Q_{k+1}(s, a^{(1)}, a^{(2)}) - \mathcal{B}_\tau(Q_k)(s, a^{(1)}, a^{(2)})| + \gamma \|Q_k - Q_*^{(\tau)}\|_\infty \\
&= \max_{s, a^{(1)}, a^{(2)}} \Big| R(s, a^{(1)}, a^{(2)}) + \gamma \mathbb{E}_{s' \sim \mathcal{P}(\cdot|s, a^{(1)}, a^{(2)})} V'_{k+1}(s') \\
&\quad - \Big[ R(s, a^{(1)}, a^{(2)}) + \gamma \mathbb{E}_{s' \sim \mathcal{P}(\cdot|s, a^{(1)}, a^{(2)})} f_\tau\big(Q_k(s'); \pi_k^{*(1)}(s'), \pi_k^{*(2)}(s')\big) \Big] \Big| + \gamma \|Q_k - Q_*^{(\tau)}\|_\infty \\
&\leq \gamma \max_{s' \in \mathcal{S}} \big| V'_{k+1}(s') - f_\tau\big(Q_k(s'); \pi_k^{*(1)}(s'), \pi_k^{*(2)}(s')\big) \big| + \gamma \|Q_k - Q_*^{(\tau)}\|_\infty,
\end{aligned}
\tag{100}
$$

where (i) uses eq. (99) and (ii) uses eq. (98). Throughout the proof, suppose that eqs. (72), (75) & (76) hold simultaneously which occurs with probability at least $1 - \delta$. In this case, we have

$$
\begin{aligned}
&\left\|\widehat{Q}_{k,t}^{(m)}(s) - Q_{k,t}^{(m)}(s)\right\|_\infty^2 \\
&\overset{(i)}{\leq} 648 Q_{\max}^2 \left[ \frac{t_{\mathrm{mix}} A_{\max}}{N_{k,t} \mu_{\min}} \ln\left( \frac{20 T_{\mathrm{sum}} |\mathcal{S}| A_{\max}}{\delta \sqrt{\mu_{\min}}} \right) \right]^{2/3} + 8 \left\| \widehat{V}_k - V_k' \right\|_\infty^2 \\
&\overset{(ii)}{\leq} 648 Q_{\max}^2 \left[ \frac{t_{\mathrm{mix}} A_{\max}}{N_{k,t} \mu_{\min}} \ln\left( \frac{20 T_{\mathrm{sum}} |\mathcal{S}| A_{\max}}{\delta \sqrt{\mu_{\min}}} \right) \right]^{2/3} \\
&\quad + 32 V_{\max}^2 \gamma^{2k} + \frac{29241 t_{\mathrm{mix}} V_{\max}^2}{N_{k+1} \mu_{\min} (1-\gamma)^2} \ln\left( \frac{4K|\mathcal{S}|}{\delta \sqrt{\mu_{\min}}} \right),
\end{aligned}
\tag{101}
$$

where (i) uses eq. (75) and the inequality that $(a+b)^2 \leq 2a^2 + 2b^2$ for any $a, b \geq 0$, (ii) uses eq. (72) and also $(a+b)^2 \leq 2a^2 + 2b^2$ for any $a, b \geq 0$. Similarly, we obtain that

$$
\begin{aligned}
&\left\| \widehat{\overline{Q}}_{k,t+1}^{(m)}(s) - \overline{Q}_{k,t+1}^{(m)}(s) \right\|_\infty^2 \\
&\leq 648 Q_{\max}^2 \left[ \frac{t_{\mathrm{mix}} A_{\max}}{\overline{N}_{k,t+1} \mu_{\min}} \ln\left( \frac{20 T_{\mathrm{sum}} |\mathcal{S}| A_{\max}}{\delta \sqrt{\mu_{\min}}} \right) \right]^{2/3} \\
&\quad + 32 V_{\max}^2 \gamma^{2k} + \frac{29241 t_{\mathrm{mix}} V_{\max}^2}{N_{k+1} \mu_{\min} (1-\gamma)^2} \ln\left( \frac{4K|\mathcal{S}|}{\delta \sqrt{\mu_{\min}}} \right).
\end{aligned}
\tag{102}
$$

Then, iterating eq. (100) yields that

$$
\begin{aligned}
&\|Q_{K-1} - Q_*^{(\tau)}\|_\infty \\
&\leq \gamma^{K-1} \|Q_0 - Q_*^{(\tau)}\|_\infty + \sum_{k=0}^{K-2} \gamma^{K-k-1} \max_{s' \in \mathcal{S}} \left| V_{k+1}(s') - f_\tau\left( Q_k(s'); \pi_k^{*(1)}(s'), \pi_k^{*(2)}(s') \right) \right| \\
&\overset{(i)}{\leq} 2\gamma^{K-1} Q_{\max} + \sum_{k=0}^{K-2} \gamma^{K-k-1} \left[ \frac{8}{\eta} (1-\eta\tau)^{T_k-1} \ln A_{\max} + \sum_{t=0}^{T_k-1} (1-\eta\tau)^{T_k-2-t} \right. \\
&\quad \left. \sum_{m=1}^2 \left[ 8\eta \left\| \widehat{Q}_{k,t}^{(m)}(s) - Q_{k,t}^{(m)}(s) \right\|_\infty^2 + \frac{33}{4\tau} \left\| \widehat{\overline{Q}}_{k,t+1}^{(m)}(s) - \overline{Q}_{k,t+1}^{(m)}(s) \right\|_\infty^2 \right] \right] \\
&\overset{(ii)}{\leq} 2\gamma^{K-1} Q_{\max} + \sum_{k=0}^{K-2} \gamma^{K-k-1} \left[ \frac{8}{\eta} (1-\eta\tau)^{T_k-1} \ln A_{\max} + \sum_{t=0}^{T_k-1} (1-\eta\tau)^{T_k-2-t} \right. \\
&\quad \sum_{m=1}^2 \left( 648 Q_{\max}^2 \left[ \frac{t_{\mathrm{mix}} A_{\max}}{\mu_{\min}} \ln\left( \frac{20 T_{\mathrm{sum}} |\mathcal{S}| A_{\max}}{\delta \sqrt{\mu_{\min}}} \right) \right]^{2/3} \left( \frac{8\eta}{N_{k,t}^{2/3}} + \frac{33}{4\tau \overline{N}_{k,t+1}^{2/3}} \right) \right. \\
&\quad \left. \left. + \left( 8\eta + \frac{33}{4\tau} \right) \left[ 32 V_{\max}^2 \gamma^{2k} + \frac{29241 t_{\mathrm{mix}} V_{\max}^2}{N_{k+1} \mu_{\min} (1-\gamma)^2} \ln\left( \frac{4K|\mathcal{S}|}{\delta \sqrt{\mu_{\min}}} \right) \right] \right) \right] \\
&\overset{(iii)}{\leq} 2\gamma^{K-1} Q_{\max} + \frac{8}{\eta} \ln A_{\max} \sum_{k=0}^{K-2} \gamma^{K-k-1} (1-\eta\tau)^{T_k-1} \\
&\quad + 324 Q_{\max}^2 \left[ \frac{t_{\mathrm{mix}} A_{\max}}{\mu_{\min}} \ln\left( \frac{20 T_{\mathrm{sum}} |\mathcal{S}| A_{\max}}{\delta \sqrt{\mu_{\min}}} \right) \right]^{2/3} \\
&\quad \sum_{k=0}^{K-2} \gamma^{K-k-1} \sum_{t=0}^{T_k-1} (1-\eta\tau)^{T_k-2-t} \left( \frac{32\eta}{N_{k,t}^{2/3}} + \frac{33}{\tau \overline{N}_{k,t+1}^{2/3}} \right) + \frac{1568 \gamma^{K-1} V_{\max}^2}{\eta \tau^2 (1-\gamma)} \\
&\quad + \frac{1432809 t_{\mathrm{mix}} V_{\max}^2}{\eta \tau^2 \mu_{\min} (1-\gamma)^2} \ln\left( \frac{4K|\mathcal{S}|}{\delta \sqrt{\mu_{\min}}} \right) \sum_{k=0}^{K-2} \gamma^{K-k-1} N_{k+1}^{-1} \\
&\overset{(iv)}{=} \mathcal{O} \left[ V_{\max} \ln A_{\max} \sum_{k=0}^{K-2} \gamma^{K-k} (1-\eta\tau)^{T_k-1} + V_{\max} \left[ \frac{t_{\mathrm{mix}} A_{\max}}{\mu_{\min}} \ln\left( \frac{T_{\mathrm{sum}} |\mathcal{S}| A_{\max}}{\delta \mu_{\min}} \right) \right]^{2/3} \right.
\end{aligned}
\tag{103}
$$

$$\sum_{k=0}^{K-2} \gamma^{K-k-1} \sum_{t=0}^{T_k-1} (1-\eta\tau)^{T_k-2-t} \Big( \frac{1}{N_{k,t}^{2/3}} + \frac{V_{\max}}{\tau \overline{N}_{k,t+1}^{2/3}} \Big)$$

$$+ \frac{V_{\max}^3 \gamma^K}{\tau^2 (1-\gamma)} + \frac{t_{\mathrm{mix}} V_{\max}^3}{\tau^2 \mu_{\min} (1-\gamma)^2} \ln \Big( \frac{K|\mathcal{S}|}{\delta \mu_{\min}} \Big) \sum_{k=0}^{K-2} \gamma^{K-k-1} N_{k+1}^{-1} \Bigg] \tag{104}$$

where (i) uses eqs. (68)&(95), (ii) uses eqs. (101)&(102), (iii) uses $\eta = [2(\tau + Q_{\max})]^{-1} \le 1/(2\tau)$, $\sum_{t=0}^{T_k-1}(1-\eta\tau)^{T_k-2-t} \le \frac{1}{\eta\tau(1-\eta\tau)} \le \frac{2}{\eta\tau}$ and $\sum_{k=0}^{K-2} \gamma^{K+k-1} \le \frac{\gamma^{K-1}}{1-\gamma}$, and (iv) uses $Q_{\max} = \mathcal{O}(V_{\max})$ for $\gamma \approx 1$ and $\eta = [2(\tau + Q_{\max})]^{-1} = \mathcal{O}(Q_{\max}^{-1}) = \mathcal{O}(V_{\max}^{-1})$ (Since $Q_{\max} \ge \gamma\tau = \mathcal{O}(\tau)$).

Using Lemma 5, the convergence rate of the duality gap in eq. (20) can be proved as follows.

$$D^{(\tau)}\big(\pi_{K-1}^{(1)}, \pi_{K-1}^{(2)}\big)$$

$$\overset{(i)}{\le} \frac{2}{1-\gamma} \max_{s,\pi^{(1)},\pi^{(2)}} \Big[ f_\tau\big[Q_*^{(\tau)}(s), \pi^{(1)}(s), \pi_{K-1}^{(2)}(s)\big] - f_\tau\big[Q_*^{(\tau)}(s), \pi_{K-1}^{(1)}(s), \pi^{(2)}(s)\big] \Big]$$

$$\le \frac{2}{1-\gamma} \max_{s,\pi^{(1)},\pi^{(2)}} \Big[ f_\tau\big(Q_{K-1}(s); \pi^{(1)}(s), \pi_{K-1}^{(2)}(s)\big) - f_\tau\big(Q_{K-1}(s); \pi_{K-1}^{(1)}(s), \pi^{(2)}(s)\big) \Big]$$

$$+ \frac{2}{1-\gamma} \max_{s,\pi^{(1)},\pi^{(2)}} \big| f_\tau\big(Q_*^{(\tau)}(s); \pi^{(1)}(s), \pi_{K-1}^{(2)}(s)\big) - f_\tau\big(Q_{K-1}(s); \pi^{(1)}(s), \pi_{K-1}^{(2)}(s)\big) \big|$$

$$+ \frac{2}{1-\gamma} \max_{s,\pi^{(1)},\pi^{(2)}} \big| f_\tau\big(Q_{K-1}(s); \pi_{K-1}^{(1)}(s), \pi^{(2)}(s)\big) - f_\tau\big(Q_*^{(\tau)}(s); \pi_{K-1}^{(1)}(s), \pi^{(2)}(s)\big) \big|$$

$$\overset{(ii)}{\le} \frac{8}{\eta(1-\gamma)} (1-\eta\tau)^{T_{K-1}-1} \ln A_{\max} + \frac{2}{1-\gamma} \sum_{t=0}^{T_{K-1}-1} (1-\eta\tau)^{T_{K-1}-2-t}$$

$$\sum_{m=1}^{2} \Big[ 16\eta \big\| \widehat{Q}_{K-1,t}^{(m)}(s) - Q_{K-1,t}^{(m)}(s) \big\|_\infty^2 + \frac{12}{\tau} \big\| \widehat{\overline{Q}}_{K-1,t+1}^{(m)}(s) - \overline{Q}_{K-1,t+1}^{(m)}(s) \big\|_\infty^2 \Big]$$

$$+ \frac{2}{1-\gamma} \max_{s,\pi^{(1)}} \big| \pi^{(1)}(s)^\top [Q_*^{(\tau)}(s) - Q_{K-1}(s)] \pi_{K-1}^{(2)}(s) \big|$$

$$+ \frac{2}{1-\gamma} \max_{s,\pi^{(2)}} \big| \pi_{K-1}^{(1)}(s)^\top [Q_{K-1}(s) - Q_*^{(\tau)}(s)] \pi^{(2)}(s) \big|$$

$$\overset{(iii)}{\le} \frac{8 \ln A_{\max}}{\eta(1-\gamma)} (1-\eta\tau)^{T_{K-1}-1} + \frac{4}{1-\gamma} \sum_{t=0}^{T_{K-1}-1} (1-\eta\tau)^{T_{K-1}-2-t}$$

$$\Big( 648 Q_{\max}^2 \Big[ \frac{t_{\mathrm{mix}} A_{\max}}{\mu_{\min}} \ln\Big( \frac{20 T_{\mathrm{sum}}|\mathcal{S}|A_{\max}}{\delta\sqrt{\mu_{\min}}} \Big) \Big]^{2/3} \Big( \frac{16\eta}{N_{K-1,t}^{2/3}} + \frac{12}{\tau \overline{N}_{K-1,t+1}^{2/3}} \Big)$$

$$+ \Big( 32\eta + \frac{24}{\tau} \Big) \Big[ 32 V_{\max}^2 \gamma^{2(K-1)} + \frac{29241 t_{\mathrm{mix}} V_{\max}^2}{N_K \mu_{\min} (1-\gamma)^2} \ln\Big( \frac{4K|\mathcal{S}|}{\delta\sqrt{\mu_{\min}}} \Big) \Big] \Big)$$

$$+ \frac{4}{1-\gamma} \big\| Q_{K-1} - Q_*^{(\tau)} \big\|_\infty$$

$$\overset{(iv)}{\le} \frac{8 \ln A_{\max}}{\eta(1-\gamma)} (1-\eta\tau)^{T_{K-1}-1} + \frac{2592 Q_{\max}^2}{1-\gamma} \Big[ \frac{t_{\mathrm{mix}} A_{\max}}{\mu_{\min}} \ln\Big( \frac{20 T_{\mathrm{sum}}|\mathcal{S}|A_{\max}}{\delta\sqrt{\mu_{\min}}} \Big) \Big]^{2/3}$$

$$\sum_{t=0}^{T_{K-1}-1} (1-\eta\tau)^{T_{K-1}-2-t} \Big( \frac{16\eta}{N_{K-1,t}^{2/3}} + \frac{12}{\tau \overline{N}_{K-1,t+1}^{2/3}} \Big)$$

$$+ \frac{160}{\eta\tau^2} \Big[ \frac{32 V_{\max}^2}{1-\gamma} \gamma^{2(K-1)} + \frac{29241 t_{\mathrm{mix}} V_{\max}^2}{N_K \mu_{\min} (1-\gamma)^3} \ln\Big( \frac{4K|\mathcal{S}|}{\delta\sqrt{\mu_{\min}}} \Big) \Big]$$

$$+ \frac{8\gamma^{K-1} Q_{\max}}{1-\gamma} + \frac{32 \ln A_{\max}}{\eta(1-\gamma)} \sum_{k=0}^{K-2} \gamma^{K-k-1} (1-\eta\tau)^{T_k-1}$$

$$+ \frac{1296 Q_{\max}^2}{1 - \gamma} \left[ \frac{t_{\mathrm{mix}} A_{\max}}{\mu_{\min}} \ln \left( \frac{20 T_{\mathrm{sum}} |\mathcal{S}| A_{\max}}{\delta \sqrt{\mu_{\min}}} \right) \right]^{2/3}$$

$$\sum_{k=0}^{K-2} \gamma^{K-k-1} \sum_{t=0}^{T_k-1} (1 - \eta\tau)^{T_k-2-t} \left( \frac{32\eta}{N_{k,t}^{2/3}} + \frac{33}{\tau \overline{N}_{k,t+1}^{2/3}} \right) + \frac{6272 \gamma^{K-1} V_{\max}^2}{\eta\tau^2 (1 - \gamma)^2}$$

$$+ \frac{5731236 t_{\mathrm{mix}} V_{\max}^2}{\eta\tau^2 \mu_{\min} (1 - \gamma)^3} \ln \left( \frac{4K |\mathcal{S}|}{\delta \sqrt{\mu_{\min}}} \right) \sum_{k=0}^{K-2} \gamma^{K-k-1} N_{k+1}^{-1}$$

$$\leq \frac{8 \gamma^{K-1} Q_{\max}}{1 - \gamma} + \frac{32 \ln A_{\max}}{\eta(1 - \gamma)} \sum_{k=0}^{K-1} \gamma^{K-k-1} (1 - \eta\tau)^{T_k-1}$$

$$+ \frac{1296 Q_{\max}^2}{1 - \gamma} \left[ \frac{t_{\mathrm{mix}} A_{\max}}{\mu_{\min}} \ln \left( \frac{20 T_{\mathrm{sum}} |\mathcal{S}| A_{\max}}{\delta \sqrt{\mu_{\min}}} \right) \right]^{2/3}$$

$$\sum_{k=0}^{K-1} \gamma^{K-k-1} \sum_{t=0}^{T_k-1} (1 - \eta\tau)^{T_k-2-t} \left( \frac{32\eta}{N_{k,t}^{2/3}} + \frac{33}{\tau \overline{N}_{k,t+1}^{2/3}} \right) + \frac{11392 \gamma^{K-1} V_{\max}^2}{\eta\tau^2 (1 - \gamma)^2}$$

$$+ \frac{5731236 t_{\mathrm{mix}} V_{\max}^2}{\eta\tau^2 \mu_{\min} (1 - \gamma)^3} \ln \left( \frac{4K |\mathcal{S}|}{\delta \sqrt{\mu_{\min}}} \right) \sum_{k=0}^{K-1} \gamma^{K-k-1} N_{k+1}^{-1}$$

$$\overset{(v)}{=} \mathcal{O} \left[ \frac{V_{\max} \ln A_{\max}}{1 - \gamma} \sum_{k=0}^{K-1} \gamma^{K-k} (1 - \eta\tau)^{T_k-1} + \frac{V_{\max}}{1 - \gamma} \left[ \frac{t_{\mathrm{mix}} A_{\max}}{\mu_{\min}} \ln \left( \frac{T_{\mathrm{sum}} |\mathcal{S}| A_{\max}}{\delta \mu_{\min}} \right) \right]^{2/3} \right.$$

$$\sum_{k=0}^{K-1} \gamma^{K-k-1} \sum_{t=0}^{T_k-1} (1 - \eta\tau)^{T_k-2-t} \left( \frac{1}{N_{k,t}^{2/3}} + \frac{V_{\max}}{\tau \overline{N}_{k,t+1}^{2/3}} \right) + \frac{V_{\max}^3 \gamma^K}{\tau^2 (1 - \gamma)^2}$$

$$\left. + \frac{t_{\mathrm{mix}} V_{\max}^3}{\tau^2 \mu_{\min} (1 - \gamma)^3} \ln \left( \frac{K |\mathcal{S}|}{\delta \mu_{\min}} \right) \sum_{k=0}^{K-1} \gamma^{K-k-1} N_{k+1}^{-1} \right],$$

where (i) uses Lemma 5, (ii) uses eq. (96) and the definition of the function $f_\tau$, (iii) uses eqs. (101)&(102), (iv) uses eq. (103) and $\eta = [2(\tau + Q_{\max})]^{-1} \leq 1/(2\tau)$, $\sum_{t=0}^{T_k-1} (1 - \eta\tau)^{T_k-2-t} \leq \frac{1}{\eta\tau(1-\eta\tau)} \leq \frac{2}{\eta\tau}$, and (v) uses $Q_{\max} = \mathcal{O}(V_{\max})$ for $\gamma \approx 1$ and $\eta = [2(\tau + Q_{\max})]^{-1} = \mathcal{O}(Q_{\max}^{-1}) = \mathcal{O}(V_{\max}^{-1})$ (Since $Q_{\max} \geq \gamma\tau = \mathcal{O}(\tau)$).

$\square$

# E  PROOF OF THEOREM 2

**Theorem 2** (Sample complexity). *Implement Algorithm 1 with $\eta = \mathcal{O}(1 - \gamma)$, $\tau = \mathcal{O}\left( \frac{\epsilon(1-\gamma)}{\ln A_{\max}} \right)$, $K = \mathcal{O}\left[ \frac{1}{1-\gamma} \ln \left( \frac{\ln A_{\max}}{\epsilon(1-\gamma)} \right) \right]$ and $T_k = 1 + \frac{k \ln \gamma^{-1}}{\ln(1-\eta\tau)^{-1}}$. Choose the following adaptive batch sizes.*

$$N_{k+1} = \widetilde{\mathcal{O}} \left( \frac{t_{mix} (\ln^2 A_{\max}) \gamma^{-\frac{k}{2}}}{\epsilon^2 \mu_{\min} (1 - \gamma)^8} \right), \quad N_{k,t} = \overline{N}_{k,t} \epsilon^{\frac{3}{2}} (1 - \gamma)^3 = \widetilde{\mathcal{O}} \left( \frac{t_{mix} A_{\max} (1 - \eta\tau)^{\frac{-3(t+1)}{5}}}{\mu_{\min} (1 - \gamma)^3} \right).$$

*Then, for any $\epsilon \leq \frac{\ln A_{\max}}{1 - \gamma}$, the overall sample complexity to achieve $D^{(0)}(\pi_{K-1}^{(1)}, \pi_{K-1}^{(2)}) \leq \epsilon$ is $\widetilde{\mathcal{O}} \left( \frac{t_{mix} A_{\max}}{\mu_{\min} \epsilon^{5.5} (1-\gamma)^{13.5}} \right)$. Please refer to (118) in Appendix E for a complete expression.*

*Proof.* Since $\tau = \mathcal{O}\left( \frac{\epsilon(1-\gamma)}{\ln A_{\max}} \right)$, we have

$$V_{\max} = \frac{1 + \tau \ln A_{\max}}{1 - \gamma} = \frac{1}{1 - \gamma} + \mathcal{O}(\epsilon) = \mathcal{O}\left( (1 - \gamma)^{-1} \right). \tag{105}$$

$$Q_{\max} = \frac{1 + \gamma\tau \ln A_{\max}}{1 - \gamma} = \frac{1}{1 - \gamma} + \mathcal{O}(\epsilon) = \mathcal{O}\left( (1 - \gamma)^{-1} \right). \tag{106}$$

Since $T_k = 1 + \frac{k \ln \gamma^{-1}}{\ln(1 - \eta\tau)^{-1}}$, we have

$$
\begin{aligned}
T_{\text{sum}} &= \sum_{k=0}^{K-1} \left( 1 + \frac{k \ln \gamma^{-1}}{\ln(1 - \eta\tau)^{-1}} \right) \\
&= K + \frac{K(K-1)}{2} \mathcal{O}\left( \frac{1-\gamma}{\eta\tau} \right) \\
&= K + \frac{K(K-1)}{2} \mathcal{O}\left( \frac{\ln A_{\max}}{\epsilon(1-\gamma)} \right) \\
&= \mathcal{O}\left( \frac{K^2 \ln A_{\max}}{\epsilon(1-\gamma)} \right) \\
&= \mathcal{O}\left[ \frac{\ln A_{\max}}{\epsilon(1-\gamma)^3} \ln^2 \left( \frac{\ln A_{\max}}{\epsilon(1-\gamma)} \right) \right].
\end{aligned}
\tag{107}
$$

Hence,

$$
\begin{aligned}
\ln \left( \frac{T_{\text{sum}}|\mathcal{S}|A_{\max}}{\delta\mu_{\min}} \right) &= \mathcal{O}\left[ \ln \left( \frac{|\mathcal{S}|A_{\max} \ln A_{\max}}{\delta\mu_{\min}\epsilon(1-\gamma)^3} \right) + \ln \left( 2\ln \left( \frac{\ln A_{\max}}{\epsilon(1-\gamma)} \right) \right) \right] \\
&= \mathcal{O}\left[ \ln \left( \frac{|\mathcal{S}|A_{\max}}{\epsilon\delta\mu_{\min}(1-\gamma)} \right) \right].
\end{aligned}
\tag{108}
$$

Similarly,

$$
\begin{aligned}
\ln \left( \frac{K|\mathcal{S}|}{\delta\mu_{\min}} \right) &= \mathcal{O}\left[ \ln \left( \left( \frac{|\mathcal{S}|}{\delta\mu_{\min}(1-\gamma)} \right) \ln \left( \frac{\ln A_{\max}}{\epsilon(1-\gamma)} \right) \right) \right] \\
&= \mathcal{O}\left[ \ln \left( \frac{|\mathcal{S}| \ln(\epsilon^{-1} \ln A_{\max})}{\delta\mu_{\min}(1-\gamma)} \right) \right] + \mathcal{O}\left[ \ln \left( \left( \frac{|\mathcal{S}|}{\delta\mu_{\min}(1-\gamma)} \right) \ln \left( \frac{1}{1-\gamma} \right) \right) \right] \\
&= \mathcal{O}\left[ \ln \left( \frac{|\mathcal{S}| \ln(\epsilon^{-1} \ln A_{\max})}{\delta\mu_{\min}(1-\gamma)} \right) \right] + \mathcal{O}\left[ \ln \left( \frac{|\mathcal{S}|}{\delta\mu_{\min}(1-\gamma)} \right) \right] \\
&= \mathcal{O}\left[ \ln \left( \frac{|\mathcal{S}| \ln(\epsilon^{-1} \ln A_{\max})}{\delta\mu_{\min}(1-\gamma)} \right) \right]
\end{aligned}
\tag{109}
$$

where (i) uses $\epsilon \leq \frac{\ln A_{\max}}{1-\gamma}$. With the above equalities, we will prove below that the hyperparameter choices in Theorem 2 are valid and satisfy the conditions of Theorem 1 with proper constants hidden in $\mathcal{O}(\cdot)$.

$$
\eta = [2(\tau + Q_{\max})]^{-1} = \mathcal{O}(1-\gamma)
\tag{110}
$$

$$
\tau = \mathcal{O}\left( \frac{\epsilon(1-\gamma)}{\ln A_{\max}} \right) \in (0, 1]
\tag{111}
$$

$$
K = \frac{1}{\ln \gamma^{-1}} \ln \left( \frac{\ln^{5/2} A_{\max}}{\epsilon^5(1-\gamma)^{7.5}} \right) = \mathcal{O}\left[ \frac{1}{1-\gamma} \ln \left( \frac{\ln A_{\max}}{\epsilon(1-\gamma)} \right) \right] \geq 1
\tag{112}
$$

$$
T_k = 1 + \frac{k \ln \gamma^{-1}}{\ln(1-\eta\tau)^{-1}} \geq 1
\tag{113}
$$

$$
\begin{aligned}
N_{k+1} &= \mathcal{O}\left[ \frac{t_{\text{mix}}(\ln^2 A_{\max})\gamma^{-k/2}}{\epsilon^2\mu_{\min}(1-\gamma)^8} \ln \left( \frac{|\mathcal{S}| \ln(\epsilon^{-1} \ln A_{\max})}{\delta\mu_{\min}(1-\gamma)} \right) \right] \\
&\geq \frac{650 t_{\text{mix}}}{\mu_{\min}} \ln \left( \frac{4}{\delta\sqrt{\mu_{\min}}} \right) = \mathcal{O}\left[ \frac{t_{\text{mix}}}{\mu_{\min}} \ln \left( \frac{1}{\delta\mu_{\min}} \right) \right]
\end{aligned}
\tag{114}
$$

$$
\begin{aligned}
N_{k,t} &= \mathcal{O}\left[ \frac{t_{\text{mix}}A_{\max}(1-\eta\tau)^{-3(t+1)/5}}{\mu_{\min}(1-\gamma)^3} \ln \left( \frac{|\mathcal{S}|A_{\max}}{\epsilon\delta\mu_{\min}(1-\gamma)} \right) \right] \\
&\geq \frac{650 t_{\text{mix}}A_{\max}}{\mu_{\min}} \ln \left( \frac{20 T_{\text{sum}}|\mathcal{S}|A_{\max}}{\delta\sqrt{\mu_{\min}}} \right) = \mathcal{O}\left[ \frac{t_{\text{mix}}A_{\max}}{\mu_{\min}} \ln \left( \frac{|\mathcal{S}|A_{\max}}{\epsilon\delta\mu_{\min}(1-\gamma)} \right) \right]
\end{aligned}
\tag{115}
$$

$$
\overline{N}_{k,t} = \mathcal{O}\left[ \frac{t_{\text{mix}}A_{\max}(\ln^{3/2} A_{\max})(1-\eta\tau)^{-3(t+1)/5}}{\mu_{\min}\epsilon^{3/2}(1-\gamma)^6} \ln \left( \frac{|\mathcal{S}|A_{\max}}{\epsilon\delta\mu_{\min}(1-\gamma)} \right) \right]
$$

$$\geq \frac{650 t_{\mathrm{mix}} A_{\max}}{\mu_{\min}} \ln\Big(\frac{20 T_{\mathrm{sum}} |\mathcal{S}| A_{\max}}{\delta \sqrt{\mu_{\min}}}\Big) = \mathcal{O}\Big[\frac{t_{\mathrm{mix}} A_{\max}}{\mu_{\min}} \ln\Big(\frac{|\mathcal{S}| A_{\max}}{\epsilon \delta \mu_{\min}(1-\gamma)}\Big)\Big], \qquad (116)$$

With these hyperparameters and eqs. (108)&(109), the duality gap bound (20) can be simplified as follows,

$$D^{(\tau)}\big(\pi_{K-1}^{(1)}, \pi_{K-1}^{(2)}\big)$$

$$\leq \mathcal{O}\Big[\frac{V_{\max} \ln A_{\max}}{1-\gamma} \sum_{k=0}^{K-1} \gamma^{K-k}(1-\eta\tau)^{T_k-1} + \frac{V_{\max}}{1-\gamma}\Big[\frac{t_{\mathrm{mix}} A_{\max}}{\mu_{\min}} \ln\Big(\frac{T_{\mathrm{sum}} |\mathcal{S}| A_{\max}}{\delta \mu_{\min}}\Big)\Big]^{2/3}$$

$$\sum_{k=0}^{K-1} \gamma^{K-k-1} \sum_{t=0}^{T_k-1} (1-\eta\tau)^{T_k-2-t}\Big(\frac{1}{N_{k,t}^{2/3}} + \frac{V_{\max}}{\tau \overline{N}_{k,t+1}^{2/3}}\Big) + \frac{V_{\max}^3 \gamma^K}{\tau^2(1-\gamma)^2}$$

$$+ \frac{t_{\mathrm{mix}} V_{\max}^3}{\tau^2 \mu_{\min}(1-\gamma)^3} \ln\Big(\frac{K|\mathcal{S}|}{\delta \mu_{\min}}\Big) \sum_{k=0}^{K-1} \frac{\gamma^{K-k-1}}{N_{k+1}}\Big]$$

$$\overset{(i)}{\leq} \mathcal{O}\Big[\frac{\ln A_{\max}}{(1-\gamma)^2} \sum_{k=0}^{K-1} \gamma^{K-k}\gamma^k$$

$$+ \frac{1}{(1-\gamma)^2}\Big[\frac{t_{\mathrm{mix}} A_{\max}}{\mu_{\min}} \ln\Big(\frac{|\mathcal{S}| A_{\max}}{\epsilon \delta \mu_{\min}(1-\gamma)}\Big)\Big]^{2/3} \sum_{k=0}^{K-1} \gamma^{K-k-1} \sum_{t=0}^{T_k-1} (1-\eta\tau)^{T_k-2-t}$$

$$\Big(\mathcal{O}\Big[\frac{t_{\mathrm{mix}} A_{\max}}{\mu_{\min}(1-\gamma)^3} \ln\Big(\frac{|\mathcal{S}| A_{\max}}{\epsilon \delta \mu_{\min}(1-\gamma)}\Big)\Big]^{-2/3} (1-\eta\tau)^{2(t+1)/5}$$

$$+ \mathcal{O}\Big(\frac{\ln A_{\max}}{\epsilon(1-\gamma)^2}\Big) \mathcal{O}\Big[\frac{t_{\mathrm{mix}} A_{\max}(\ln^{3/2} A_{\max})}{\mu_{\min} \epsilon^{3/2}(1-\gamma)^6} \ln\Big(\frac{|\mathcal{S}| A_{\max}}{\epsilon \delta \mu_{\min}(1-\gamma)}\Big)\Big]^{-2/3} (1-\eta\tau)^{2(t+1)/5}\Big)$$

$$+ \frac{\gamma^K (\ln^2 A_{\max})}{\epsilon^2(1-\gamma)^7} + \frac{t_{\mathrm{mix}}(\ln^2 A_{\max})}{\epsilon^2 \mu_{\min}(1-\gamma)^8} \ln\Big(\frac{|\mathcal{S}| \ln(\epsilon^{-1} \ln A_{\max})}{\delta \mu_{\min}(1-\gamma)}\Big)$$

$$\sum_{k=0}^{K-1} \gamma^{K-k-1} \mathcal{O}\Big[\frac{t_{\mathrm{mix}}(\ln^2 A_{\max})\gamma^{-k/2}}{\epsilon^2 \mu_{\min}(1-\gamma)^8} \ln\Big(\frac{|\mathcal{S}| \ln(\epsilon^{-1} \ln A_{\max})}{\delta \mu_{\min}(1-\gamma)}\Big)\Big]^{-1}\Big]$$

$$\leq \mathcal{O}\Big[\frac{\ln A_{\max}}{(1-\gamma)^2} K\gamma^K + \sum_{k=0}^{K-1} \gamma^{K-k-1} \sum_{t=0}^{T_k-1} (1-\eta\tau)^{T_k-1-3(t+1)/5}$$

$$+ \frac{\gamma^K (\ln^2 A_{\max})}{\epsilon^2(1-\gamma)^7} + \sum_{k=0}^{K-1} \gamma^{K-k/2-1}\Big]$$

$$\overset{(ii)}{\leq} \mathcal{O}\Big[\frac{\ln A_{\max}}{(1-\gamma)^2} K\gamma^K + \sum_{k=0}^{K-1} \gamma^{K-1} \frac{(1-\eta\tau)^{-3/5}[(1-\eta\tau)^{-3T_k/5}-1]}{(1-\eta\tau)^{-3/5}-1}$$

$$+ \frac{\gamma^K (\ln^2 A_{\max})}{\epsilon^2(1-\gamma)^7} + \gamma^{K-1} \frac{\gamma^{-K/2}-1}{\gamma^{-1/2}-1}\Big]$$

$$\overset{(iii)}{\leq} \mathcal{O}\Big[\frac{\ln A_{\max}}{(1-\gamma)^2} K\gamma^K + \sum_{k=0}^{K-1} \frac{\gamma^{K-1-3k/5}}{1-(1-\eta\tau)^{3/5}} + \frac{\gamma^K (\ln^2 A_{\max})}{\epsilon^2(1-\gamma)^7} + \frac{\gamma^{(K-1)/2}}{1-\gamma^{1/2}}\Big]$$

$$\overset{(iv)}{\leq} \mathcal{O}\Big[\frac{\ln A_{\max}}{(1-\gamma)^2} K\gamma^K + \frac{\gamma^{K-1}(\gamma^{-3K/5}-1)}{\eta\tau(\gamma^{-3/5}-1)} + \frac{\gamma^K (\ln^2 A_{\max})}{\epsilon^2(1-\gamma)^7} + \frac{\gamma^{K/2}}{1-\gamma}\Big]$$

$$\overset{(v)}{\leq} \mathcal{O}\Big[\frac{\gamma^K \ln A_{\max}}{(1-\gamma)^3} \ln\Big(\frac{\ln A_{\max}}{\epsilon(1-\gamma)}\Big) + \frac{\gamma^{2K/5} \ln A_{\max}}{\epsilon(1-\gamma)^3} + \frac{\gamma^K (\ln^2 A_{\max})}{\epsilon^2(1-\gamma)^7} + \frac{\gamma^{K/2}}{1-\gamma}\Big]$$

$$\overset{(vi)}{\leq} \mathcal{O}\Big[\epsilon + \epsilon^3 \sqrt{\frac{1-\gamma}{\ln A_{\max}}} + \frac{\epsilon^{5/2}(1-\gamma)^{2.75}}{\ln^{5/4} A_{\max}}\Big] = \mathcal{O}(\epsilon) \overset{(vii)}{\leq} c\epsilon \qquad (117)$$

where (i), (ii) and (iii) use $(1-\eta\tau)^{T_k-1} = \gamma^k$ based on eq. (113), (i) also uses eqs. (105), (108)-(111) & (114)-(116), (iv) uses $1 - (1-\eta\tau)^{3/5} = \mathcal{O}(\eta\tau)$, $1 - \gamma^{1/2} = \mathcal{O}(1-\gamma)$, (v) uses eqs. (110)-(112), (vi) uses $\mathcal{O}\left[\frac{\gamma^K \ln A_{\max}}{(1-\gamma)^3} \ln\left(\frac{\ln A_{\max}}{\epsilon(1-\gamma)}\right)\right] \le \mathcal{O}\left[\frac{\gamma^K (\ln^2 A_{\max})}{\epsilon^2(1-\gamma)^7}\right]$ and $\gamma^K = \frac{\epsilon^5(1-\gamma)^{7.5}}{\ln^{5/2} A_{\max}}$ implied by eq. (112), and the numeric constant $c > 0$ in (vii) exists. By replacing $\epsilon$ with $\epsilon/c$, we obtain that $D_\tau\left(\pi_{K-1}^{(1)}, \pi_{K-1}^{(2)}\right) \le \epsilon$ which means $(\pi_{K-1}^{(1)}, \pi_{K-1}^{(2)})$ is an $\epsilon$-Nash equilibrium policy pair, and the orders of all the hyperparameter choices remain the same. In this case, the overall sample complexity is given by

$$\sum_{k=0}^{K-1}\left[N_{k+1} + 2\sum_{t=0}^{T_k-1}(N_{k,t} + \overline{N}_{k,t+1})\right]$$

$$\overset{(i)}{=} \mathcal{O}\left[\frac{t_{\mathrm{mix}}(\ln^2 A_{\max})}{\epsilon^2 \mu_{\min}(1-\gamma)^8} \ln\left(\frac{|\mathcal{S}|\ln(\epsilon^{-1}\ln A_{\max})}{\delta\mu_{\min}(1-\gamma)}\right)\right]\sum_{k=0}^{K-1}\gamma^{-k/2}$$

$$+ \mathcal{O}\left[\frac{t_{\mathrm{mix}}A_{\max}(\ln^{3/2} A_{\max})}{\mu_{\min}\epsilon^{3/2}(1-\gamma)^6} \ln\left(\frac{|\mathcal{S}|A_{\max}}{\epsilon\delta\mu_{\min}(1-\gamma)}\right)\right]\sum_{k=0}^{K-1}\sum_{t=0}^{T_k-1}(1-\eta\tau)^{-3(t+1)/5}$$

$$= \mathcal{O}\left[\frac{t_{\mathrm{mix}}(\ln^2 A_{\max})}{\epsilon^2 \mu_{\min}(1-\gamma)^8} \ln\left(\frac{|\mathcal{S}|\ln(\epsilon^{-1}\ln A_{\max})}{\delta\mu_{\min}(1-\gamma)}\right)\right]\frac{\gamma^{-K/2}-1}{\gamma^{-1/2}-1}$$

$$+ \mathcal{O}\left[\frac{t_{\mathrm{mix}}A_{\max}(\ln^{3/2} A_{\max})}{\mu_{\min}\epsilon^{3/2}(1-\gamma)^6} \ln\left(\frac{|\mathcal{S}|A_{\max}}{\epsilon\delta\mu_{\min}(1-\gamma)}\right)\right]\sum_{k=0}^{K-1}\frac{[(1-\eta\tau)^{-3T_k/5}-1]}{1-(1-\eta\tau)^{3/5}}$$

$$\overset{(ii)}{\le} \mathcal{O}\left[\frac{t_{\mathrm{mix}}(\ln^2 A_{\max})}{\epsilon^2 \mu_{\min}(1-\gamma)^8} \ln\left(\frac{|\mathcal{S}|\ln(\epsilon^{-1}\ln A_{\max})}{\delta\mu_{\min}(1-\gamma)}\right)\right]\gamma^{-K/2}$$

$$+ \mathcal{O}\left[\frac{t_{\mathrm{mix}}A_{\max}(\ln^{3/2} A_{\max})}{\mu_{\min}\epsilon^{3/2}(1-\gamma)^6} \ln\left(\frac{|\mathcal{S}|A_{\max}}{\epsilon\delta\mu_{\min}(1-\gamma)}\right)\frac{\ln A_{\max}}{\epsilon(1-\gamma)^2}\right]\sum_{k=0}^{K-1}\gamma^{-3k/5}$$

$$\le \mathcal{O}\left[\frac{t_{\mathrm{mix}}(\ln^2 A_{\max})}{\epsilon^2 \mu_{\min}(1-\gamma)^8} \ln\left(\frac{|\mathcal{S}|\ln(\epsilon^{-1}\ln A_{\max})}{\delta\mu_{\min}(1-\gamma)}\right)\right]\gamma^{-K/2}$$

$$+ \mathcal{O}\left[\frac{t_{\mathrm{mix}}A_{\max}(\ln^{5/2} A_{\max})}{\mu_{\min}\epsilon^{5/2}(1-\gamma)^8} \ln\left(\frac{|\mathcal{S}|A_{\max}}{\epsilon\delta\mu_{\min}(1-\gamma)}\right)\right]\frac{\gamma^{-3K/5}}{\gamma^{-3/5}-1}$$

$$\overset{(iii)}{\le} \mathcal{O}\left[\frac{t_{\mathrm{mix}}(\ln^2 A_{\max})}{\epsilon^2 \mu_{\min}(1-\gamma)^8} \ln\left(\frac{|\mathcal{S}|\ln(\epsilon^{-1}\ln A_{\max})}{\delta\mu_{\min}(1-\gamma)}\right)\right]\frac{\ln^{5/4} A_{\max}}{\epsilon^{5/2}(1-\gamma)^{3.75}}$$

$$+ \mathcal{O}\left[\frac{t_{\mathrm{mix}}A_{\max}(\ln^{5/2} A_{\max})}{\mu_{\min}\epsilon^{5/2}(1-\gamma)^8} \ln\left(\frac{|\mathcal{S}|A_{\max}}{\epsilon\delta\mu_{\min}(1-\gamma)}\right)\right]\frac{\ln^{3/2} A_{\max}}{\epsilon^3(1-\gamma)^{5.5}}$$

$$\overset{(iv)}{=} \mathcal{O}\left[\frac{t_{\mathrm{mix}}A_{\max}(\ln^4 A_{\max})}{\mu_{\min}\epsilon^{5.5}(1-\gamma)^{13.5}} \ln\left(\frac{|\mathcal{S}|A_{\max}}{\epsilon\delta\mu_{\min}(1-\gamma)}\right)\right] \tag{118}$$

$$= \widetilde{\mathcal{O}}\left(\frac{t_{\mathrm{mix}}A_{\max}}{\mu_{\min}\epsilon^{5.5}(1-\gamma)^{13.5}}\right)$$

where (i) uses $N_{k,t} \le \overline{N}_{k,t}$ and eqs. (114)&(116), (ii) uses $\gamma^{-1/2}-1 = \mathcal{O}(1-\gamma)$, $1-(1-\eta\tau)^{3/5} = \mathcal{O}(\eta\tau) = \mathcal{O}\left(\frac{\epsilon(1-\gamma)^2}{\ln A_{\max}}\right)$ and $(1-\eta\tau)^{T_k-1} = \gamma^k$, (iii) uses $\gamma^{-3/5}-1 = \mathcal{O}(1-\gamma)$ and $\gamma^K = \frac{\epsilon^5(1-\gamma)^{7.5}}{\ln^{5/2} A_{\max}}$ implied by the hyperparameter choice (112), and (iv) uses the fact that the second term of (iii) is larger than the first term of (iii). $\square$

## F    SUMMARY OF COMPARISON OF SAMPLE COMPLEXITIES

In the column "Model-free", a ✓ means that an algorithm does not need any prior knowledge of the environment, including reward mapping and transition kernel.

In the column "Private update", a ✓ means that the algorithm updates do not involve the opponent's sensitive information, including action and policy.

Table 1: Summary of sample complexity of algorithms for solving discounted infinite-horizon zero-sum Markov games. ($|\mathcal{S}|$ is the number of states. $A_{\max}$ is the maximum number of actions between the players. $\mu_{\min}$ is the lower bound of state stationary distribution. $\gamma$ is the discount factor.)

| Work | Model -free | Private update | Symmetric update | Data type | Sample complexity (duality gap $\leq \epsilon$) |
|---|---|---|---|---|---|
| Zou et al. (2019) | ✓ | × | ✓ | Markovian | – |
| Zhao et al. (2021) | ✓ | ✓ | × | i.i.d. | – |
| Guo et al. (2021) | ✓ | ✓ | × | i.i.d. | – |
| Cen et al. (2021) | × | ✓ | ✓ | – | – |
| Wei et al. (2021) | ✓ | ✓ | ✓ | Markovian | $\widetilde{O}\big(\frac{A_{\max}^3|\mathcal{S}|^{10.5}}{\epsilon^8\mu_{\min}(1-\gamma)^{29.5}}\big)$ |
| **Our work** | ✓ | ✓ | ✓ | **Markovian** | $\widetilde{\mathcal{O}}\big(\frac{A_{\max}}{\epsilon^{5.5}\mu_{\min}(1-\gamma)^{13.5}}\big)$ |

In the column "Symmetric update", a ✓ means that both players perform symmetric updates.

In the column "Data type", "Markovian" means that the algorithm uses samples queried from the dependent Markov decision process. The "–" sign means no stochastic samples are used.

The sample complexity is defined as the total number of samples required to achieve an $\epsilon$-duality gap for the Markov game. A "–" sign in this column means that such type of sample complexity is unavailable in that paper or its setting is different from ours. We use $\widetilde{\mathcal{O}}$ to hide all the logarithm factors.

We next explain how the sample complexity of (Wei et al., 2021) is calculated. Corollary 4 of (Wei et al., 2021) shows that the iterate-average duality gap is smaller than $\epsilon$ with the hyper-parameter choices: number of iterations $T = \widetilde{\mathcal{O}}\Big[\frac{|\mathcal{S}|^2}{\eta^2(1-\gamma)^4\epsilon^2}\Big]$ and number of samples queried per iteration $L = \widetilde{\mathcal{O}}\Big[\frac{A_{\max}^3|\mathcal{S}|^6}{(1-\gamma)^{13}\mu_{\min}\eta^3\epsilon^6}\Big]$ (we replace $\xi$ with $\epsilon$). Hence, the overall sample complexity is

$$LT = \widetilde{\mathcal{O}}\Big[\frac{A_{\max}^3|\mathcal{S}|^8}{(1-\gamma)^{17}\mu_{\min}\eta^5\epsilon^8}\Big] \overset{(i)}{\geq} \widetilde{\mathcal{O}}\Big[\frac{A_{\max}^3|\mathcal{S}|^{10.5}}{(1-\gamma)^{29.5}\mu_{\min}\epsilon^8}\Big],$$

where (i) uses the choice of learning rate $\eta \leq \mathcal{O}\Big[\sqrt{\frac{(1-\gamma)^5}{|\mathcal{S}|}}\Big]$ required by their Algorithm 1.

## G  RATIONALITY

In this section, we will prove that the policy extragradient (PE) algorithm Cen et al. (2021) is rational. The following analysis can be directly extended to our stochastic policy extragradient (SPE) algorithm by applying the estimation error bounds in Appendix B.

An algorithm is called rational if, whenever the player 2 adopts an arbitrary stationary policy $\pi^{(2)}$, the policy of the player 1 converges to the best response to $\pi^{(2)}$, i.e., $\widetilde{\pi}_{*\tau}^{(1)}(s) = \arg\max_{\pi^{(1)}(s)} V_{\pi^{(1)},\pi^{(2)}}^{(\tau)}(s)$ for all $s \in \mathcal{S}$. We add tilde in $\widetilde{\pi}_{*\tau}^{(1)}(s)$ to differentiate it from the Nash policy pair $(\pi_{*\tau}^{(1)}(s), \pi_{*\tau}^{(2)}(s))$ and some consequent notations are similar. Denote $\widetilde{Q}_*^{(\tau)} := Q_{\widetilde{\pi}_{*\tau}^{(1)},\pi^{(2)}}^{(\tau)}$, $\widetilde{V}_*^{(\tau)} := V_{\widetilde{\pi}_{*\tau}^{(1)},\pi^{(2)}}^{(\tau)}$ as the optimal value functions and define the soft Bellman operator as follows

$$\widetilde{\mathcal{B}}_\tau(Q)(s, a^{(1)}, a^{(2)})$$
$$:= R(s, a^{(1)}, a^{(2)}) + \gamma\mathbb{E}_{s'\sim\mathcal{P}(\cdot|s,a^{(1)},a^{(2)})}\Big[\max_{\pi^{(1)}(s')} f_\tau\big(Q(s'); \pi^{(1)}(s'), \pi^{(2)}(s')\big)\Big]. \quad (119)$$

Similar to eqs. (98)&(99), it can be proved that the soft Bellman operator $\widetilde{\mathcal{B}}_\tau$ is also a contraction operator and has a unique fixed point $\widetilde{Q}_*^{(\tau)}$, i.e.,

$$\|\widetilde{\mathcal{B}}_\tau(Q') - \widetilde{\mathcal{B}}_\tau(Q)\|_\infty \leq \gamma\|Q' - Q\|_\infty, \quad (120)$$

$$\widetilde{\mathcal{B}}_\tau(\widetilde{Q}_*^{(\tau)}) = \widetilde{Q}_*^{(\tau)}. \tag{121}$$

To solve modified Markov game with fixed policy $\pi^{(2)}$, we develop the single-player-perspective version of the PE algorithm in Algorithm 2, which basically approximates the value iteration $Q_{k+1} = \widetilde{\mathcal{B}}_\tau(Q_k)$. The main modification from the original PE algorithm is that the new policy update (126) only keeps the third expression of the PU steps (8) since only that affects the final policy of player 1 with fixed $\pi^{(2)}$. This new policy update (126) aims to find $\pi^{(1)}$ that maximizes the following single-player Markov game problem with fixed $Q_k(s)$ and $\pi^{(2)}(s)$ for all $s \in \mathcal{S}$, while the original PU algorithm seeks the Nash equilibrium $(\pi^{(1)}, \pi^{(2)})$ of the following function.

$$\max_{\pi^{(1)}(s)} f_\tau(Q_k(s); \pi^{(1)}(s), \pi^{(2)}(s))$$
$$:= [\pi^{(1)}(s)]^\top Q_k(s)\pi^{(2)}(s) + \tau\mathcal{H}\big(\pi^{(1)}(s)\big) - \tau\mathcal{H}\big(\pi^{(2)}(s)\big). \tag{122}$$

Similar to eq. (23), it can be derived that the solution $\widetilde{\pi}_k^{*(1)}(s)$ to the above optimization problem satisfies the following condition Cen et al. (2021).

$$\widetilde{\pi}_k^{*(1)}(s) \propto \exp(Q_k(s)\pi_2(s)/\tau). \tag{123}$$

---

**Algorithm 2** Single-player perspective PE algorithm for entropy-regularized Markov game.

**Initialize:** $V_0(s)$ for all $s \in \mathcal{S}$.
**for** *value iterations* $k = 0, 1, \ldots, K-1$ **do**

Compute the following Q functions for all $(s, a^{(1)}, a^{(2)})$.

$$Q_k(s, a^{(1)}, a^{(2)}) = R(s, a^{(1)}, a^{(2)}) + \gamma\mathbb{E}_{s' \sim \mathcal{P}(s, a^{(1)}, a^{(2)})}V_k(s') \tag{124}$$

$$Q_k^{(1)}(s, a^{(1)}) = \mathbb{E}_{a^{(2)} \sim \pi^{(2)}(s)}Q_k(s, a^{(1)}, a^{(2)}) \tag{125}$$

Initialize $\pi_{k,0}^{(1)}$ with uniform distribution.
**for** *PU iterations* $t = 0, 1, \ldots, T-1$ **do**

Player 1 implements the $t$-th policy update for all $s, a^{(1)}$ as follows.

$$\pi_{k,t+1}^{(1)}(a^{(1)}|s) \propto \pi_{k,t}^{(1)}(a^{(1)}|s)^{1-\eta\tau} \exp\big(\eta Q_k^{(1)}(s, a^{(1)})\big), \tag{126}$$

**end**

Let $\pi_k^{(1)} = \pi_{k,T}^{(1)}$, and perform the following value iteration for all $s$

$$V_{k+1}(s) := f_\tau(Q_k(s); \pi_k^{(1)}(s), \pi^{(2)}(s)) \tag{127}$$

**end**
**Output:** $\pi_{K-1}^{(1)}$.

---

We first prove that the value functions $V_k$ and $Q_k$ in Algorithm 2 are bounded as follows.

**Lemma 7.** *If* $\|V_0\|_\infty \le V_{\max} := \frac{1+\tau\ln A_{\max}}{1-\gamma}$ *in Algorithm 2, then for all* $k \ge 0$, $0 \le t \le T_k - 1$, *we have*

$$\max\big(\|V_k\|_\infty, \|\widetilde{V}_*^{(\tau)}\|_\infty\big) \le V_{\max}, \tag{128}$$

$$\max\big(\|Q_k\|_\infty, \widetilde{Q}_*^{(\tau)}\|_\infty\|_\infty\big) \le Q_{\max}, \tag{129}$$

*where* $Q_{\max} := 1 + \gamma V_{\max} = \frac{1+\gamma\tau\ln A_{\max}}{1-\gamma}$ *with* $A_{\max} := \max\big(|\mathcal{A}^{(1)}|, |\mathcal{A}^{(2)}|\big)$.

*Proof.* The proof is similar to that of Lemma 3. $\square$

Similar to Lemma 1, we can prove the following lemma about the convergence rates of the modified PU steps defined by eq. (126).

**Lemma 8.** *Apply Algorithm 2 to solve the entropy regularized game with $\tau > 0$. Choose learning rate $\eta \leq \tau^{-1}$ and initialization $\|V_0\|_\infty \leq V_{\max}$. Then, the policy update defined by eq. (126) for maximizing the function (122) has the following convergence rates.*

$$KL(\widetilde{\pi}_k^{*(1)}(s)\|\pi_{k,T}^{(1)}(s)) + KL(\pi_{k,T}^{(1)}(s)\|\widetilde{\pi}_k^{*(1)}(s)) \leq \frac{2Q_{\max}}{\tau}(1 - \eta\tau)^T \tag{130}$$

$$f_\tau(Q_k(s); \widetilde{\pi}_k^{*(1)}(s), \pi^{(2)}(s)) - V_{k+1}(s) = \tau KL(\pi_{k,T}^{(1)}(s)\|\widetilde{\pi}_k^{*(1)}(s)) \leq 2Q_{\max}(1 - \eta\tau)^T \tag{131}$$

*Proof.* Taking logarithm of eq. (126) yields that there exists $c_{k,t} \in \mathbb{R}$ such that

$$\ln \pi_{k,t+1}^{(1)}(s) = (1 - \eta\tau) \ln \pi_{k,t}^{(1)}(s) + \eta Q_k(s)\pi_2(s) + c_{k,t}\mathbf{1}.$$

Iterating the above equality over $t = 0, 1, \ldots, t$ yields that there exists $c \in \mathbb{R}$ such that

$$\ln \pi_{k,T}^{(1)}(s) = \frac{1 - (1 - \eta\tau)^T}{\tau} Q_k(s)\pi_2(s) + c\mathbf{1} \tag{132}$$

where we use the uniform policy initialization $\pi_{k,0}^{(1)}(s) = \mathbf{1}/|\mathcal{A}^{(1)}|$.

Taking logarithm of eq. (123) yields that there exists $c^* \in \mathbb{R}$ such that

$$\ln \widetilde{\pi}_k^{*(1)}(s) = Q_k(s)\pi_2(s)/\tau + c^*\mathbf{1} \tag{133}$$

Therefore, eq. (130) can be proved as follows.

$$KL(\widetilde{\pi}_k^{*(1)}(s)\|\pi_{k,T}^{(1)}(s)) + KL(\pi_{k,T}^{(1)}(s)\|\widetilde{\pi}_k^{*(1)}(s))$$
$$= \left\langle \widetilde{\pi}_k^{*(1)}(s), \ln \widetilde{\pi}_k^{*(1)}(s) - \ln \pi_{k,T}^{(1)}(s) \right\rangle + \left\langle \pi_{k,T}^{(1)}(s), \ln \pi_{k,T}^{(1)}(s) - \ln \widetilde{\pi}_k^{*(1)}(s) \right\rangle$$
$$\overset{(i)}{=} \left\langle \widetilde{\pi}_k^{*(1)}(s) - \pi_{k,T}^{(1)}(s), \frac{(1 - \eta\tau)^T}{\tau} Q_k(s)\pi_2(s) + (c^* - c)\mathbf{1} \right\rangle$$
$$\overset{(ii)}{=} \frac{(1 - \eta\tau)^T}{\tau}[\widetilde{\pi}_k^{*(1)}(s) - \pi_{k,T}^{(1)}(s)]^\top Q_k(s)\pi_2(s)$$
$$\overset{(iii)}{\leq} \frac{2Q_{\max}}{\tau}(1 - \eta\tau)^T$$

where (i) uses eqs. (132)&(133), (ii) uses $\langle \widetilde{\pi}_k^{*(1)}(s) - \pi_{k,T}^{(1)}(s), \mathbf{1} \rangle = 0$ and (iii) uses $\|Q_k(s)\|_\infty \leq Q_{\max}$, $\|\widetilde{\pi}_k^{*(1)} - \pi_{k,T}^{(1)}\|_1 \leq 2$, $\|\pi_2(s)\|_1 = 1$ and $1 - \eta\tau > 0$.

$$f_\tau(Q_k(s); \widetilde{\pi}_k^{*(1)}(s), \pi^{(2)}(s)) - V_{k+1}(s)$$
$$= f_\tau(Q_k(s); \widetilde{\pi}_k^{*(1)}(s), \pi^{(2)}(s)) - f_\tau(Q_k(s); \pi_{k,T}^{(1)}(s), \pi^{(2)}(s))$$
$$= \left\langle \widetilde{\pi}_k^{*(1)}(s) - \pi_{k,T}^{(1)}(s), Q_k(s)\pi^{(2)}(s) \right\rangle + \tau\left\langle \pi_{k,T}^{(1)}(s), \ln \pi_{k,T}^{(1)}(s) \right\rangle - \tau\left\langle \widetilde{\pi}_k^{*(1)}(s), \ln \widetilde{\pi}_k^{*(1)}(s) \right\rangle$$
$$\overset{(i)}{=} \left\langle \widetilde{\pi}_k^{*(1)}(s) - \pi_{k,T}^{(1)}(s), \tau\ln \widetilde{\pi}_k^{*(1)}(s) \right\rangle + \tau\left\langle \pi_{k,T}^{(1)}(s), \ln \pi_{k,T}^{(1)}(s) \right\rangle - \tau\left\langle \widetilde{\pi}_k^{*(1)}(s), \ln \widetilde{\pi}_k^{*(1)}(s) \right\rangle$$
$$= \tau KL(\pi_{k,T}^{(1)}(s)\|\widetilde{\pi}_k^{*(1)}(s))$$
$$\overset{(ii)}{\leq} 2Q_{\max}(1 - \eta\tau)^T$$

where (i) uses eq. (133) and $\langle \widetilde{\pi}_k^{*(1)}(s) - \pi_{k,T}^{(1)}(s), \mathbf{1} \rangle = 0$ and (ii) uses eq. (130). $\qquad\square$

Finally, we prove the convergence rate of Algorithm 2 as follows.

**Theorem 3.** *Apply Algorithm 2 to solve the entropy regularized game with $\tau > 0$. Choose learning rate $\eta \leq \tau^{-1}$ and initialization $\|V_0\|_\infty \leq V_{\max}$. Then, the Q function estimation error $\|Q_{K-1} - \widetilde{Q}_*^{(\tau)}\|_\infty$ and the function value gap $\max_s \left[\widetilde{V}_*^{(\tau)}(s) - V_{\pi_{K-1}^{(1)}, \pi^{(2)}}^{(\tau)}(s)\right]$ converge at the following rates*

$$\|Q_{K-1} - \widetilde{Q}_*^{(\tau)}\|_\infty \leq 2Q_{\max}\gamma^K + \frac{2\gamma Q_{\max}}{1 - \gamma}(1 - \eta\tau)^T, \tag{134}$$

$$\max_s \left[\widetilde{V}_*^{(\tau)}(s) - V_{\pi_{K-1}^{(1)}, \pi^{(2)}}^{(\tau)}(s)\right] \leq \frac{2Q_{\max}}{1 - \gamma}\left(2\gamma^{K-1} + \frac{1 + \gamma}{1 - \gamma}(1 - \eta\tau)^T\right). \tag{135}$$

*Proof.*

$$\|Q_{k+1} - \widetilde{Q}_*^{(\tau)}\|_\infty$$

$$\overset{(i)}{\leq} \|Q_{k+1} - \widetilde{\mathcal{B}}_\tau(Q_k)\|_\infty + \|\widetilde{\mathcal{B}}_\tau(Q_k) - \widetilde{\mathcal{B}}_\tau(\widetilde{Q}_*^{(\tau)})\|_\infty$$

$$\overset{(ii)}{\leq} \max_{s,a^{(1)},a^{(2)}} |Q_{k+1}(s,a^{(1)},a^{(2)}) - \widetilde{\mathcal{B}}_\tau(Q_k)(s,a^{(1)},a^{(2)})| + \gamma\|Q_k - \widetilde{Q}_*^{(\tau)}\|_\infty$$

$$= \max_{s,a^{(1)},a^{(2)}} \Big| R(s,a^{(1)},a^{(2)}) + \gamma\mathbb{E}_{s'\sim\mathcal{P}(\cdot|s,a^{(1)},a^{(2)})} V_{k+1}(s')$$

$$\quad - \Big[ R(s,a^{(1)},a^{(2)}) + \gamma\mathbb{E}_{s'\sim\mathcal{P}(\cdot|s,a^{(1)},a^{(2)})} f_\tau\big(Q_k(s'); \widetilde{\pi}_k^{*(1)}(s'), \pi^{(2)}(s')\big)\Big] \Big| + \gamma\|Q_k - \widetilde{Q}_*^{(\tau)}\|_\infty$$

$$\leq \gamma\max_{s'\in\mathcal{S}} \big| V_{k+1}(s') - f_\tau\big(Q_k(s'); \widetilde{\pi}_k^{*(1)}(s'), \pi^{(2)}(s')\big)\big| + \gamma\|Q_k - \widetilde{Q}_*^{(\tau)}\|_\infty$$

$$\overset{(iii)}{\leq} 2\gamma Q_{\max}(1 - \eta\tau)^T + \gamma\|Q_k - \widetilde{Q}_*^{(\tau)}\|_\infty,$$

where (i) uses eq. (121), (ii) uses eq. (120), (iii) uses eq. (131). Iterating the above inequality yields that

$$\|Q_{K-1} - \widetilde{Q}_*^{(\tau)}\|_\infty \leq \gamma^K\|Q_0 - \widetilde{Q}_*^{(\tau)}\|_\infty + 2\gamma Q_{\max}(1 - \eta\tau)^T \frac{1 - \gamma^T}{1 - \gamma}$$

$$\leq 2Q_{\max}\gamma^K + \frac{2\gamma Q_{\max}}{1 - \gamma}(1 - \eta\tau)^T,$$

which proves eq. (134). Finally, eq. (135) can be proved as follows

$$\widetilde{V}_*^{(\tau)}(s) - V_{\pi_{K-1}^{(1)},\pi^{(2)}}^{(\tau)}(s)$$

$$= \sum_{a^{(1)},a^{(2)}} \big[\widetilde{Q}_*^{(\tau)}(s,a^{(1)},a^{(2)})\widetilde{\pi}_{*\tau}^{(1)}(a^{(1)}|s) - Q_{\pi_{K-1}^{(1)},\pi^{(2)}}^{(\tau)}(s,a^{(1)},a^{(2)})\pi_{K-1}^{(1)}(a^{(1)}|s)\big]\pi^{(2)}(a^{(2)}|s)$$

$$\quad + \tau\big[\mathcal{H}\big(\widetilde{\pi}_{*\tau}^{(1)}(s)\big) - \mathcal{H}\big(\pi_{K-1}^{(1)}(s)\big)\big]$$

$$= \sum_{a^{(1)},a^{(2)}} \big[\widetilde{Q}_*^{(\tau)}(s,a^{(1)},a^{(2)}) - Q_{\pi_{K-1}^{(1)},\pi^{(2)}}^{(\tau)}(s,a^{(1)},a^{(2)})\big]\pi_{K-1}^{(1)}(a^{(1)}|s)\pi^{(2)}(a^{(2)}|s)$$

$$\quad + \sum_{a^{(1)},a^{(2)}} \widetilde{Q}_*^{(\tau)}(s,a^{(1)},a^{(2)})\big[\widetilde{\pi}_{*\tau}^{(1)}(a^{(1)}|s) - \pi_{K-1}^{(1)}(a^{(1)}|s)\big]\pi^{(2)}(a^{(2)}|s)$$

$$\quad + \tau\big[\mathcal{H}\big(\widetilde{\pi}_{*\tau}^{(1)}(s)\big) - \mathcal{H}\big(\pi_{K-1}^{(1)}(s)\big)\big]$$

$$= \gamma\sum_{a^{(1)},a^{(2)}} \Big[\pi_{K-1}^{(1)}(a^{(1)}|s)\pi^{(2)}(a^{(2)}|s)\mathbb{E}_{s'\sim\mathcal{P}(\cdot|s,a^{(1)},a^{(2)})}\big[\widetilde{V}_*^{(\tau)}(s') - V_{\pi_{K-1}^{(1)},\pi^{(2)}}^{(\tau)}(s')\big]\Big]$$

$$\quad + \big(\widetilde{\pi}_{*\tau}^{(1)}(s) - \pi_{K-1}^{(1)}(s)\big)^\top \widetilde{Q}_*^{(\tau)}(s)\pi^{(2)}(s) + \tau\big[\mathcal{H}\big(\widetilde{\pi}_{*\tau}^{(1)}(s)\big) - \mathcal{H}\big(\pi_{K-1}^{(1)}(s)\big)\big]$$

$$\leq \gamma\max_{s'} \big[\widetilde{V}_*^{(\tau)}(s') - V_{\pi_{K-1}^{(1)},\pi^{(2)}}^{(\tau)}(s')\big] + \big(\widetilde{\pi}_{*\tau}^{(1)}(s) - \pi_{K-1}^{(1)}(s)\big)^\top \widetilde{Q}_*^{(\tau)}(s)\pi^{(2)}(s)$$

$$\quad + \tau\big[\mathcal{H}\big(\widetilde{\pi}_{*\tau}^{(1)}(s)\big) - \mathcal{H}\big(\pi_{K-1}^{(1)}(s)\big)\big].$$

Applying $\max_s$ to both sides of the above inequality and rearranging it yields that

$$\max_s \big[\widetilde{V}_*^{(\tau)}(s) - V_{\pi_{K-1}^{(1)},\pi^{(2)}}^{(\tau)}(s)\big]$$

$$\leq \frac{1}{1-\gamma}\max_s \Big(\big(\widetilde{\pi}_{*\tau}^{(1)}(s) - \pi_{K-1}^{(1)}(s)\big)^\top \widetilde{Q}_*^{(\tau)}(s)\pi^{(2)}(s) + \tau\big[\mathcal{H}\big(\widetilde{\pi}_{*\tau}^{(1)}(s)\big) - \mathcal{H}\big(\pi_{K-1}^{(1)}(s)\big)\big]\Big)$$

$$\leq \frac{1}{1-\gamma}\max_s \Big(f_\tau\big(\widetilde{Q}_*^{(\tau)}(s); \widetilde{\pi}_{*\tau}^{(1)}(s), \pi^{(2)}(s)\big) - f_\tau\big(\widetilde{Q}_*^{(\tau)}(s); \pi_{K-1}^{(1)}(s), \pi^{(2)}(s)\big)\Big)$$

$$\leq \frac{1}{1-\gamma}\max_s \Big(f_\tau\big(\widetilde{Q}_*^{(\tau)}(s); \widetilde{\pi}_{*\tau}^{(1)}(s), \pi^{(2)}(s)\big) - f_\tau\big(Q_{K-1}(s); \widetilde{\pi}_{K-1}^{*(1)}, \pi^{(2)}(s)\big)\Big)$$

$$\quad + \frac{1}{1-\gamma}\max_s \Big(f_\tau\big(Q_{K-1}(s); \widetilde{\pi}_{K-1}^{*(1)}(s), \pi^{(2)}(s)\big) - f_\tau\big(Q_{K-1}(s); \pi_{K-1}^{(1)}(s), \pi^{(2)}(s)\big)\Big)$$

$$+ \frac{1}{1 - \gamma} \max_s \left( f_\tau \big( Q_{K-1}(s); \pi^{(1)}_{K-1}(s), \pi^{(2)}(s) \big) - f_\tau \big( \widetilde{Q}^{(\tau)}_*(s); \pi^{(1)}_{K-1}(s), \pi^{(2)}(s) \big) \right)$$

$$\leq \frac{1}{1 - \gamma} \max_s \left| \left( \max_{\pi^{(1)}(s)} f_\tau \big( \widetilde{Q}^{(\tau)}_*(s); \pi^{(1)}(s), \pi^{(2)}(s) \big) \right) - \left( \max_{\pi^{(1)}(s)} f_\tau \big( Q_{K-1}(s); \pi^{(1)}(s), \pi^{(2)}(s) \big) \right) \right|$$

$$+ \frac{1}{1 - \gamma} \max_s \left( f_\tau \big( Q_{K-1}(s); \widetilde{\pi}^{*(1)}_{K-1}(s), \pi^{(2)}(s) \big) - V_K(s) \right)$$

$$+ \frac{1}{1 - \gamma} \max_s \left| f_\tau \big( Q_{K-1}(s); \pi^{(1)}_{K-1}(s), \pi^{(2)}(s) \big) - f_\tau \big( \widetilde{Q}^{(\tau)}_*(s); \pi^{(1)}_{K-1}(s), \pi^{(2)}(s) \big) \right|$$

$$\overset{(i)}{\leq} \frac{2}{1 - \gamma} \left( 2 Q_{\max} \gamma^K + \frac{2 \gamma Q_{\max}}{1 - \gamma} (1 - \eta\tau)^T \right) + \frac{2 Q_{\max}}{1 - \gamma} (1 - \eta\tau)^T$$

$$\leq \frac{2 Q_{\max}}{1 - \gamma} \left( 2 \gamma^{K-1} + \frac{1 + \gamma}{1 - \gamma} (1 - \eta\tau)^T \right),$$

where (i) uses eq. (131) and the following inequality that holds for all $s, \pi^{(1)}(s), \pi^{(2)}(s)$, (ii) uses eq. (134) and $V_{K-1}(s) = f_\tau \big( Q^{(\tau)}_{K-1}(s); \pi^{(1)}_{K-1}(s), \pi^{(2)}(s) \big)$.

$$\left| f_\tau \big( Q_{K-1}(s); \pi^{(1)}_{K-1}(s), \pi^{(2)}(s) \big) - f_\tau \big( \widetilde{Q}^{(\tau)}_*(s); \pi^{(1)}_{K-1}(s), \pi^{(2)}(s) \big) \right|$$

$$\leq \| Q_{K-1}(s) - \widetilde{Q}^{(\tau)}_*(s) \|_\infty \overset{eq.(134)}{\leq} 2 Q_{\max} \gamma^K + \frac{2 \gamma Q_{\max}}{1 - \gamma} (1 - \eta\tau)^T.$$

$\square$

In Theorem 3, the convergence rates of both the Q function estimation error $\| Q_{K-1} - \widetilde{Q}^{(\tau)}_* \|_\infty$ and the function value gap $\max_s \left[ \widetilde{V}^{(\tau)}_*(s) - V^{(\tau)}_{\pi^{(1)}_{K-1}, \pi^{(2)}}(s) \right]$ consist of two terms $\mathcal{O}(\gamma^K)$ and $\mathcal{O}(1 - \eta\tau)^T$, which characterize the exponential convergence rates of the $K$ outer value iterations and the $T$ inner policy updates respectively. Such a convergence result indicates that Algorithm 2 converges to the optimal solution to the regularized Markov game, and thus can be arbitrarily close to the optimal solution of the unregularized Markov game with sufficiently small $\tau$. As a result, the PU algorithm Cen et al. (2020) is rational. Similar result can be directly extended to our stochastic policy extragradient (SPE) algorithm by applying the estimation error bounds in Appendix B.

