# OpenReview forum: "Sample Efficient Stochastic Policy Extragradient Algorithm for Zero-Sum Markov Game"
_ICLR.cc/2022/Conference — ICLR 2022 Poster_

### Official Review · Reviewer_sRxd · 2021-11-01

**Correctness:** 4
**Technical Novelty And Significance:** 2
**Empirical Novelty And Significance:** Not applicable
**Recommendation:** 6
**Confidence:** 4

**Main Review:**

Strengths:

First of all I think the theoretical result is solid, and in terms of the result, the improvement over existing work is clear ($\epsilon$ dependency from $1/\epsilon^8$ to $1/\epsilon^{5.5}$). I also find it satisfying that the sample complexity analysis is placed at a central position in this paper, as with gradient-based algorithms sometimes only the population analysis is presented (assuming full game knowledge). Given that two-player zero-sum MGs is maybe one of the most important problem settings for multi-agent RL / games, I think this result should be of good interest to the community.

Overall the presentation of this paper seems clear. All the problem settings, assumptions, and results seem clearly presented as well. There is a reasonable amount of discussions on the proof techniques (for example, the improved stochastic estimators) from which I can get a hint of where the theoretical improvements come from.

Weaknesses:

After comparing with the closely related prior works of Cen et al. (2021) and Wei et al. (2021), I find the present contribution perhaps a bit incremental. The relationship seems to be that Cen et al. proposed the population version of the extragradient algorithm but did not do a sample complexity analysis. Wei et al. gives the sample complexity analysis, but for the vanilla policy gradient algorithm (without extragradient, and with Euclidean projection instead of mirror map). In this sense, at a high level, the present paper may be seen as somewhat a direct combination of these two papers, though some technical improvements are made (e.g. the better stochastic estimators in (14)(15)).

Related to the above point, I also find the paper lacking some high-level discussions in terms of where the theoretical gains over Wei et al. come from. First, I wonder in the population case, what is the comparison between extragradient (+mirror map) vs. policy gradient of Wei et al. The authors mentioned on Page 5 the regularized (smoothed) extragradient algorithm converges exponentially for each $\tau$, but how about picking $\tau$ to get convergence to $\epsilon$-unregularized Nash? In this case what is the difference in the convergence rates between extragradient and gradient? (Feels like this may be resolved by presenting a head-to-head comparison between the population results of Cen et al. and Wei et al.)? Then, after going from population -> samples, how many additional $\epsilon$’s does the present paper pay (and how does Wei et al. do?) Since this paper uses improved stochastic estimators, is the number of additional $\epsilon$’s better than Wei et al.?

Finally, I am very suspicious of the motivation of “private policy updates” in the abstract / intro, as many prior algorithms also have this feature (as the paper mentions in Table 1) and this is quite common for any sample-based decentralized learning algorithm. Further, the joint features of “model-free, convergent, sample efficient, symmetric, and private policy updates” are also not that uncommon, for example in the V-learning algorithm of Bai et al. (2020), as well as the vanilla policy gradient algorithm of Wei et al. (2021). Therefore the claim of “we develop algorithms… with all these properties” needs to be revised. I suggest the authors modify the tone here to focus more on the improved sample complexity for the gradient-based algorithm (which is the key contribution in my opinion) instead of the above features.

Other comments / typos:


Equation (14), (15): Denominators should be $\pi^{'(m)}_{k, t}(a^{(m)} | s)$?

---
Update: I thank the authors for their response and revisions to the paper. I think the authors have addressed my concerns about the theoretical improvement of population vs. samples via the added discussions below Theorem 2, and I am now more convinced about the technical contributions. I have raised my score accordingly.

**Summary Of The Paper:**

This paper theoretically studies gradient-based algorithms for two-player zero-sum Markov Games (MGs), an important problem in multi-agent reinforcement learning. The main contribution of this paper is a sample-based policy extragradient algorithm for finding an approximate Nash equilibrium of the MG, with improved sample complexity guarantees over existing policy gradient based algorithms.

**Summary Of The Review:**

Overall, I think this paper makes a solid theoretical contribution in the problem of gradient-based algorithms for zero-sum MGs. However currently the contributions seem slightly incremental over the two recent prior works, and there is a lack of high-level discussions on where (population vs. samples) the exact improvements come from.

---

> ### Author Response · Authors · 2021-11-19
> **Response to Review**
>
> Thank you very much for reviewing our manuscript and providing valuable feedback.  Below is a response to the review comments. We have submitted a revised version with all revisions marked in `red`. Please let us know if further clarifications are needed.
>
> **Q1:** The relationship seems to be that (Cen 2021) proposed the population version of the extragradient algorithm but did not do a sample complexity. Contribution is incremental.
>
> **A:** We agree that this work develops a stochastic version of the population extragradient algorithm proposed in (Cen 2021). We want to clarify that our analysis of the stochastic part is substantially different from that of (Wei 2021), and we need to address more technical challenges to establish the improved sample complexity. In the revision, at the bottom of Page 8, we added a ’Technical Novelty’ paragraph that lists the major technical challenges and the technical novelties.
> Specifically, (Wei 2021) bounds the estimation error of the stochastic estimators in a simple way by applying Azuma-Hoeffding’s inequality with independent transition samples (see their proof of Theorem 3 in Appendix H.1). Hence, their results only apply to the independent setting, although their setup in the main text claims to use Markovian samples. As a comparison, our analysis of the estimation error bounds the additional bias induced by the correlated Markovian samples. Moreover, (Wei 2021)'s convergence rate analysis bounds the estimation errors in all iterations to be a universal small constant, which requires a large number of samples. As a comparison, we develop a special contraction property of the estimation error over the iterations that leads to a tight error bound.
>
> **Q2:**  What's our sample complexity improvement over from Wei et al. (2021) in both population level vs. sample estimation?
>
> **A:** Great question. In the revision, below Theorem 2, we added one paragraph that remarks on the improvement of sample complexity over (Wei 2021) in both the population level and stochastic level. To briefly summarize, in the population level, the original PE algorithm in (Cen 2021) improves the iteration complexity over that of the OGDA-based algorithm in (Wei 2021) by a factor of $\mathcal{O}(\epsilon^{-1})$. In the stochastic level, by using advanced techniques to improve the estimation error bounds, our stochastic PE algorithm further improves the sample complexity by a factor of $\mathcal{O}(\epsilon^{-1.5})$. Please refer to our remark for more details.
>
>
> **Q3:** The joint features of “model-free, convergent, sample efficient, symmetric, and private policy updates” are common. The claim of contribution should focus on sample complexity.
>
> **A:** Thank you for the suggestion and we agree. In the revision, we have revised the abstract and focuses on the improvement of sample complexity. In the introduction, we combined the two questions into one and also focuses on the improvement of sample complexity.
>
>
> **Q4:** Typo in eqs. (14), (15): Denominators should be $\pi_{k,t}'^{(m)}(a^{(m)}|s)$.
>
> **A:** Thank you for pointing that out. We have corrected it in our revision.
>
> As suggested by the other reviewer, we have also included a proof of the rationality of the policy extragradient algorithm in the Appendix G.

---

### Official Review · Reviewer_RwGu · 2021-11-02

**Correctness:** 4
**Technical Novelty And Significance:** 3
**Empirical Novelty And Significance:** Not applicable
**Recommendation:** 6
**Confidence:** 4

**Main Review:**

The paper investigates a multi-agent RL problem from a policy gradient perspective, which is an interesting problem. The paper is well-written. Below are my comments and questions regarding the paper.

- In the single-agent case, entropy regularization with coefficient $\tau > 0$ has two effects: improves convergence rate in the regularized MDP, and encourages exploration (Mei et al., 2020; Bhandari & Russo, 2019; Shani et al., 2019). In particular, entropy regularization makes $\min_{s,a}\pi(a|s)$ bounded away from 0, which seems to be important to prove convergence. In this paper, the regularization coefficient $\tau$ is chosen arbitrarily small to show convergence in the unregularized MDP. The first effect of entropy regularization (i.e., convergence rate) is clearly observed. What is the impact of small $\tau$ on exploration, for example, in terms of minimum action probabilities? Is it hidden in some parameters (e.g., $\mu_{min}$)?

- The paper assumes that the minimum steady-state probability under each policy pair throughout the trajectory of the policy optimization is bounded away from 0, which is a strong assumption as it imposes a condition on the nature of the policies. Furthermore, $1/\mu_{min} \geq |\mathcal{S}|$ in the most optimistic case. The dependence on $|\mathcal{S}|$ can be more clearly stated for the sake of clarity. If one considers a duality gap notion based on an initial state distribution where the initial state is distributed according to $s_0\sim\rho$, could it be possible establish a similar convergence bound under milder assumptions (e.g., $\|d_{\rho}^{\pi^{(1)}, \pi^{(2)}}/\rho\|_\infty < \infty$)?

- Concentration inequalities to estimate quantities of type $\frac{\mathbb{E}[X]}{\mathbb{E}[Y]}$ by using $\{(X_i,Y_i): i \geq 0\}$ are used in other domains (e.g., (Xia et al., 2016; Hsu et al., 2012)). It can be clarified that the number of samples should be sufficiently large to bound the denominator away from zero with high probability for the stability of the estimator.

- An interesting scenario is the function approximation regime, where a restricted policy class is considered to address large state spaces. Can the results in this paper be extended to that regime? This question is related to my first two questions, i.e., dependence on $\mu_{min} \geq |\mathcal{S}|$ and the distribution mismatch. I would be glad if the authors could provide insights on these.

---
Y. Xia et al. "Budgeted bandit problems with continuous random costs." Asian conference on machine learning. PMLR, 2016.

D. Hsu et al., "Random design analysis of ridge regression." In Conference on learning theory, pp. 9-1. JMLR Workshop and Conference Proceedings, 2012.


**Summary Of The Paper:**

The paper proposes an entropy-regularized policy extragradient method for zero-sum Markov games in the tabular setting. By using low-variance value function estimators, and entropy regularization, the algorithm is shown to achieve improved sample complexity for a given target duality gap for a Nash equilibrium solution.

**Summary Of The Review:**

The impact of entropy regularization on the exploration is elusive. The assumption that the minimum steady-state probability under each policy pair during the policy optimization steps is strong. It would be good if further insights could be provided on these.

---

> ### Author Response · Authors · 2021-11-19
> **Response to Review**
>
> Thank you very much for reviewing our manuscript and providing valuable feedback.  Below is a response to the review comments. We have submitted a revised version with all revisions marked in `red`. Please let us know if further clarifications are needed.
>
> **Q1:** What is the impact of small $\tau$ on exploration? Is $\tau$ hidden in some parameters like $\mu_{\min}$.
>
> **A:** Great question. Regarding the first question, in our technical proof, the major advantage of the entropy regularization parameter $\tau$ is to improve the geometry of the objective function to be strongly-convex-strongly-concave, but we found that choosing a small $\tau$ does not provide sufficient encouragement for exploration.
> To guarantee sufficient exploration, in the design of the stochastic estimators in eqs.(14,15), we query the samples following $\epsilon'$-smoothed policies instead of the original policies. This helps to bound the entries of the policy vectors away from zero.
> Regarding the second question, $\tau$ is not hidden in other parameters such as $\mu_{\min}$, since it only appears in the definition of the entropy-regularized value function.
>
> **Q2:** Can we use weaker assumption than $\mu_{\min}$?
>
> **A:** Great question. We first note that both our sample complexity and (Wei 2021)'s sample complexity have the same dependence on $\mu_{min}$, up to a logarithm factor. In the revision, we have included the dependence on $\mu_{\min}$ in the comparison of Table 1.
>
> We found that it is hard to relax the dependence on $\mu_{\min}$ in our context. This is because in the proof of Theorem 1, the duality gap of our SPE algorithm has to be bounded by the $\ell_{\infty}$ norm of the optimality gap of the Q-value function $||Q_k - Q^*_\tau||_\infty$.
>
> This means that we need to control this gap for all states, which requires visiting these states infinitely often. On the other hand, the distribution mismatch coefficient mentioned by the reviewer is usually involved in the application of the performance difference lemma, which turns out to be hard (at least for now) to develop for the expected duality gap $E_{s\sim \rho}[D^{\pi^{(1)}, \pi^{(2)}}(s)]$ of the Markov game. The main challenge is that the duality gap involves two separate max and min operators that lead to a completely different structure from that of the standard expected state value function $E_{s\sim \rho}[V^{\pi}(s)]$.
>
> **Q3:** There are existing works on estimating $\frac{\mathbb{E}X}{\mathbb{E}Y}$. It can be clarified that the number of samples should be sufficiently large to bound the denominator away from zero with high probability for the stability of the estimator.
>
> **A:** Thanks for pointing out the related works and we appreciate it. We agree that these estimators are not entirely new and their basic forms have been proposed/used in other works. In the revision, we added a Remark 1 above the Section 4 to clarify this and cited several works that used estimators with similar structures. We also would like to point out one major technical difference, i.e., all these works (including (Wei 2021)) analyze the estimation error with independent samples, while our analysis addresses the additional bias induced by the correlated Markovian samples. Therefore, our analysis of the estimator error is technically more involved. Please refer to the 'Technical Novelty' paragraph on Page 8 for more elaborations on this point. We also want to point out that we bound the denominator of the stochastic estimators in eqs.(14,15) away from zero by using smoothed policies. This avoids the need for a large sample size.
>
>
> **Q4:** Could the results be extended to function approximation regime.
>
> **A:** Great question. In the algorithm formulation, we can use parameterized policies instead. Specifically, suppose the two players use parameterized policies $\pi_{\omega_1}^{(1)}$, $\pi^{(2)}_{\omega_2}$, where $\omega_1, \omega_2$ denote the policy parameters.
>
> Then, in each outer iteration $k$, we propose to solve the parameterized minimax optimization problem $\max_{\omega_{1}}\min_{\omega_{2}} E_{s\sim \xi} [\pi_{\omega_{1}}^{(1)}(s)^T Q_k(s) \pi_{\omega_{2}}^{(2)}(s) +\tau \mathcal{H} (\pi_{\omega_{1}}^{(1)}(s))-\tau \mathcal{H} (\pi_{\omega_{2}}^{(2)}(s))]$, where $\xi$ denotes the initial state distribution. This is different from the tabular case, where we need to solve a entropy regularized bilinear matrix game for every $s$. However, with parameterized policies, such a minimax optimization problem is in general nonconvex-nonconcave, and no longer has the bilinear structure. For this complex problem, its duality gap may not be tractable under the PE/SPE algorithm, and instead we can consider tracking the gradient norm of the duality gap $\|\nabla D^{(\tau)}(\pi_{\omega_{1}}^{(1)},\pi_{\omega_{2}}^{(2)})\|$ as the convergence measure. We also expect that the convergence result will involve a fundamental approximation error associated with the restricted policy class used.

---

### Official Review · Reviewer_5TU3 · 2021-11-02

**Correctness:** 4
**Technical Novelty And Significance:** 3
**Empirical Novelty And Significance:** Not applicable
**Recommendation:** 6
**Confidence:** 3

**Main Review:**


## Strength

Overall the paper is well-written and clear to follow. Given that the standard policy extra gradient algorithm requires both players to coordinate with each other and share a Q-table, and updating the Q-table requires full knowledge of the transition kernel P and the reward mapping, the proposed stochastic version of it nicely addressed these limitations. Though the proposed algorithm, which uses the Markovian stochastic samples is intuitive and natural, the finite-time analysis turns out to be more involved and required some new analysis techniques.

## Weakness

As mentioned above, the stochastic variant of the policy extra gradient algorithm appears to be a very natural design choice. In terms of the technical novelty, I acknowledge that there has been a fast growing line of works on Markov games and due to my limited familiarity, I can not fully confirm on the technical novelty part of this paper.

Moreover although the main contribution is purely theoretical, the paper did not mention or include possible simulations to demonstrate the sample complexity bounds, or improvements over the past algorithms.

## Further comments
Is there any way to say about the optimality of the sample complexity that is achieved using the stochastic samples?

**Summary Of The Paper:**

## Summary

This paper studies a decentralized stochastic policy extra-gradient algorithm for solving two-player zero-sum Markov game. In comparison to the standard policy extra-gradient algorithm, this algorithm uses a set of stochastic estimators to estimate the value functions involved in the stochastic updates. Improved sample complexity guarantee is also derived.


**Summary Of The Review:**

Overall I find the paper well-written and provides good contribution to the current line of works for solving two-player zero-sum Markov games. In comparison to the standard policy extra-gradient algorithm, the proposed stochastic policy extra-gradient algorithm is model-free, and is proven to obtain a better sample complexity than the state-of-art. The paper also presented clearly the intuition of the algorithm and the new technical techniques that were involved.



-----
Post rebuttal update: I have read the authors' response and will keep my original score.

---

> ### Author Response · Authors · 2021-11-19
> **Response to Review**
>
> Thank you very much for reviewing our manuscript and providing valuable feedback.  Below is a response to the review comments. We have submitted a revised version with all revisions marked in `red`. Please let us know if further clarifications are needed.
>
> **Q1**: Cannot fully confirm on the technical novelty part of this paper.
>
> **A:** Thanks. In the revision, at the bottom of Page 8, we added a 'Technical Novelty' paragraph that lists the major technical challenges and the technical novelties. As a brief summary, our analysis provides a tight analysis of the estimator errors induced by the stochastic estimators in two perspectives: (1) we apply concentration inequalities for dependent samples to bound the bias induced by Markovian samples and establish tight high-probability estimation error bounds; (2) By developing a special error decomposition strategy and leveraging the recursive structure of the estimators, we establish a contraction property of the estimation error, which is the key to tighten the estimation error bound. Finally, we develop a finite-time convergence analysis of the duality gap of the SPU algorithm under inexact value function estimations, this together with the developed tight estimator error bounds leads to the desired improved sample complexity.
>
> Moreover, after Theorem 2, we also added more elaborations on how much our stochastic PE algorithm design and technical analysis contributes to the improvement of the sample complexity, compared with the existing literature.
>
> **Q2:** About the optimality of the sample complexity that is achieved using the stochastic samples.
>
> **A:** Great question. There is still a gap between our sample complexity and the lower bound $\widetilde{\mathcal{O}}(SAB(1-\gamma)^{-3}\epsilon^{-2}\ln(\delta^{-1}))$ established in [1] (listed below), where $A,B$ are the cardinality of the action spaces of the two players. However, to the best of our knowledge, our sample complexity is the best among all model-free algorithms with symmetric and private policy updates for solving Markov games.
>
> [1] Zhang, K., Kakade, S., Basar, T., and Yang, L. (2020). Model-based multi-agent rl in zero-sum markov games with near-optimal sample complexity. In Proc. Advances in Neural Information Processing Systems (Neurips), volume 33.

---

### Official Review · Reviewer_8RN2 · 2021-11-02

**Correctness:** 3
**Technical Novelty And Significance:** 2
**Empirical Novelty And Significance:** Not applicable
**Recommendation:** 6
**Confidence:** 3

**Main Review:**

The concepts of this paper mainly adopt from the well-studied PE algorithm and the sampling technique, which slightly hinders the novelty of this paper. However, with the elaborate design of the stochastic estimators and refined analysis, this paper achieves a significant improvement over the sample complexity. This improvement is a valid contribution to the understanding of the Markov Games. Besides, by utilizing the concentration inequalities for dependent samples as in [Paulin, 2015], the assumptions in this paper are slightly weaker. The paper is clearly presented, with each theoretical result followed by interpretations and comparisons with [Wei et al, 2021].

The main concern of this paper is on the novelty, it is not clear whether the improvement over [Wei et al. 2021] is due to the alternation from OGDA to EG, and the entropy regularization. While both techniques are previously analyzed in [Cen et al., 2021]. I am not very sure if the novel design of the estimators is a natural adaptation to the PE algorithm, or does the two approaches: the sampling from the state-action frequency to the state frequency, and the accurate original policy, have also contributed to the improvement in the sample complexity. Besides, as this paper introduces several hyperparameters taking completely different values, intuitive illustration of the convergence rate, like theoretical convergence curves under different choices of the batch sizes, the $T_k$, is preferred in comparison with [Wei et al., 2021].

**Summary Of The Paper:**

This paper focuses on the two-player zero-sum Markov Game setting and proposes a stochastic version of the policy extragradient (PE). The PE algorithm, which solves an entropy-regularized minimax matrix game problem by predictive update (PU) [Cen et al., 2021], has been well studied under the deterministic case in the work of [Cen et al., 2021], proven to converge exponentially to a unique solution. However, with an inquiry into the full Q-table, and the full transition matrix P, the original PE algorithm has its limitations in practice.

The authors of this paper design a stochastic counterpart of the PE algorithm, the main sampling technique of introducing stochasticity is similar to the stochastic OGDA in [Wei et al. 2021] with several substantial technical improvements. Theoretically, by tightly bounding the estimation error via a novel error decomposition, this paper obtains a significant sample complexity of $\tilde{\mathcal{O}}(\frac{A_{\max}}{\epsilon^{5.5}(1 - \gamma)^{13.5}})$, compared with the previous state-of-the-art result $\tilde{\mathcal{O}}(\frac{A_{\max}^3 |S|^{10.5}}{\epsilon^8 (1 - \gamma)^{29.5}})$.

**Summary Of The Review:**

Overall, this paper is well written. The techniques of extending from a deterministic PE algorithm to a stochastic PE algorithm through Markovian sampling are similar as in [Wei et al, 2021], with novel design on the estimators and necessary adaptation to the PE framework. I am leaning towards accepting this paper but have some concerns about the novelty.

---

> ### Author Response · Authors · 2021-11-19
> **Response to Review**
>
> Thank you very much for reviewing our manuscript and providing valuable feedback.  Below is a response to the review comments. We have submitted a revised version with all revisions marked in `red`. Please let us know if further clarifications are needed.
>
> **Q1:** Do the improvement over Wei et al. (2021) come from extragradient, the entropy regularization or the sample estimator?
>
> **A:** Great question. In the revision, below Theorem 2, we added one paragraph that remarks on the improvement of sample complexity over (Wei 2021) in both the population level and stochastic level. To briefly summarize, in the population level, the original PE algorithm in (Cen 2021) improves the iteration complexity over that of the OGDA-based algorithm in (Wei 2021) by a factor of $\mathcal{O}(\epsilon^{-1})$. This is due to the use of entropy regularization. In the stochastic level, by using advanced techniques to improve the estimation error bounds, our stochastic PE algorithm further improves the sample complexity by a factor of $\mathcal{O}(\epsilon^{-1.5})$. Please  refer  to  our remark for more details.
>
> **Q2:** Add intuitive illustration of the convergence rate under different choices of the batch sizes, the $T_k$, is preferred in comparison with [Wei et al., 2021].
>
> **A:** Thank you for your suggestion. In the revision, below Remark 2, we added one paragraph that presents a simplified convergence bound under constant choices of hyper-parameters. By comparing this simplified bound with that obtained in (Wei 2021), we can see that the outer iterations of our algorithm have a faster linear convergence (v.s. sublinear convergence in (Wei 2021)), and the estimation errors of our estimators have better dependence on the sample size than those of the estimators used in (Wei 2021).

---

> > ### Comment · Reviewer_8RN2 · 2021-11-30
> > **Thanks for the response**
> >
> > I thank the authors for their detailed response. I have read the discussions and have decided to keep my score as it is.

---

### Official Review · Reviewer_Hsr5 · 2021-11-02

**Correctness:** 4
**Technical Novelty And Significance:** 3
**Empirical Novelty And Significance:** Not applicable
**Recommendation:** 6
**Confidence:** 3

**Main Review:**

The problem the authors consider is important in competitive RL, where the authors show some improvement on the state-of-the-art result, so in general I think this is a good paper. To me the most significant contribution is the usage of new estimators in (12), (14), (15). Other than that, it seems like the algorithms come directly from Cen et al. However, the estimators seem not very novel to me. I meant to successfully apply them in this setting may not be trivial, but the estimators themselves should be used in other papers I think (or very similar ones). I guess the authors should do some survey and cite these paper when introducing these estimators.

Comparing with Wei et al. I think authors should discuss more on the contributions of each component to their improved bounds. Because the base algorithm is very different from Wei et al., I am not sure if the improvement comes from the estimators, EG, or the entropy regularization. Or put another way, is it possible to analyze the sample complexity of the algorithm in Wei et al. using the new estimators, or analyze the SPE algorithm with the estimator in Wei et al. I guess either way would help us to understand the significance of the estimators, as well as the other components.

Regarding the technical contributions, I think that is also one of my concern, First, the authors use Assumption 2, but I think Wei et al. didn't need this. If that is the case, I think the paper, strictly speaking, is not comparable with Wei et al. Secondly, I didn't notice the authors explain the technical difficulty when analyzing this algorithm in the main text. If it turns out it is straightforward to combine the new estimator with the algorithm in Cen et al., the paper becomes less interesting to me. Lastly, as highlighted in Wei et al., their algorithm is rational (converging to the opponent’s best response when it uses a stationary policy). It doesn't seem like SPE has this property, which I think may be important.



**Summary Of The Paper:**

The authors consider two-player zero-sum Markov games. The goal is to develop decentralized, model-free, symmetric algorithm. With entropy regularization and new estimators, the authors show the stochastic policy extragradient algorithms have these properties and the sample complexity improves the sate-of-the-art.

**Summary Of The Review:**

It is an interesting paper, which improves the best known bound in the field. However, I have some concerns regarding the novelty, comparison with the previous work, and technical significance. If the authors can address my concerns, I am happy to increase my score and recommend acceptance.

After the discussion period, my concerns are addressed so I changed my score accordingly.

---

> ### Author Response · Authors · 2021-11-19
> **Response to Review**
>
> Thank you very much for reviewing our manuscript and providing valuable feedback.  Below is a response to the review comments. We have submitted a revised version with all revisions marked in `red`. Please let us know if further clarifications are needed.
>
>
> **Q1:** The estimators seem not very novel. Should cite some related works.
>
> **A:** Thanks for the suggestion and we appreciate it. We agree that these estimators are not entirely new and their basic forms have been proposed/used in other related works. In the revision, we added a Remark 1 above the Section 4 to clarify this and cited several works that used estimators with similar structures. We also would like to point out one major technical difference, i.e., all these works (including (Wei 2021)) analyze the estimation error with independent samples, while our analysis addresses the additional bias induced by the correlated Markovian samples. Please refer to our response to Q3 for more clarifications on (Wei 2021)'s proof technique.
>
>
> **Q2:** Where does our improvement from Wei et al. (2021) come?
>
> **A:** Great question. In the revision, below Theorem 2,
> we added one paragraph that remarks on the improvement of sample complexity over (Wei 2021) in both the population level and stochastic level. To briefly summarize, in the population level, the original PE algorithm in (Cen 2021) improves the iteration complexity over that of the OGDA-based algorithm in (Wei 2021) by a factor of $\mathcal{O}(\epsilon^{-1})$. In the stochastic level, by using advanced techniques to improve the estimation error bounds, our stochastic PE algorithm further improves the sample complexity by a factor of $\mathcal{O}(\epsilon^{-1.5})$. Please refer to our remark for more details.
>
> **Q3:** (Wei 2021) does not need Assumption 2.
>
> **A:** This is an important question and thanks for pointing out. After carefully checking the published COLT version of (Wei 2021), we found the following fact. Although they claim to use Markovian samples in their estimators (see the paragraph above their eq.(7)), their proof of Theorem 3 in Appendix H.1. explicitly uses Azuma-Hoeffding’s inequality with independent transition samples. Hence, their results only apply to the independent setting, which actually corresponds to $t_{mix}=0$ and is stronger than our Assumption 2 (we assume $t_{mix}<+\infty$). We hope this answers your question.
>
> **Q4:** Technical difficulty is not explained in the main text.
>
> **A:** Thanks for the suggestion. In the revision, at the bottom of Page 8, we added a 'Technical Novelty' section that lists the major technical challenges and the technical novelties. As a brief summary, our analysis provides a tight analysis of the estimator errors induced by the stochastic estimators in two perspectives: (1) we apply concentration inequalities for dependent samples to bound the bias induced by Markovian samples and establish tight high-probability estimation error bounds; (2) By developing a special error decomposition strategy and leveraging the recursive structure of the estimators, we establish a contraction property of the estimation error, which is the key to tighten the estimation error bound. Finally, we develop a finite-time convergence analysis of the duality gap of the SPU algorithm under inexact value function estimations, this together with the developed tight estimator error bounds leads to the desired improved sample complexity.
>
>
> **Q5:** We did not prove that our algorithm is rational.
>
> **A:** Good question. In our revision, we have added Appendix G that proves the rationality of the policy extragradient (PE) algorithm and establishes the convergence rate. We sketch the proof as follows. Consider a Markov game where Player 1 plays the game while Player 2 follows a fixed policy $\pi^{(2)}$, which is essentially a single-player game where the Player 1 aims to find her policy $\pi_{*\tau}^{(1)}(s)=\arg\max_{\pi^{(1)}(s)} V_{\pi^{(1)},\pi^{(2)}}^{(\tau)}(s)$. With $\pi^{(2)}$ being fixed, this single-player PE algorithm takes much simpler updates, i.e., without the predictive update steps of Player 2, and the steps of Player 1 aim to maximize the entropy regularized function
>
> $\max_{\pi^{(1)}(s)} f_{\tau}(Q_k(s); \pi^{(1)}(s), \pi^{(2)}(s)):=\pi^{(1)}(s)^{\top} Q_k(s)\pi^{(2)}(s) +\tau \mathcal{H}\big(\pi^{(1)}(s)\big)-\tau \mathcal{H}\big(\pi^{(2)}(s)\big).$
>
> It can be proved straightforwardly that the simplified predictive update steps converge exponentially fast since both the solution and the policy obtained by predictive update have analytical expressions. Moreover, the final convergence rate of the single-player PE algorithm is the sum of both the exponential rate of the simplified predictive update steps and that of the value iteration. The proof can be directly extended to our stochastic PE algorithm by applying the stochastic estimation error bounds developed in Appendix B. Due to time limitation, we have not completed this part.

---

> > ### Comment · Reviewer_Hsr5 · 2021-11-30
> > **Thanks for the response**
> >
> > I thank the authors for their detailed response. Regarding Assumption 2, I think the results in Wei et al. are not limited to the case t_mix = 0. They require that two players interact with each other for a sequence of L steps, following a mixed strategy with a certain amount of uniform exploration, though. In this paper, the authors make an extra assumption (Assumption 2) but remove this extra coordination, which seems reasonable. Anyway, I think I still appreciate the authors' hard work. Overall, my concerns are addressed and I will increase my score accordingly.

---

> > > ### Author Response · Authors · 2021-11-30
> > > **Thanks for the update**
> > >
> > > Thanks a lot for the update before the deadline and we appreciate your additional feedback and comment. In the future revision, we will clarify the setting of the players' interaction adopted in Wei 2021.

---

### Decision · Program_Chairs · 2022-01-20

**Decision:**

Accept (Poster)

**Comment:**

This work presents  a new sample-based policy extragradient algorithm for finding an approximate Nash equilibrium in tabular two-player zero-sum Markov games with improved sample complexity guarantees. While originally the reviewers had concerns regarding the novelty and technical difficulty of the paper, these were successfully resolved during the rebuttal, and now all reviewers agree that this is an interesting contribution. Hence, I recommend acceptance of the paper.

In the final version the authors should make the following changes:
- Please mention early on (e.g., in the abstract and the introduction, as well as in the definition of the Markov game) that you consider a tabular problem (finite state and action spaces). Furthermore, it would be important to define informally the quantities in the bound in the abstract and when presenting Table 1.
- While not entirely uncommon, Assumption 1 is quite strong, requiring mixing for any policies. It would be great if the authors could also add a comment on this, emphasizing that this is the case, as well as explaining how weakening the assumption would introduce problems (as explained in the response to Reviewer RwGu).
- The comparison to the lower bound of Zhang et al. (2020) should also be included, as discussed in the response to Reviewer 5TU3.
- Please discuss Assumption 2 in relation to the work of Wei et al. (2021), and rephrase the relation to the latter paper accordingly, as promised in the discussion with Reviewer Hsr5.